# iPSC-derived models of PACS1 syndrome reveal transcriptional and functional deficits in neuron activity

Lauren Rylaarsdam [1], Jennifer Rakotomamonjy[1], Eleanor Pope[1] & Alicia Guemez-Gamboa [1] ✉

PACS1 syndrome is a neurodevelopmental disorder characterized by intellectual disability and distinct craniofacial abnormalities resulting from a de novo p.R203W variant in phosphofurin acidic cluster sorting protein 1 (PACS1). PACS1 is known to have functions in the endosomal pathway and nucleus, but how the p.R203W variant affects developing neurons is not fully understood. Here we differentiated stem cells towards neuronal models including cortical organoids to investigate the impact of the PACS1 syndrome-causing variant on neurodevelopment. While few deleterious effects were detected in PACS1[(+/R203W)] neural precursors, mature PACS1[(+/R203W)] glutamatergic neurons exhibited impaired expression of genes involved in synaptic signaling processes. Subsequent characterization of neural activity using calcium imaging and multielectrode arrays revealed the p.R203W PACS1 variant leads to a prolonged neuronal network burst duration mediated by an increased interspike interval. These findings demonstrate the impact of the PACS1 p.R203W variant on developing human neural tissue and uncover putative electrophysiological underpinnings of disease.

Neurodevelopmental disorders including autism spectrum disorder (ASD) and intellectual disability (ID) are highly prevalent in the population, both affecting 1–2% of individuals. Despite the pervasiveness of these disorders, few therapies exist due to a myriad of challenges including the genetic heterogeneity, historic inaccessibility of nervous tissue, and diagnosis typically occurring after the brain has developed. In certain instances, the genetic etiology of a neurodevelopmental disorder can be traced to a single source, providing a more constrained genetic background to study disease mechanisms and work towards therapy development. One such condition is PACS1-syndrome, an ID caused by a single recurrent missense variant.

PACS1 syndrome (MIM615009) was first reported in 2012 when two patients in an ID cohort presented with strikingly similar craniofacial abnormalities and comparable developmental delays[1]. Upon sequencing, both individuals were found to have the same de novo c.607 C > T variant in the gene encoding phosphofurin acidic cluster sorting protein 1 (PACS1), which results in an arginine to tryptophan

(R203W) substitution. Expression of the variant in a zebrafish model also produced craniofacial abnormalities, supporting the hypothesis that this was the variant responsible for patient symptoms[1]. Over 200 patients have since been identified with the same p.R203W pathogenic variant along with one report of a patient with an arginine to glutamine (R203Q) substitution[2–6]. All patients have ID and distinct craniofacial abnormalities in addition to frequent comorbidities, including epilepsy, ASD, hypotonia, feeding dysfunction, and cardiac disorders[3,7].

PACS1 is highly expressed in the embryonic brain and maintained at lower levels after birth, suggesting it plays an important role during neurodevelopment and continues to perform functions in adult tissues[8]. Genome-wide association studies have identified *PACS1* as a susceptibility gene in developmental delay[9], bipolar disorder[10], and severe early-onset obesity[11], further suggesting a critical role for *PACS1* in neural development. However, most previous research on PACS1 function has focused on a non-neuronal context, revealing multifunctional roles in the endosomal pathway and nucleus. PACS1 was

---

[1]Department of Neuroscience, Feinberg School of Medicine, Northwestern University, Chicago, IL, USA. ✉e-mail: alicia.guemez@northwestern.edu

first discovered for its capacity as a cytosolic sorting protein that identifies targets with a specific phosphorylated acidic cluster motif and facilitates localization to the *trans*-Golgi Network (tGN) in concert with adaptors such as AP-1 (adaptor protein 1)[12–15]. The p.R203W variant characteristic of PACS1 syndrome lies within this furin/cargo binding domain (FBR) and putatively modulates its affinity to targets. PACS1 was first reported and named in the context of shuttling furin to the tGN[13], but since has been found to interact with dozens of targets that perform diverse roles in vesicle trafficking, ion transport, gene expression, and other processes[14,16–26]. More recently, PACS1 was also discovered to have nuclear functions including mediating DNA replication repair[19,27], export of viral transcripts[28], RNA binding[20], and regulation of chromatin stability through interaction with histone deacetylase (HDAC) proteins[19]. However, the role of PACS1 during human neurodevelopment and the impact of the p.R203W variant have never been reported, and very few therapeutic options exist for patients[29].

In this study, we used induced pluripotent stem cell (iPSC)-derived neuronal models to begin to understand the pathogenic impact of the PACS1 syndrome-causing variant during cortical development. We performed single-cell RNA sequencing on dorsal cortical organoids at multiple time points and found that mature excitatory neurons containing the p.R203W variant have impaired expression of genes regulating synaptic function. To characterize these effects, we performed both calcium imaging in organoids and multielectrode array (MEA) recording of iPSC-derived neurons and found a prolonged network burst duration in PACS1$^{(+/R203W)}$ cells. This is the first study to investigate the impact of the PACS1 variant in a knock-in model of the developing human brain, revealing electrophysiological patterns that could be underlying patient symptoms and putative therapy targets for further characterization.

## Results

### PACS1 p.R203W variant does not impact cytoarchitectural organization of cortical organoids

To investigate how the p.R203W variant in PACS1 impacts developing neural cell types, we generated dorsal cortical organoids from induced pluripotent stem cell (iPSC) lines derived from three control (PACS1$^{(+/+)}$) and two affected (PACS1$^{(+/R203W)}$) individuals: Control 1 (C1) was derived from a healthy female, C2 from a healthy male, and C3 from an unaffected mother of a PACS1 syndrome patient; Affected 1 (A1) was derived from a male with PACS1 syndrome, and A2 from the affected daughter of the C3 donor. In addition, we utilized CRISPR/Cas9 gene editing technology to introduce the heterozygous variant which leads to p.R203W into the C1 background (CRISPR R203W) and correct it in the two patient lines (CRISPR A1 and CRISPR A2) for a total of three isogenic pairs (Fig. 1a; see Figs. S1–S3 for line characterization; see Table S1 for breakdown of lines used in all experiments). PACS1$^{(+/+)}$ and PACS1$^{(+/R203W)}$ iPSCs were then directed towards excitatory dorsal cortical organoids through exposure to an established sequence of growth factors (Fig. S4a)[30,31]. We chose to assess two early time points – day 40 and 80 – in order to capture the onset of *PACS1* upregulation, which is highly expressed early in development[8].

After 40 days of differentiation, organoids of both genotypes strongly expressed dorsal forebrain markers such as the telencephalic protein FOXG1, cortical protein EMX1, and dorsal precursor protein PAX6, but lacked mature upper cortical layer marker SATB2 as expected (Fig. 1b, c; Fig. S4b, c). PAX6$^+$FOXG1$^+$ proliferative cells spontaneously developed ventricular zone (VZ)-like rosette structures surrounded by an intermediate zone containing CTIP2$^+$ and TBR1$^+$ excitatory neurons, indicating our organoids were differentiating as had been previously described (Fig. 1b, c; Fig. S4b, c)[30,31]. By day 80, these rosette structures had largely dissipated in favor of more mature neuronal markers including SATB2 (Fig. 1b–e; Fig. S4b–e). There were

no differences in ratios of these markers between genotypes (Fig. 1d, e, shown by line in Fig. S4d, e), suggesting that the p.R203W variant did not affect cytoarchitectural organization of the organoid. This hypothesis was further supported by our initial analyses using neural precursor models, which did not show a difference in rates of proliferation or apoptosis (Fig. S5). All this suggests that p.R203W PACS1 does not affect early proliferation and specification dynamics during cortical development, which is in line with no consistent clinical reports of microcephaly or macrocephaly in PACS1 syndrome patients[3,7]. Furthermore, quantification of PACS1 protein levels in our organoid model showed no difference between PACS1$^{(+/+)}$ and PACS1$^{(+/R203W)}$ lines (Fig. 1f, g), suggesting that symptoms resulted from a change in PACS1 function and not abundance.

To simultaneously characterize the impact of the variant in multiple developing neural populations, we performed single-cell RNA sequencing (scRNAseq) in cortical organoids at day 40 and 88 time points (*Methods* and Fig. S6). We hypothesized the variant would have a transcriptomic impact as PACS1 has been shown to interact with histone deacetylase proteins[19]. Analysis of both time points revealed 16 total cell groups (Fig. 2a–c) with distinct transcriptomic signatures (Fig. 2d; Fig. S7) including radial glia at various states of proliferation and maturity, intermediate progenitors, newborn neurons, more developed immature neurons, and day 88-enriched mature excitatory neurons (Fig. 2a–c). Small clusters expressing features of inhibitory neurons, choroid plexus, and vascular leptomeningeal cells were also resolved (Fig. S7c) in accordance with publicly available databases where these markers (such as *DLX6, GAD2*, and *COL1A2*) are sporadically present in organoids produced with nearly identical protocols[32,33]. As expected, cell type composition drastically shifts to more mature cell types between day 40 and 88 (Fig. 2b, c, shown by line in Fig. S8a–d). The transcriptomes of the same cell type across different time points hierarchically cluster together (Fig. 2d), supporting our classification of cell groups across time points. Genotype is not a strong driver of clustering, indicating expression patterns of top variable features – e.g., canonical marker genes - are similar between conditions. In line with this observation, transcriptomic analysis showed no differences in the proportions of cell types between PACS1$^{(+/R203W)}$ and control organoids at either time point (Fig. 2b, c, shown by line in Fig. S8b, c). *PACS1* is expressed across all cell groups but is drastically enriched in more mature neuron populations (Fig. 2f, shown by line in Fig. S8e). Its low abundance in proliferative groups agrees with the absence of changes detected in PACS1$^{(+/R203W)}$ neural precursor models (Fig. S5). Altogether, initial characterizations of cell type proportions with scRNAseq is in consensus with our immunohistochemistry data and suggests the p.R203W variant does not impact proliferation and specification of dynamics of early cortical development, but instead likely impacts more mature neuronal cell types in which *PACS1* is highly expressed.

### PACS1$^{(+/R203W)}$ mature glutamatergic neurons have impaired expression of transcripts involved in ion transport and synaptic signaling

To uncover molecular pathways dysregulated by the PACS1 p.R203W variant in distinct developing neural cell types, we performed differential expression (DE) analysis of cortical organoid transcriptomes for all cell groups. DE analysis was run separately for day 40 and 88 samples due to minor but consistent differences in the number of average counts and features per cell (Fig. S8d). Our conservative method of identifying differentially expressed genes (DEGs) using tools from the Monocle3 package in conjunction with a DESeq2-based pseudobulk approach (see *Methods*) yielded a total of 277 combined DEGs across the 16 cell types and two time points (Data S1; Fig. S9). The majority of DEGs were identified in day 88 organoids, with proliferative radial glia having the highest number - potentially confounded by the highly dynamic state of a dividing cell - followed by mature glutamatergic

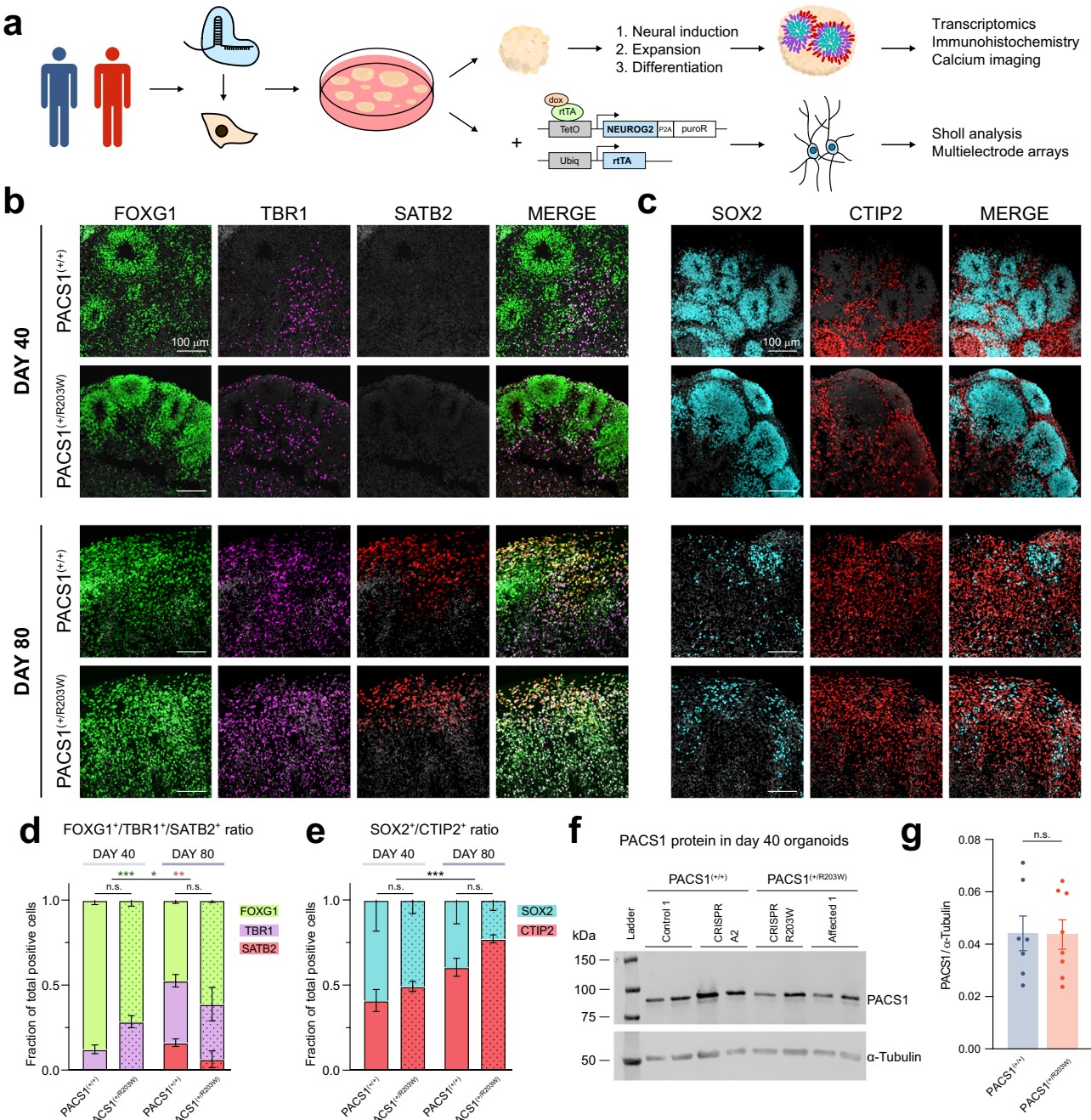

neuron groups (Fig. 3a). Top candidates include *CHODL*, which plays roles in mediating axon growth[34] and is upregulated in immature glutamatergic neurons at day 40; *LHX9*, a homeobox transcription factor expressed in pioneer neurons in the cerebral cortex[35] that is upregulated in newborn and CUX2+ glutamatergic neurons at day 40; *TRPC5*, a CNS-enriched $Ca^{2+}$-permeable cation channel[36] which is upregulated in all three mature glutamatergic neuron subpopulations at day 88; and *NTS*, a neuropeptide with known roles in modulating glutamatergic transmission[37] upregulated in newborn and SATB2+ glutamatergic neurons (see Data S1 for a complete DEG list). Examples of top candidates for each day and direction of change are shown on the RNA level in Fig. 3b and on the protein level in Fig. S10.

Instead of focusing on single candidates from our DEG list, we sought to determine overarching patterns and enriched pathways across DEG functions using gene ontology (GO) and Qiagen Ingenuity Pathway Analysis (IPA) tools. We plotted the hierarchical GO structure in donut charts (Fig. 3c–e)–where the inner and outer circles display

root and branch GO terms respectively, and width of the section denotes significance–in order to reduce redundancy and visualize ontology enrichment results as a whole. GO analysis of all 277 DEGs reveals top enrichments involve neuron projection development and ion transport (Fig. 3c). These terms are driven by DEGs upregulated in immature neurons for neuron projection development (Fig. 3d) from transcripts such as *SEMA3E, CHODL*, and *LHX2*; and DEGs upregulated in mature neuron groups for synaptic-related GO terms (Fig. 3e) from transcripts such as *SLC24A3, TRPC5*, and *CACNA2D3*. The clear pattern of synaptic transmission-related GO terms dominating results prompted us to further explore the relationship between mature neuron DEGs using IPA[38,39], as this cell group has the most likely potential for therapeutic interventions. IPA specifically implicated GABA receptor signaling as an upregulated pathway (Fig. S11) in addition to functional predictions of depressed neurotransmission and increased dendritic growth/branching (Fig. 3f). These enrichments resulted from aberrantly expressed transcripts such as *CNTNAP4*, a

**Fig. 1 | PACS1 p.R203W variant does not impact cytoarchitectural organization of cortical organoids. a** Experimental overview. Control, patient, and isogenic pluripotent cell lines were generated using CRISPR/Cas9 and differentiated to dorsal cortical organoids through exposure to a series of growth factors. Patient and control organoids were then assessed using immunohistochemistry, single-cell RNA sequencing, and calcium imaging. Additional downstream functional analyses were also performed with rapidly directed glutamatergic neurons. **b** Expression of telencephalic protein FOXG1, dorsal neuronal transcription factor TBR1, and upper layer marker SATB2 in day 40 and 80 organoids. Scale bar = 100 μm. DAPI is in greyscale. **c** Expression of neural precursor protein SOX2 and projection neuron marker CTIP2 in day 40 and 80 organoids. Scale bar = 100 μm. DAPI is in greyscale. **d** A quantification of (**b**). Mean ± SEM fraction of FOXG1⁺, TBR1⁺, or SATB2⁺ cells contributing to the total number of positive cells per field of view. Stains are different by day (FOXG1 $F_{1,9}$ = 24.004, $p_{adj}$ = 8.43E-04; TBR1 $F_{1,9}$ = 9.459, $p_{adj}$ = 0.013; SATB2 $H_1$ = 9.466, $p_{adj}$ = 0.004) but not genotype (FOXG1 $F_{1,9}$ = 0.061, $p_{adj}$ = 0.811; TBR1 $F_{1,9}$ = 1.811, $p_{adj}$ = 0.211; SATB2 $H_1$ = 0.183, $p_{adj}$ = 0.669) using a two-way ANOVA unless $p$ < 0.05 in a Shapiro test of normality, in which case a one-sided Kruskal-Wallis test was performed between conditions and adjusted using a Benjamini Hochberg correction (see *Methods*). $n$ = 3 organoids for each condition (Day 40: three C1, one CRISPR R203W, and two A1 organoids from four independent

differentiations; Day 80: two C2, one CRISPR A1, two CRISPR R203W, and one A1 organoids from three independent differentiations. See Table S1). Data from PACS1$^{(+/R203W)}$ lines is indicated by the dot pattern. Results per organoid are shown in Fig. S4d. **e** A quantification of (**c**). Mean ± SEM fraction of CTIP2⁺ or SOX2⁺ cells contributing to the total number of positive cells per field of view was significant between time points ($H_1$ = 12.808, $p_{adj}$ = 6.90E-04) but not genotypes ($H_1$ = 3.360, $p_{adj}$ = 0.067) using a one-sided Kruskal-Wallis test ($p_{Shapiro}$ = 0.003) and Benjamini Hochberg correction. $n$ = 7+ organoids per condition (Day 40: five C1, one C3, one CRISPR A1, one CRISPR A2, one CRISPR R203W, five A1, and one A2 organoids from four independent differentiations; Day 80: two C1, three C2, one C3, one CRISPR A1, three CRISPR R203W, and four A1 organoids from three independent differentiations. See Table S1). Data from PACS1$^{(+/R203W)}$ lines is indicated by the dot pattern. Results per organoid are shown in Fig. S4e. **f** Western blot for the PACS1 protein abundance in day 40 organoid samples. An uncropped version can be found in the Source Data file. **g** Quantification of the ratio of PACS1 to α-tubulin signal in day 40 organoids shown in (**f**). The bar represents mean ± SEM. $U$ = 26, $p$ = 0.867 using a two-sided Mann–Whitney test. $n$ = 7 PACS1$^{(+/+)}$ organoids (two C1, three C2, and two CRISPR A1) and $n$ = 8 PACS1$^{(+/R203W)}$ organoids (four CRISPR R203W and four A1) from two independent differentiations. See Table S1. Source data for (**d**–**g**) are provided in the Source Data file.

---

presynaptic neurexin involved in GABAergic transmission; *GPRIN3*, a GPCR involved in neuron projection development; *CACNA2D3*, a voltage-dependent calcium channel subunit; *GABRG3*, a GABA receptor type A subunit; and *TRPC5*, a CNS-enriched Ca²⁺-permeable cation channel. The expression levels of these genes are visually different between genotypes in feature plots (Fig. 3f, g; shown by line in Fig. S12), providing support for the validity of GO and IPA predictions.

### PACS1$^{(+/R203W)}$ organoids have increased GABAergic synaptic density but no differences in neural activity detected by calcium imaging

A clear pattern observed in DEGs was aberrant expression of genes involved in synaptic signaling in the day 88-enriched mature neuron populations (Fig. 3c, e–g), suggesting the presence of p.R203W PACS1 may affect electrophysiological properties. We next sought to investigate the functional consequence of this DE with calcium imaging (Fig. 4a). First, to see if synapses themselves were altered, we quantified glutamatergic synaptic abundance by staining for colocalization of a pre- and postsynaptic marker in day 80 organoids (Fig. 4b). We also probed for GABAergic synapse markers (Fig. 4c) as a top upregulated pathway predicted by IPA was GABAergic receptor signaling (Fig. S11). PACS1$^{(+/R203W)}$ organoids had no change in glutamatergic synaptic puncta (Fig. 4b, shown by line in Fig. S13a), but increased GABAergic synapse density (Fig. 4c, shown by line in Fig. S13a), suggesting that any functional effects could be related to GABAergic transmission.

To investigate the functional impact of upregulated genes involved in synaptic signaling and increased density of GABAergic synapses in PACS1$^{(+/R203W)}$ organoids, we sought to characterize electrical activity by measuring intracellular calcium dynamics using the Ca²⁺ sensor Fluo4 (Fig. 4a). We recorded both in spontaneous conditions and in the presence of GABA$_A$ antagonist bicuculline (BCU) as our IPA analysis implicated this pathway was upregulated in PACS1 syndrome pathology (Fig. S11). At both day 40 and 80, spontaneous tetrodotoxin (TTX)-susceptible (Fig. S13b) calcium events were recorded in PACS1$^{(+/+)}$ and PACS1$^{(+/R203W)}$ organoid sections (Fig. 4d). Clear differences were observed between the two time points, including increased amplitude, density of active cells, and Ca²⁺ event width (Fig. 4e–g, shown by line in Fig. S13c–e), indicative of maturing neuronal tissue. However, no differences were detected in any of our measured variables between PACS1$^{(+/+)}$ and PACS1$^{(+/R203W)}$ groups (Fig. 4c–i, shown by line in Fig. S13c–f). This suggests the p.R203W variant does not affect gross measures of single-neuron activity, as network-level metrics were only sparsely collected by the calcium imaging technique. A significant increase in calcium event frequency was prompted by application of BCU, but PACS1$^{(+/+)}$ and PACS1$^{(+/R203W)}$

organoids responded similarly (Fig. 4i, shown by line in Fig. S13f), indicating no functional difference in GABA$_A$ receptor activity in this organoid model system.

### Multielectrode arrays reveal PACS1$^{(+/R203W)}$ glutamatergic neurons display prolonged network bursting patterns and decreased synchrony

To complement our calcium imaging data with an alternative approach that offered much higher temporal resolution and throughput, we shifted our efforts towards recording electrical activity with multielectrode arrays (MEAs). In an effort to reduce variability, we focused on the glutamatergic population in which the DEGs related to synaptic signaling were identified (Fig. 3e–g), since blocking GABA$_A$ receptors with BCU did not differentially affect Ca²⁺ event rate in PACS1$^{(+/R203W)}$ organoids (Fig. 4i). Mature excitatory neurons from all three isogenic iPSC pairs were generated by virally enabling dox-inducible expression of Neurogenin 2 (NEUROG2)[40], a protocol chosen due to the strong overlap of *NEUROG2* on the top three most mature neuron clusters of interest (Fig. 5a), the homogeneity of the differentiation product, and the ability to culture in 2D for long periods of time given the absence of proliferative cells that contaminate small molecule differentiation methods. Neurons generated with this approach (Fig. 5b) express PACS1 (Fig. S14a), develop complex morphology (Fig. 5c, d), and express neuronal forebrain markers including CUX1 and FOXG1 (Fig. S14b, c). Sholl analysis was performed since neuron projection development was a strongly implicated pattern in DEGs (Fig. 3c, d, f) but did not reveal differences in dendritic arborization (Fig. 5d–g), suggesting any electrophysiological differences between genotypes would be due to synaptic or ionic alterations.

To record electrical activity, neurons were cultured along with rat astrocytes on an MEA system over a two-month time course. Synapsin-expressing cells were counted at the end of each of three experiments to ensure any differences could not be attributed to discrepancies in density (Fig. 5h). Many parameters (Fig. 5i–o) were the same between PACS1$^{(+/+)}$ and PACS1$^{(+/R203W)}$ neurons, including mean firing rate (Fig. 5i) and progression of active electrodes over the recording period (Fig. 5j). Number of near-synchronous "network" bursting events across multiple electrodes within a well differed when analyzed as a whole, but not at any individual time point (Fig. 5k). However, PACS1$^{(+/R203W)}$ neurons displayed a striking increase in the duration of network bursts that was consistent across all isogenic pairs and differentiations (Fig. 5l, Fig. S15a). Single-electrode bursts were also prolonged (Fig. S15b), but to a lesser extent than network bursts. Despite the duration doubling, the number of spikes within a burst remained the

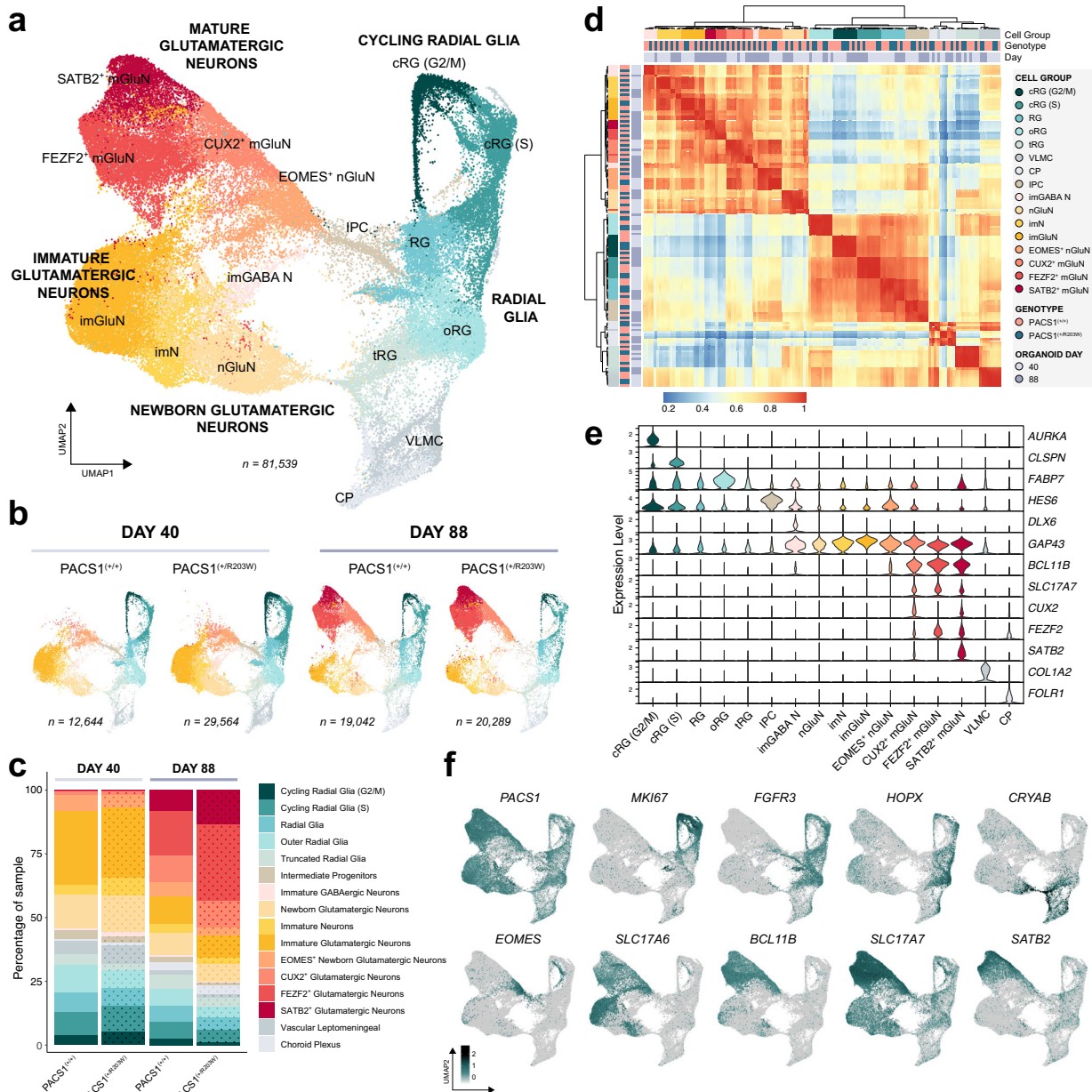

**Fig. 2 | Single-cell RNA sequencing analysis reveals progression of neuronal maturity in cortical organoids but no difference in cell type composition between genotypes. a** Uniform manifold approximation and projection (UMAP) dimensionality reduction of all sequenced cells reveals distinct populations, including: cycling radial glia in G2/M and S phases (cRG (G2/M), cRG (S)); non-cycling radial glia (RG); outer and truncated radial glia (oRG, tRG); vascular lepto-meningeal cells (VLMC); choroid plexus (CP); intermediate progenitors (IPC); newborn glutamatergic neurons (nGluN, EOMES+ nGluN); immature neurons of a mixed population (imN), GABAergic immature neurons (imGABA N), glutamatergic immature neurons (imGluN); and mature glutamatergic neuron groups with high expression of layer II-III marker *CUX2* (CUX2+ mGluN), subcortical projection marker *FEZF2* (FEZF2+ mGluN), or upper layer neuron marker *SATB2* (SATB2+ mGluN). Data in (**a**–**f**) is shown from 9 organoids total across four independent differentiations (Day 40: one C1, one C3, one CRISPR R203W, one A1, and one A2 organoid; Day 88: two C2, one CRISPR R203W, and one A1 organoid. See Table S1). **b** UMAP reductions divided by genotype and time point. Individual samples are shown in Fig. S8a. **c** Bar chart showing the cell type composition of samples in each

genotype and time point. Individual samples are shown in Fig. S8b. Data from PACS1(+/R203W) lines is indicated by the dot pattern. **d** Hierarchical clustering of Pearson's correlation of variable feature expression across sample and cell type. **e** Violin plots of feature expression. G2/M phase marker *AURKA* is restricted to cRG (G2/M) group and S phase marker *CLSPN* is strongly enriched in cRG (S). Radial glia marker *FABP7* is enriched in proliferative groups. *HES6* is enriched in IPCs. *DLX6* expression is restricted to the imGABA N group. *GAP43* is expressed across young and mature neurons while *BCL11B* and *SLC17A7* are enriched in mature glutamatergic neurons. Enrichment of *CUX2, FEZF2,* and *SATB2* define mGluN subpopulations. VLMC-like cells express collagen marker *COL1A2* and CP exclusively express *FOLR1*. **f** Feature plots of all cells from both genotypes in UMAP space show expression of *PACS1*, proliferative marker *MKI67*, RG marker *FGFR3*, oRG marker *HOPX*, tRG marker *CRYAB*, IPC/newborn neuron marker *EOMES*, immature neuronal vesicular glutamate transporter *SLC17A6* (vGLUT2), more mature neuronal transcripts *BCL11B* and *SLC17A7* (vGLUT1), and upper cortical layer marker *SATB2*. Scale is in units of $\log_{10}(\text{value} + 0.1)$.

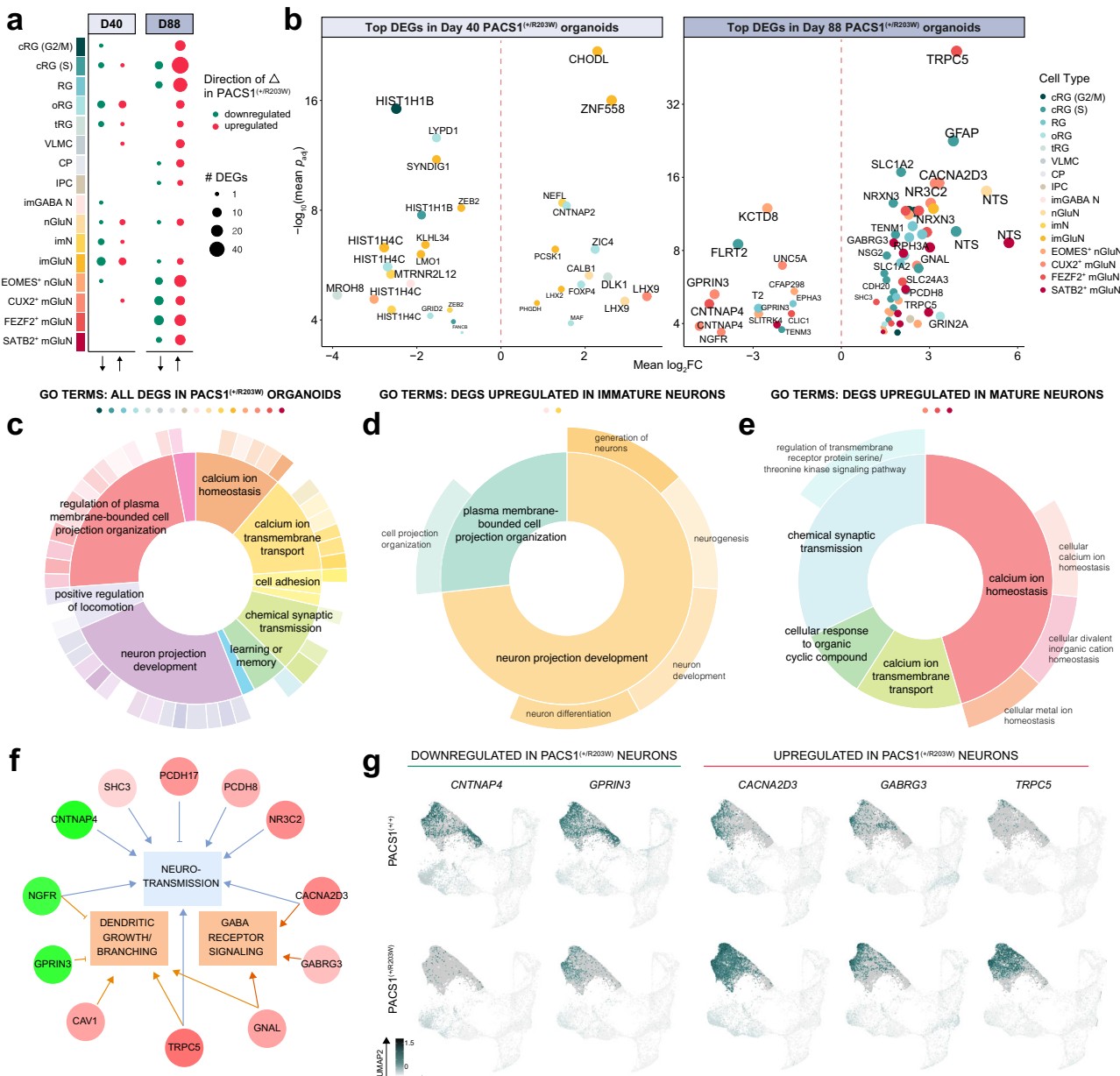

**Fig. 3 | PACS1$^{(+/R203W)}$ mature glutamatergic neurons have impaired expression of transcripts involved in ion transport and synaptic signaling. a** Dot plot showing the number of differentially expressed genes (DEGs) identified in each cell group at the two time points. Size indicates the number of DEGs while color indicates direction of change. **b** A volcano plot showing select top DEGs. Color indicates cell type and size correlates to a combined metric of log$_2$FC and $p_{adj}$. All 277 DEGs are shown in Fig. S9, Data S1, and the Source Data file. y axis is shown on a log$_{10}$ scale with a cutoff of $p_{adj} < 0.05$ for both DEG identification methods used. **c–e** Nested pie chart showing root and branch significant gene ontology enrichments with the colored dots above indicating which cell types were included in the analysis and width of the pie correlating to -log($p$) of the GO term. Statistics were calculated using the Fisher's Exact test with correction facilitated within the elim algorithm (see Methods). GO terms for DEGs up and downregulated across all PACS1$^{(+/R203W)}$ cell types are shown in (**c**); GO terms for DEGs upregulated in PACS1$^{(+/R203W)}$ immature neurons are shown in (**d**); and the GO terms for DEGs upregulated in PACS1$^{(+/R203W)}$ mature neuron groups are shown in (**e**). **f** Functional enrichment and pathway analysis results of PACS1$^{(+/R203W)}$ DEGs in the mature glutamatergic neuron groups generated with IPA analysis. Gene color indicates fold change with green being downregulated and red upregulated in PACS1$^{(+/R203W)}$ models. Orange indicates IPA predicts activation of the pathway, and blue repression. $p_{adj} = 0.0035$ for neurotransmission, $p_{adj} = 0.019$ for dendritic growth/branching, and $p_{adj} = 0.016$ for GABA receptor signaling using a right-tailed Fisher's Exact test with a Benjamini-Hochberg correction for multiple testing. The GABA receptor signaling term refers to analysis of upregulated genes while the other terms include both up and downregulated genes. **g** Feature plots of top PACS1$^{(+/R203W)}$ DEGs implicated in IPA analysis. Cells have been down-sampled so the same number is shown for each genotype. Data is shown from two C2, one CRISPR R203W, and one A1 day 88 organoid from one independent differentiation. Plots for each sample are in Fig. S12. Scale is in units of log$_{10}$(value + 0.1).

same (Fig. 5m), with the difference arising from an increase in the interspike interval within bursts (Fig. 5n) and a delay in attainment of max spike rate (Fig. S15c, d). This suggests that PACS1$^{(+/R203W)}$ neurons experience aberrant regulation of ion flux during synaptic transmission events that delay the fast afterhyperpolarization (fAHP) period, which occurs on a 2–5 ms timescale[41]. DEGs such as *KCNMB2, TRPC5*, or *CACNA2D3* could certainly underlie this phenotype. Closer investigation of spike trains (Fig. 5p, q) also reveals a consistent phasic, multi-modal burst pattern in PACS1$^{(+/R203W)}$ neurons, implicating an additional component to fAHP at play such as change in intracellular calcium ion

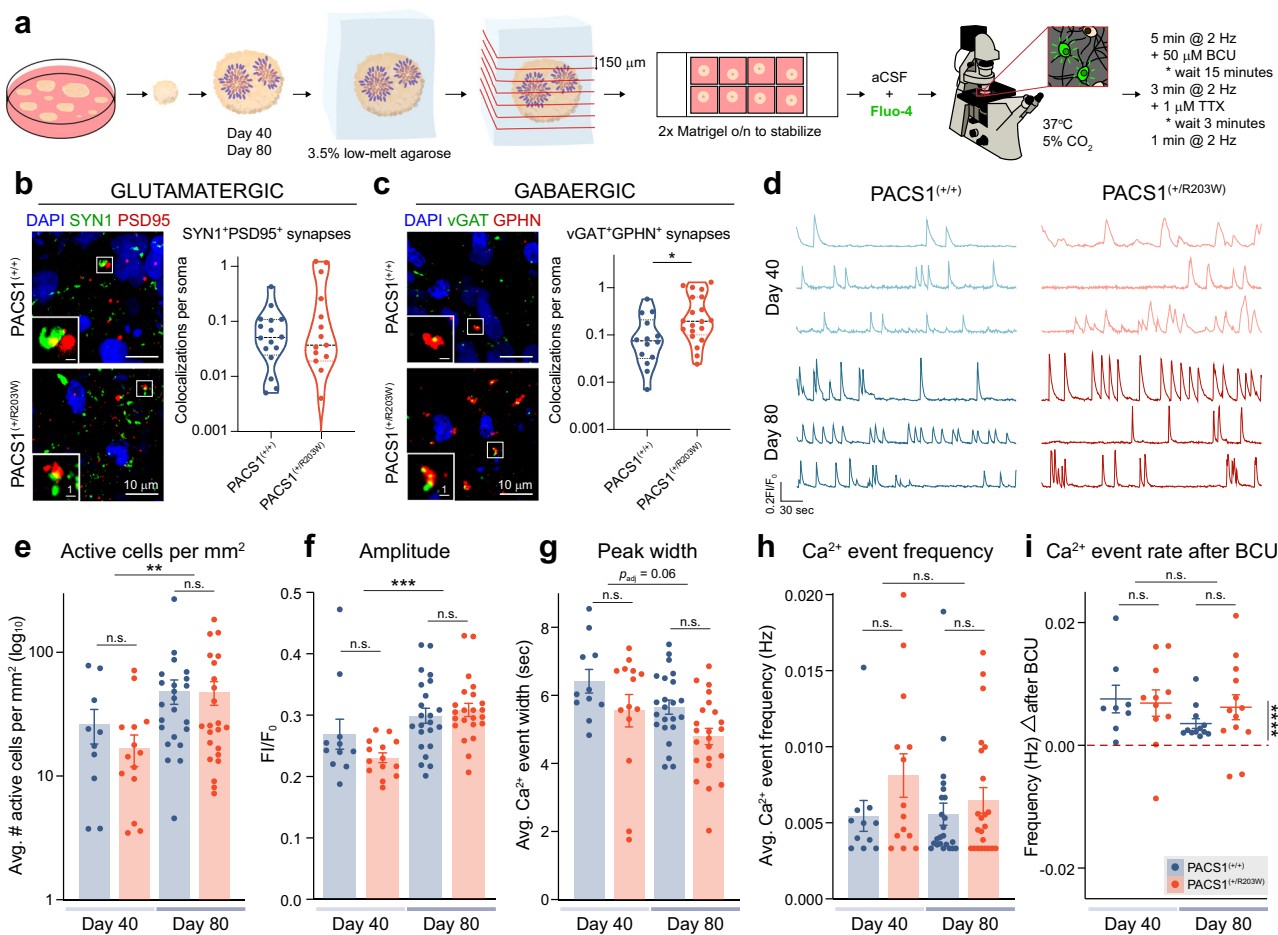

**Fig. 4 | PACS1$^{(+/R203W)}$ organoids have increased GABAergic synaptic density but no differences in neural activity detected by calcium imaging. a** Experimental workflow for calcium imaging in organoids from two independent differentiations. **b** Expression of glutamatergic synapse markers in day 79 organoid sections. Co-localization of pre-synaptic marker synapsin 1 (SYN1) and post-synaptic density protein 95 (PSD95) was used to identify glutamatergic synapses. Scale bar = 10 μm in the main panels and 1 μm in the zoomed panel. The median value is indicated by dashed lines and the dotted lines represent quartiles. Data is from $n = 16$ total sections across two C2, one C3, and one CRISPR A1 PACS1$^{(+/+)}$ organoids and $n = 15$ total sections across two CRISPR R203W, one A1, and two A2 PACS1$^{(+/R203W)}$ organoids from three independent differentiations. See Table S1. $U = 118$, $p = 0.953$ using a two-sided Mann-Whitney test. y axis is on a $\log_{10}$ scale. Individual cell line results are shown in Fig. S13a. **c** Expression of GABAergic synapse markers in day 79 organoid sections. Co-localization of pre-synaptic marker vesicular GABA transporter (vGAT) and post-synaptic marker Gephyrin (GPHN) was used to identify synapses. Scale bar = 10 μm in the main panels and 1 μm in the zoomed panel. The median value is indicated by dashed lines and the dotted lines represent quartiles. Data is from $n = 14$ sections across two C2, one C3, and one CRISPR A1 PACS1$^{(+/+)}$ organoids and $n = 20$ sections across two CRISPR R203W, one A1, and two A2 PACS1$^{(+/R203W)}$ organoids from two independent differentiations. See Table S1. $U = 71$, $p = 0.015$ calculated using a two-sided Mann–Whitney test. y axis is on a $\log_{10}$ scale. Individual cell line results are shown in Fig. S13a. **d** Representative calcium traces of PACS1$^{(+/+)}$ (blue) and PACS1$^{(+/R203W)}$ (red) cells over 300 s of recording. **e** Bar chart showing mean ± SEM of number of active cells per mm² of an organoid section. Scale is $\log_{10}$ for clarity. There is an effect of day ($F_{1,23} = 8.576$, $p_{adj} = 0.008$) but not genotype ($F_{1,23} = 0.482$, $p_{adj} = 0.495$) using a two-way ANOVA test with a Tukey HSD correction on log-transformed values. Dots represent sections from $n = 5$ Day 40 PACS1$^{(+/+)}$ organoids (three C1 and two CRISPR A1), $n = 6$ Day 40 PACS1$^{(+/R203W)}$

organoids (three CRISPR R203W and three A1), $n = 8$ Day C2 80 PACS1$^{(+/+)}$ organoids, and $n = 7$ Day 80 PACS1$^{(+/R203W)}$ organoids (four CRISPR R203W and three A1) from two independent differentiations for (**e–h**). See Table S1 for more details on number of replicates per line used in each experiment. Data for each cell line is shown in Fig. S13c. **f** Bar chart displaying the mean ± SEM of Ca²⁺ event amplitude. There is an effect of day ($F_{1,23} = 15.481$, $p_{adj} = 6.815E-04$) but not genotype ($F_{1,23} = 0.484$, $p_{adj} = 0.494$) using a two-way ANOVA test with a Tukey HSD correction. Data for each cell line is shown in Fig. S13d. **g** Bar chart displaying the mean ± SEM of calcium event width. There may be an effect of day ($F_{1,23} = 3.811$, $p_{adj} = 0.064$) but not genotype ($F_{1,23} = 2.648$, $p_{adj} = 0.117$) using a two-way ANOVA test with a Tukey HSD correction. Data for each cell line is shown in Fig. S13e. **h** Bar chart showing mean ± SEM of average frequency of calcium events in untreated conditions. There is no effect of genotype ($H_1 = 0.111$, $p_{adj} = 0.815$) or day ($H_1 = 0.055$, $p_{adj} = 0.815$) on event frequency using a one-sided Kruskal-Wallis test for genotype and day followed by a Benjamini Hochberg correction as data was non-parametric ($p_{shapiro} = 9.67E-04$). **i** Plot showing mean ± SEM of the effect of GABA$_A$ antagonist bicuculline (BCU) on calcium event frequency. While BCU has a strong effect on event frequency ($t_{22} = 6.473$, $p = 1.63E-06$ calculated using a one sample two-sided $t$ test), there is no effect of day ($F_{1,20} = 2.430$, $p_{adj} = 0.135$) or genotype ($F_{1,20} = 1.441$, $p_{adj} = 0.244$) using a two-way ANOVA test with a Tukey HSD correction. Dots represent the difference in event frequency in paired recordings from organoid sections following a 15 min incubation with 50 μM bicuculline. Statistics were calculated from $n = 5$ Day 40 PACS1$^{(+/+)}$ organoids (three C1 and two CRISPR A1), $n = 6$ Day 40 PACS1$^{(+/R203W)}$ organoids (three CRISPR R203W and three A1), $n = 6$ Day C2 80 PACS1$^{(+/+)}$ organoids, and $n = 6$ Day 80 PACS1$^{(+/R203W)}$ organoids (three CRISPR R203W and three A1) from two independent differentiations. Data for each cell line is shown in Fig. S13f. Legend also applies to (**e–g**). Source data for (**b–c**) and (**e–i**) can be found in the Source Data file.

homeostasis, a top GO term for DEGs upregulated in PACS1$^{(+/R203W)}$ neurons (Fig. 3e).

In addition to drastic deficits in network bursting, PACS1$^{(+/R203W)}$ neurons also display aberrations in synchrony. As neurons matured on

the MEA system, an increasing percentage of spikes occurred within network burst events. While this steadily increased in PACS1$^{(+/+)}$ neurons, PACS1$^{(+/R203W)}$ neurons lagged (Fig. S15e). This value converged by the end of our recording period, but PACS1$^{(+/R203W)}$ neurons maintained

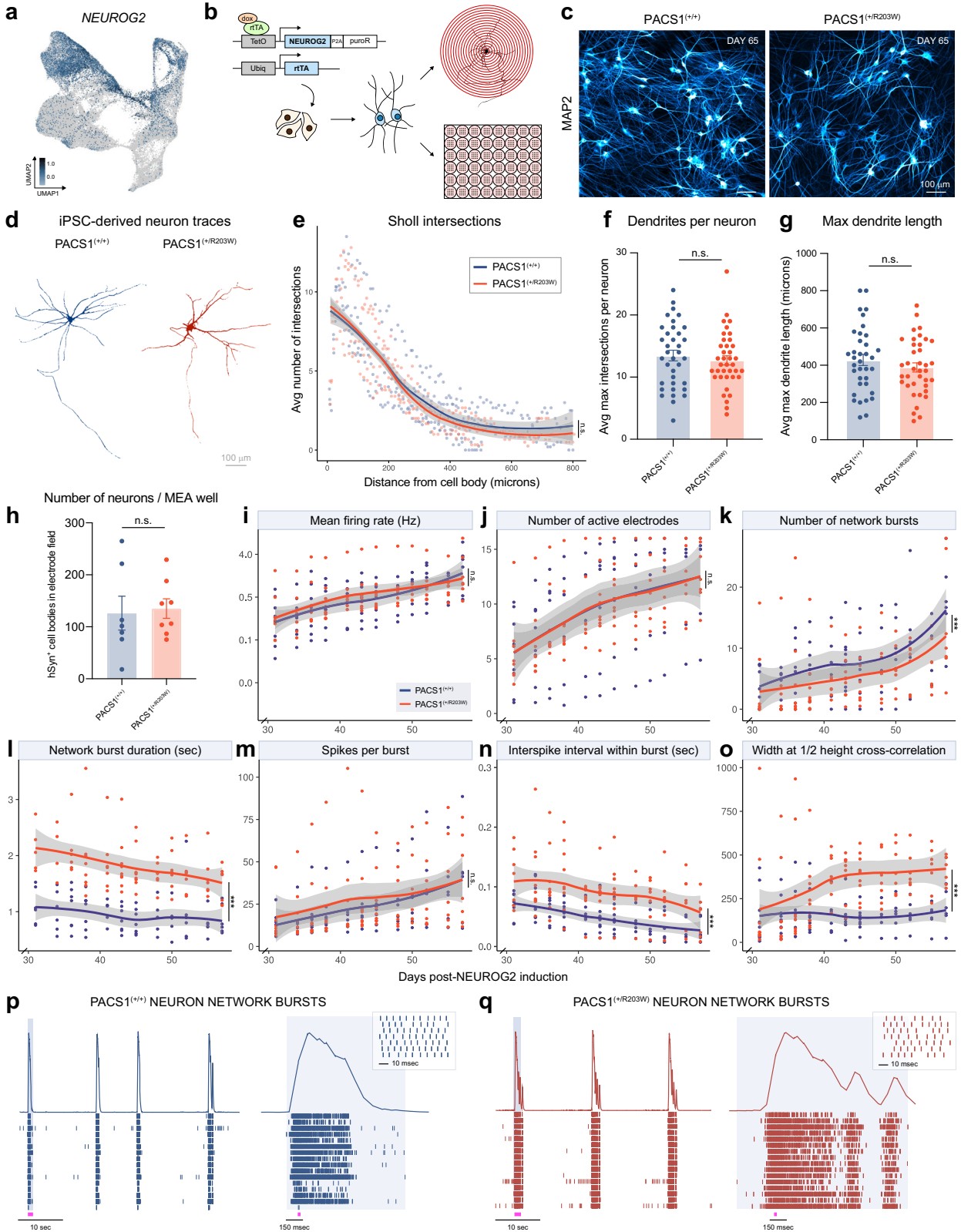

a wider cross-correlogram (Fig. 5o), indicating a weaker relationship between firing patterns than in controls. The proper coordination of synchronous neuron activity during development is critical for appropriate circuit formation and its disruption could underlie PACS1 syndrome patient phenotypes, as has already been shown for other ASD and ID models[42–49].

## Discussion

Here we present the first study investigating the impact of the PACS1 p.R203W variant on developing human neural tissue. Single-cell RNA sequencing of PACS1(+/R203W) cortical organoids revealed deficits in the expression of ion channels in mature glutamatergic neuron populations, and subsequent functional analysis exposed a drastic increase in

**Fig. 5 | PACS1$^{(+/R203W)}$ excitatory neurons display prolonged network bursting patterns and decreased synchrony. a** *NEUROG2* expression in single-cell RNA sequencing data. Scale is in units of $\log_{10}$(value + 0.1). **b** Experimental protocol for rapid induction of glutamatergic neurons using a dox-inducible system enabling overexpression of NEUROG2. **c** MAP2 expression in neurons produced with the NEUROG2 overexpression protocol 65 days after viral induction. Scale bar = 100 μm. **d** Representative day 46 neurons transfected and traced for Sholl analysis. Scale bar =100 μm. The blue and orange colors are used to represent data from PACS1$^{(+/+)}$ neurons and PACS1$^{(+/R203W)}$ neurons respectively in all remaining panels. **e** Number of neurite intersections with concentric circles of increasing distance from the cell body. Dots are averages from $n$ = 4 PACS1$^{(+/+)}$ cell lines (one C1, two CRISPR A1, and one CRISPR A2 differentiations) and $n$ = 4 PACS1$^{(+/R203W)}$ cell lines (one CRISPR R203W, two A1, and one A2 differentiations) from two independent experiments. The smoothed curve shows genotype averages with error bands indicating a 95% confidence interval. $p$ > 0.05 for all micron bins using a two-sided student's $t$ test. **f** Average dendrites per neuron. Dots are neuronal metrics from $n$ = 4 PACS1$^{(+/+)}$ cell lines (one C1, two CRISPR A1, and one CRISPR A2 differentiations) and $n$ = 4 PACS1$^{(+/R203W)}$ cell lines (one CRISPR R203W, two A1, and one A2 differentiations) from two independent experiments; bar represents the mean ± SEM. See Table S1 for further details regarding which cell lines were used. $t_{5.6}$ = 0.193, $p$ = 0.854 was calculated using a Welch two sample two-sided $t$ test. **g** Maximum dendritic length as determined by the furthest concentric circle from the cell body with more than 1 intersection, to avoid axon-related skewing. Dots are neuronal metrics from $n$ = 4 PACS1$^{(+/+)}$ cell lines (one C1, two CRISPR A1, and one CRISPR A2 differentiations) and $n$ = 4 PACS1$^{(+/R203W)}$ cell lines (one CRISPR R203W, two A1, and one A2 differentiations) from two independent experiments; bar represents the mean ± SEM. $t_{4.7}$ = 0.180, $p$ = 0.865 was calculated using a Welch two sample two-sided $t$ test. **h** Dots are number of synapsin$^{+}$ cell bodies visible within the multielectrode array (MEA) electrode field averaged by cell line ($n$ = three C1, three CRISPR A1, one

CRISPR A2, three CRISPR R203W, three A1, and two A2 differentiations) for each of three independent experiments; bar represents the mean ± SEM. $t_{13}$ = 0.243, $p$ = 0.812 using a two-sided $t$ test. **i–o** MEA metrics for neurons days 30–57 of differentiation. Dots represent averages from each cell line, while the smoothed curve shows genotype averages with error bands indicating a 95% confidence interval. $n$ = 8 PACS1$^{(+/+)}$ lines (four C1, three CRISPR A1, and one CRISPR A2) and $n$ = 9 PACS1$^{(+/R203W)}$ lines (four CRISPR R203W, three A1, and two A2) from three independent experiments. $p$ values were calculated by conducting an ANOVA between a linear regression with and without genotype included as a covariate. In instances where $p$ values are reported between specific time points, a Kruskal-Wallis test was run. **i** Mean firing rate in Hz. Genotype had no effect ($F_{0.2}$ = 0.933, $p$ = 0.395). y axis is on a $\log_2$ scale for clarity. **j** Number of electrodes with ≥1 spikes/minute. Genotype had no effect on progression of active electrodes ($F_{0.2}$ = 0.090, $p$ = 0.914). **k** Number of network bursts over 5 min, as defined by ≥18% electrodes simultaneously active within a 100 msec window. Genotype had a significant effect on number of network bursts collectively ($F_{0.2}$ = 7.655, $p$ = 5.63E-04), but no individual time point achieved significance. **l** Network burst duration. Genotype had a significant effect on network burst duration ($F_{0.2}$ = 123.17, $p$ < 2.2E-16). Results by isogenic pair and differentiation are shown in Fig. S15a. **m** Spikes per burst. Genotype had no effect on spikes per burst ($F_{0.2}$ = 1.663, $p$ = 0.192). **n** Interspike interval within a burst. Genotype had a significant effect on interspike interval ($F_{0.2}$ = 8.622, $p$ = 2.35E-04). **o** Width of a correlogram at ½ height of cross-correlation. Genotype had a significant effect ($F_{0.2}$ = 37.918, $p$ = 3.11E-15). Individual time points reached significance at day 41 onwards. Example of a neuron activity raster plot across 16 electrodes in PACS1$^{(+/+)}$ (**p**) or PACS1$^{(+/R203)}$ (**q**) wells. Above the raster plots is a spike histogram showing the collective density of activity over time. A network burst is highlighted in the light gray panels, which is zoomed in to the right. The pink dash below indicates the approximate location of the subsequent zoom panel. Source data for (**e–o**) can be found in the Source Data file.

---

network burst duration. This could certainly contribute to the epilepsy and ASD/ID experienced by patients, as glutamatergic neuron network activity is impaired in many human iPSC-derived models of epilepsy and ASD including *KCNQ2, MECP2, FMR1, KCNT1, ATRX, AFF2, SCN2A, ASTN2, CNTN5, EHMT2*, and idiopathic cases[50–61]. Most of these studies do not report how burst duration is affected, with two exceptions: a p.P924L KCNT1 variant was associated with shorter bursts[57], and no change in burst duration was observed in 16p11.2 duplication or deletion models[61]. The increased network burst duration and decreased synchrony in our system could be related to patient EEG reports (though rare and variable) for both PACS1 syndrome and a strikingly similar PACS2 syndrome of "excess discontinuity", "generalized slowing", and "slow wave bursts"[62–64]. However, these are very different systems, and it is possible that variable firing patterns in neurons initiate epileptic activity but are not reflected in EEG recordings. Further experiments with more advanced model systems would be needed to fully make this connection.

Which channels are contributing the most to the increased interspike interval in mature PACS1$^{(+/R203W)}$ glutamatergic neurons can be hypothesized by our DEG analysis, but should be further investigated with pharmacological intervention studies and patch clamp recordings. *KCNMB2*, a DEG upregulated in multiple cell types, is a promising candidate as it encodes an auxiliary subunit to BK channels, which are direct critical modulators of fAHP duration[65]. Given the phasic nature of the aberrant burst patterns in PACS1$^{(+/R203W)}$ neurons, DEGs involved in calcium regulation could also underly the mechanism, as intracellular $Ca^{2+}$ levels affect fast, medium, and slow AHP[66]. Examples include *CACNA2D3* or *TRPC5*, the top-most upregulated gene in mature PACS1$^{(+/R203W)}$ neuron groups. *TRPC5* is a promising candidate as two reports have demonstrated a direct role of PACS1 in trafficking TRP channels[1,16] and TPRC5 activity has been suggested to prolong group I metabotropic glutamate receptor-mediated epileptiform discharges[67,68]. The p.R203W variant could impair binding to TRPC5 and result in a similar mechanism to what has been shown with TRPP2, which aggregates in the cell membrane upon disruption of the acidic cluster sorting motif identified by PACS1, resulting in increased ion

channel activity[16]. However, other top DEGs including *CNTNAP4, SLC1A2*, and *GABRG3* have also been shown to underly irregular firing patterns in the context of epilepsy or autism and could be contributing to the disease mechanism[69–74]. It should finally be considered that protein-level interactions could play a larger role as PACS1 has been shown to form a complex with Wdr37 and regulate endoplasmic reticulum $Ca^{2+}$ homeostasis in lymphocytes[24]. Patch clamp recording in combination with pharmacological modulation of candidate ion channels are critical future directions in order to better understand how p.R203W PACS1 impacts action potential dynamics, potentially revealing targets for therapeutic intervention.

Since the work described here is a pioneering effort to understand the p.R203W variant's impact on the developing nervous system, there is much more room for further understanding steps between the presence of the variant and functional deficits. We did not find the variant impacted PACS1 abundance on the RNA or protein level, so it is likely that some kinetic or steric property could be altered by the introduction of the variant. This hypothesis is supported by the observation that the c.607 C > T substitution results in a positive arginine being replaced by a large, nonpolar tryptophan within the cargo-binding FBR of PACS1, which in silico modeling with AlphaFold predicts would strengthen its interaction with targets. Future studies will involve crystallography to resolve the structure of the FBR in the mutant and control PACS1, immunoprecipitation assays in neurons to establish if binding partners are lost or acquired upon introduction of the variant, and immunohistochemistry analysis of PACS1 and its targets to assess how the variant impacts subcellular localization in order to better inform therapeutic development.

In addition, we reported an increase in GABAergic synaptic density and an upregulation of DEGs involved in GABA signaling such as *GABRG3*, but largely focused on deficits in glutamatergic neurons after application of one GABA$_A$ receptor antagonist BCU did not differentially affect PACS1$^{(+/R203W)}$ organoids. The role of GABAergic interneurons in PACS1 syndrome pathology is a key area of future exploration as an abundance of literature points to their role in ASD[42,75–80]. It is possible no differential response to BCU was observed

because GABA$_B$ receptors play a larger role in PACS1 syndrome pathology, or the relatively low abundance of GABAergic neurons in our dorsal forebrain model dilutes our ability to detect GABAergic mechanisms of disease. Ultimately, our model system was not best-suited to interrogate GABAergic interneuron properties, as both our organoid and neuron differentiation protocols primarily produce excitatory neurons. A critical future direction is therefore utilizing additional models – such as ventral forebrain organoids, or a coculture system with excitatory and *ASCL1/DLX2*-derived inhibitory neurons. Applying a diverse panel of agonists and antagonists to these systems will facilitate a better understand the relative contributions of glutamatergic and GABAergic neurons to PACS1 syndrome pathology[81]. Eventually, complementary studies with a model that has functional circuits between diverse brain regions are also needed to better understand how phenotypes observed here correlate to patient symptoms. For example, increased density of GABAergic synapses is initially a counterintuitive mechanism of epilepsy, but it could be temporally or spatially restricted in a manner that initiates epileptiform activity in vivo. Collaborative efforts within the PACS1 Syndrome Research Foundation[29] are therefore underway to generate a mouse model in order to better characterize the electrophysiological deficits across different brain regions, the resulting cognitive impact, and the safety of candidate therapies.

This is the first study to investigate the impacts of the p.R203W variant in developing human neural tissue and reveals many novel insights into PACS1 syndrome pathology. We have shown that the variant has a disproportionate impact on more mature neuron populations, providing hope that an intervention could be effective even after diagnosis. Initial therapeutic development efforts will focus on identification of pharmacological modulators to rescue the network bursting phenotype, or generation of antisense oligonucleotides to block translation of the p.R203W PACS1 mRNA altogether. The PACS1 Syndrome Research Foundation is working hard to coordinate the pursuit of all viable treatment avenues. Unfortunately, most other genetic origins of ASD and ID are much more complex, and developing therapies for each combination of variants is not feasible. A momentous leap forward for the field would be to develop a way to target common phenotypes across a multitude of etiologies, such as correction of convergent aberrant neuron firing patterns. This is a monumental task, but the potential provides hope for PACS1 patients and millions of others suffering from ID and ASD.

## Methods

### Generation and maintenance of induced pluripotent stem cell lines

This research was performed in compliance with relevant ethical regulations and approved by the Northwestern University Institutional Review Board (IRB STU00215054). Five PACS1$^{(+/+)}$ and three PACS1$^{(+/R203W)}$ induced pluripotent stem cell (iPSC) lines were utilized in total. All donors consented to tissue being used for research purposes. Two control lines derived from a 25-year-old healthy female (C1; GM03651) and 24-year-old male (C2; GM03652) were purchased as fibroblasts from Coriell and reprogrammed to pluripotency by the Northwestern University Stem Cell Core using Sendai viral vectors containing OCT4, SOX2, KLF4 and CMYC[82]. A third control line was derived from a 39-year-old unaffected mother (C3; S033751*B/GM27160) of a 3-year-old daughter with PACS1 syndrome (A2; S033745*B/GM27159). These lines were purchased as iPSCs from Coriell but are now available at Wicell (PACS1001i-GM27160 and PACS1002i-GM27159). An additional 6-year-old male patient line (A1; PACS1003i-GM27161) was purchased as iPSCs from WiCell. Three isogenic pairs were generated in total using CRISPR/Cas9 gene editing technology by introducing the heterozygous variant which leads to p.R203W into the C1 background (CRISPR R203W) and correcting the variant in both patient lines (CRISPR A1 and CRISPR

A2; Fig. S1). Karyotype of all lines was done by WiCell and determined normal (Fig. S2). All iPSC lines expressed markers of pluripotency (Fig. S3a) and demonstrated the capacity to differentiate the three germ layers (Fig. S3b–d) using the STEMdiff™ Trilineage Differentiation Kit (Stem Cell Technologies, 05230) according to manufacturer instructions. iPSCs were maintained in mTeSR+ medium (Stem Cell Technologies, 05825) on plates pre-coated with Matrigel (Corning, 354234) dissolved in DMEM/F12 at a concentration of 10 µg/mL. Stem cells were passaged as needed using ReLeSR medium (Stem Cell Technologies, 05872). All cell lines were regularly tested for mycoplasma contamination and genotyped with Sanger sequencing. Please see Table S1 for information regarding which lines were used in each experiment.

### CRISPR/Cas9 gene editing

Generation of isogenic cell lines was carried out utilizing the CRISPR/Cas9 system. Briefly, the Cas9 was delivered to cells as a px459-derived plasmid (Addgene, 62988) with a cloned guide that was designed using crispr.mit.edu software. The forward sequence was: caccgCAAGGTCTTATAGCCCAAGA and reverse: aaacTCTTGGGCTATAAGACCTTGc (lowercase letters are sticky ends to clone into the plasmid). 5 µg of plasmid and 5 µg of donor DNA (sequence TGCAGATCATGCTGCAAAGGAGAAAACGTTACAAGAATTGGACTATCTTAGGCTATAAGACCTTGGCCGTGGGACTCATCAAC) were delivered to 1 million cells using the NEON transfection system (Thermo Scientific, MPK5000). 24 h after electroporation, cells were exposed to 0.5 µg/mL puromycin for 24 h to select against cells without the plasmid. Cells were left to expand for roughly 10 days. When visible colonies had formed, they were manually isolated with a pipet, expanded, and Sanger sequenced.

To check for off-target effects and confirm the genotype, DNA from clones of interest was sent to Novogene for whole genome or whole exome sequencing. The resulting sequences were mapped using a Burrows-Wheeler Aligner (BWA, v0.7.15) and manually checked for off-target effects with the Integrative Genomics Viewer (IGV, v2.8.10). Of the top 100 off-target effects predicted by the IDT CRISPR design tool, only two other sites were exonic (MVP chr16:-29844665; PACS2 chr14:-105355130). These sites did not have any mismatched base pairs in any of the three isogenic sets of lines. The top five off-target sites predicted by CHOPCHOP (v3) were all non-coding, but analysis of the CRISPR R203W and CRISPR A1 lines - which received whole genome sequencing analysis - did not reveal any unintended edits.

### Generation of neural precursor cells (NPCs)

iPSCs were differentiated to NPCs using the STEMdiff™ SMADi Neural Induction Kit (Stem Cell Technologies, 08581) according to manufacturer instructions. In brief, 3 million iPSCs per cell line were plated into 1 well of an AggreWell™800 24-well plate (Stem Cell Technologies, 34811) with 2 mL STEMdiff™ Neural Induction Medium + SMADi + 10 µM Y-27632. The next four days, 1.5 mL STEMdiff™ Neural Induction Medium + SMADi media was replaced very carefully so as not to disturb the embryoid bodies (EBs). On day 5, EBs were dislodged from the AggreWell™ plate by forcefully dispensing DMEM/F12 and applied to a 37 µm cell strainer (Stem Cell Technologies, 27250) to get rid of dead cells. The strainer was then inverted to transfer purified EBs to a fresh Matrigel-coated 6-well plate with 2 mL STEMdiff™ Neural Induction Medium + SMADi. Media was changed every day until around day 10 when clear neural rosette structures had formed. When neural rosettes covered nearly all EBs, rosette selection was performed by incubating for 2 h with Neural Rosette Selection Reagent (Stem Cell Technologies, 05832), forcefully dislodging with warm DMEM/F12, and transferring the solution to a conical vial when rosettes had visibly detached. Rosettes were centrifuged at 350 xg for 5 min and plated on a plo/laminin-covered dish in NBF media (DMEM/F12 + 0.5x N2 + 0.5x B27 + 10 ng/mL FGF). This was designated passage 0. After expansion,

NPCs were stained with Nestin and PAX6 to validate ectodermal lineage.

## Quantification of NPC proliferation rate with BrdU

To assess proliferation rate, NPCs were plated at 200k per 18 mm plo/laminin-coated coverslip. The following day, 10 μM BrdU was added to each coverslip and cells were fixed with ice-cold methanol after 1 h or 24 h of further incubation. After fixation, coverslips were treated with 2 M HCl for 30 min, then neutralized with sodium borate for 30 min. After 3 × 10 min PBS washes, coverslips were blocked with 5% NDS, 0.1% Triton X-100 in 1x PBS for at least 1 h at room temperature. Coverslips were incubated with BrdU (BD Biosciences, 347580, Clone B44, Lot 6042756, Dilution 1:100) and Ki67 (Abcam, ab15580, Dilution 1:200) antibodies overnight, then imaged on a BZX710 Keyence epifluorescent microscope. Percent BrdU+, Ki67+, and BrdU+Ki67+ nuclei were calculated using ImageJ (Fiji v2.3.0) software. Cell cycle re-entry was determined by percentage of Ki67+ cells labeled with BrdU after 24 h. The progenitor BrdU labeling index represents the percentage of Ki67+ progenitors that also incorporated BrdU after 1 h. Cells exiting the cell cycle were determined by number of BrdU+ cells minus BrdU+Ki67+ cells over total BrdU+Ki67+ cells for both time points (see Fig. S5d).

## Quantification of NPC apoptosis rate with TUNEL

The Click-iT™ Plus TUNEL Assay (Invitrogen, C10617-C10619) was used to determine the rate of apoptosis in NPCs. In brief, NPCs were plated at 200 k per 18 mm plo/laminin-coated coverslip. The following day, cells were fixed with 4% PFA for 15 min at room temperature. Samples were incubated with 0.25% Triton X-100 in PBS for 20 min at room temperature, then washed twice with deionized water. To generate a positive control, one coverslip was incubated with 1 unit of DNase I diluted into 1X DNase I reaction buffer for 30 min at room temperature. Following a deionized water wash, 100 μL kit component A was added to each coverslip and incubated for 10 min at 37 °C. This was replaced with 50 μL prepared TdT reaction buffer and incubated for 60 min at 37 °C in a humid environment to prevent evaporation. Following the TdT reaction, the coverslips were washed 2 × 5 min with 3% BSA in PBS. Next the Click-iT™ Plus reaction was performed by adding 50 μL prepared Click-iT™ Plus TUNEL reaction cocktail to each coverslip and incubating for 30 min at 37 °C in the dark. Each coverslip was again washed with 3% BSA in PBS. DNA was stained by incubating with 2 μg/mL DAPI in PBS for 15 min at room temperature. Coverslips were then mounted with Fluoromount-G (Thermo Scientific, 00-4958-02) and imaged on BZX710 Keyence epifluorescent microscope. Two images were taken per coverslip and %TUNEL+ nuclei were calculated using ImageJ (Fiji v2.3.0) software. Statistics on biological replicate averages were calculated in R (v4.0.3) using a one-sided Kruskal-Wallis test as $p_{shapiro} = 0.0174$.

## Generation of dorsal cortical organoids

Cortical organoids were generated according to the previously described protocol by Sloan and coworkers[30,31]. iPSCs were grown on matrigel-coated plates in mTeSR medium until colonies were ~5 mm across. Colonies were detached from the plate by incubating at 37 °C with 0.35 mg/mL dispase for 20 min. Detached colonies were then transferred to an ultra-low attachment dish (Corning, 3262) with DMEM/F12 (1:1, Life Technologies, 1130-032) containing 20% vol/vol knockout serum replacement (Life Technologies, 10828-028), 1% vol/vol non-essential amino acids (Life Technologies, 1140-050), 0.1 mM 2-mercaptoethanol (Sigma-Aldrich, M3148), 5 μM of the SMAD inhibitor dorsomorphin (Sigma-Aldrich, P5499), 10 μM SMAD inhibitor SB-4321542 (R&D Systems/Tocris, 1614), and 10 μM Rock inhibitor Y-27632 (Selleckchem, S1049). Media was changed daily for the next 5 days with the same reagents as described above, minus the Y-27632. At day 6, media was changed to a Neurobasal-A base containing 1x B-27, 1x

Glutamax, 20 ng/mL FGF2, and 20 ng/mL EGF (R&D Systems, 236-EG). This media was changed every other day until day 25. At day 25, growth factors were switched to 20 ng/mL BDNF (PeproTech, 450-02) and 20 ng/mL Neurotrophin 3 (PeproTech, 450-03). Organoids were transitioned to neurobasal-A medium supplemented with 1x B-27 beginning at day 43 until the desired endpoint.

Later differentiations also utilized a kit developed based on this protocol (Stem Cell Technologies, 08620) according to manufacturer instructions. In brief, one well per line of an AggreWell™800 24-well plate (Stem Cell Technologies, 34811) was prepared by centrifuging 500 μL Anti-Adherence Rinsing Solution (Stem Cell Technologies, 07010) at 1300 × g for 5 min. 3 million cells were seeded in each prepared well by spinning for 100 × g for 3 min in 2 mL STEMdiff™ Seeding Medium. For the next 5 days, 1.5 mL per well of fresh STEMdiff™ Forebrain Organoid Formation Medium was carefully exchanged. On day 6, one Aggrewell of embryoid bodies were filtered through a 37 μM strainer (Stem Cell Technologies, 27250) and transferred to two wells of a 6-Well Ultra-Low Adherent Plate (Corning, 3471) containing 3 mL STEMdiff™ Forebrain Organoid Expansion Medium. STEMdiff™ Forebrain Organoid Expansion Medium was changed every other day until day 26 at which point the medium was changed to STEMdiff™ Forebrain Organoid Differentiation Medium. Organoids were cultured in differentiation media until day 43. Following day 43, organoids were cultured in STEMdiff™ Forebrain Organoid Maintenance Medium indefinitely. Maintenance media was changed every 2–3 days.

## Organoid cryopreservation and sectioning

Organoids were fixed with 4% paraformaldehyde in PBS at 4 °C overnight. They were then washed with PBS and transferred to a 30% sucrose solution in PBS and incubated at 4 °C for at least 16 h. To cryopreserve, organoids were embedded in OCT medium using a mold and snap frozen in dry ice and ethanol. Organoids were stored at −80 °C indefinitely. Frozen organoids were sectioned 15–20 μm thick at −16 °C in a cryostat and applied to Thermo Scientific slides. Slides were left to dry at room temperature overnight and then stored indefinitely at −80 °C.

## Immunohistochemistry

Organoid sections were dried for at least 1 h either at room temperature or 37 °C. OCT was removed in a series of 3 × 10 min washes in PBS. An antigen retrieval step was performed by steaming slices at 99 °C for 10 min in a 10 mM citrate solution at pH 6.0. To remove the citrate buffer, slides were washed 1 × 5 min in dH2O and 2 × 5 min in PBS. Tissue was blocked in a 10% NDS, 0.3% Triton X-100 solution in 1x PBS for 1–2 h at room temperature. Following blocking, tissue was incubated in a 5% NDS, 0.15% Triton X-100 solution in 1x PBS containing primary antibodies at 4 °C overnight. Primary antibodies used in this study for immunohistochemistry are: a-Tubulin (Abcam, ab7291, Clone DM1A, Lot GR3341361-10, Dilution 1:200); Brachyury (Invitrogen, 14-9770-82, Clone X1A02, Lot 2681461, Dilution 1:500); BrdU (BD Biosciences, 347580, Clone B44, Lot 6042756, Dilution 1:100); CTIP2 (Abcam, ab18465, Clone 25B6, Lot GR3420263-2, Dilution 1:200); CUX1 (Proteintech, 11733-1-AP); CXCR4 (Abcam, ab181020, Clone EPUMBR3, Lot 1002835-10, Dilution 1:200); EMX1 (Atlas Antibodies, HPA006421, Lot C106561, Dilution 1:100); FOXG1 (Abcam, ab196868, Clone EPR18987, Lot GR3242662-19, Dilution 1:100); Gephyrin (Synaptic Systems, 147-021, Clone mAb7a, Dilution 1:100); GFAP (Synaptic Systems, 173044, Dilution 1:200); GFP (Abcam, ab13970, Lot GPR236651-4, Dilution 1:200); Ki67 (Abcam, ab15580, Dilution 1:200); LHX9 (Sigma, HPA009695, Lot 000033857, Dilution 1:200); MAP2 (Invitrogen, MAB3418, Clone AP20, Dilution 1:200); Nestin (EMD Millipore, MAB5326, Lot 3430617, Dilution 1:300); NTS (Synaptic Systems, 418005, Dilution 1:500); OCT4A (Cell Signaling Technology, C30A3); OCT3/4 (R&D Systems, MAB1759, Clone 240408, Lot KQX0420031, Dilution 1:100); PACS1 (Generated by Gary Thomas's Lab; Dilution

1:200); PAX6 (BioLegend, 901301, Lot B386304, Dilution 1:100); PSD95 (NeuroMab, 75-028, Clone K28/43, Dilution 1:1000. Gift from the lab of Peter Penzes); SATB2 (Synaptic Systems, 327004, Dilution 1:100); SOX2 (EMD Millipore, AB5603, Lot 3587118, Dilution 1:200); SOX17 (Invitrogen, MA5-24885, Clone OTI3B10, Lot YE3927121, Dilution 1:100); SYN1 (Cell Signaling Technology, 5297, Clone D12G5, Dilution 1:300. Gift from the lab of Peter Penzes); TBR1 (Proteintech, 66564-1-Ig, Dilution 1:200); TRA1-60 (R&D systems, MAB1658, Clone 222328, Lot JKW0219121, Dilution 1:100); TRA1-60 (Sigma, MAB4360); TRPC5 (Invitrogen, MA5-27657, Clone N67/15, Lot YE3931332, Dilution 1:500); and vGAT (Thermo Scientific, PA5-27569, Lot WL3447439, Dilution 1:200). The overnight incubation was followed by $3 \times 10$ min PBS washes, then secondary antibody incubation at 1:200 in a 5% NDS, 0.15% Triton X-100 solution in 1x PBS for 90 min at room temperature. Secondary antibodies used in this study include: donkey anti-mouse Alexa Fluor 488 (Invitrogen, A21202, Lot 1890861); donkey anti-mouse Alexa Fluor 594 (Invitrogen, A21203, Lot 2294985); donkey anti-mouse Alexa Fluor 647 (Invitrogen, A31571, Lot 2720365); donkey anti-rabbit Alexa Fluor 488 (Invitrogen, A21206, Lot 1874771); donkey anti-rabbit Alexa Fluor 594 (Invitrogen, A21207, Lot 2441375); donkey anti-rabbit Alexa Fluor 647 (Invitrogen, A31573, Lot 2359136); donkey anti-chicken Alexa Fluor 488 (Jackson ImmunoResearch, 703-545-155, Lot 119854); donkey anti-chicken Alexa Fluor 594 (Jackson ImmunoResearch, 703-585-155, Lot 147524); donkey anti-rat Alexa Fluor 488 (Invitrogen, A21208, Lot 2273677); donkey anti-rat Alexa Fluor 594 (Invitrogen, A21209, Lot 2078918); and goat anti-guinea pig Alexa Fluor 488 (Invitrogen, A11073, Lot 46214 A). Excess secondaries were removed with $3 \times 10$ min PBS washes and a 20-min incubation in 2 μg/mL DAPI. Slides were mounted with Fluoromount-G (Thermo Scientific, 00-4958-02) and imaged on a Nikon W1 spinning disk confocal microscope or BZX710 Keyence epifluorescent microscope.

For quantifications in Fig. 1d, e, 1 field of view per section and up to 5 sections per organoid were imaged blinded. Sections with necrotic-looking tissue were not included in quantifications. Organoids from multiple lines within each genotype were used; see Table S1 for further information. Blinded images were analyzed using ImageJ (Fiji v2.3.0) and CellProfiler (v4.2.4). To make visualizations in Fig. 1d, e comparable, the total number of positive cells was determined per image field and the fraction each marker contributed to the total number of positive cells was calculated, then plotted as averages per genotype and day in Fig. 1d, e and per organoid in Fig. S4d, e.

## Organoid protein extraction and Western blotting

Proteins were extracted from day 40 neural organoids using RIPA buffer (Cell Signaling Technology, 9806) supplemented with protease and phosphatase inhibitor cocktail (Thermo Scientific, 78440). Protein concentration was determined using the Pierce BCA protein assay kit (Thermo Scientific, 23227), and lysates were denatured by adding Laemmli buffer (Bio-Rad, 1610747) and boiling at 95 °C. 30 μg of protein extract was resolved on 4–15% polyacrylamide gels (Bio-Rad, 4561085), transferred to PVDF membranes (Thermo Scientific, 88518), and then immunoblotted with specific antibodies anti-PACS1 (Sigma Aldrich, HPA038914, Lot A117139, Dilution 1:500) and α-Tubulin (Abcam, ab7291, Clone DM1A, Lot GR3341361-10, Dilution 1:1000). Membranes were then incubated with goat anti-rabbit IRDye 800CW (Li-cor, 926-32211, Dilution 1:10,000) and goat anti-mouse IRDye 680RD (Li-cor, 926-68070, Dilution 1:10,000) near-infrared dye-conjugated secondary antibodies. Protein signal was detected using the Odyssey® Fc Imaging System (Li-Cor) and quantified with Image Studio (Li-Cor). All samples that underwent Western blot analysis were included in quantifications. Uncropped scans are available in the Source Data file.

## Dissociation of cortical organoids for single-cell RNA sequencing

Dissociation was done using the Worthington Papain Dissociation System (LK003150). Steps were performed with sterile materials in a tissue culture hood. At day 40–42, two organoids between 3 and 4 mm were selected per genotype. Just one day 88 organoid was used per sample due to limited availability of tissue. Organoids were pooled and washed with PBS + 0.2% BSA, then chopped into ~0.5 mm pieces using a #10 blade. Organoids were incubated in a sterile-filtered 5 mL solution containing 20 U/mL papain in 1x EBSS, 0.46% sucrose, 26 mM NaHCO$_3$ (included in kit), 0.5 mM EDTA (included in kit), and DNase (included in kit) equilibrated with 95%O$_2$:5% CO$_2$. Organoid pieces were incubated for 75 min on a 95 RPM orbital shaker in a 37 °C incubator and triturated using wide bore tips (Rainin, 30389218) every 30 min to help break up pieces. Following incubation, organoids were triturated one more time and transferred to a 15 mL conical vial. Cells were spun at $300 \times g$ for 5 min. The pellet was resuspended in a 3 mL papain inhibitor solution containing 2 mg/mL ovomucoid inhibitor, 2 mg/mL albumin, 0.46% sucrose, and DNAse in 1x EBSS (all from Worthington, kit LK003150). This cell suspension was gently applied on top of a 5 mL solution of 10 mg/mL ovomucoid inhibitor, 10 mg/mL albumin, and 0.46% sucrose to create a density gradient. The gradient was spun at $70 \times g$ for 6 min. The pellet was washed with 5 mL PBS + 0.2% BSA and spun again at 300 x g for 5 min. The pellet was resuspended in 130 μL PBS + 0.2% BSA and filtered to remove cell clumps (Bel-Art, H13680-0040), resulting in the final single-cell suspension. In total, nine samples were processed from four distinct differentiations (three at day 40 and one at day 88).

## Single-cell RNA sequencing and downstream analysis

Preparation of samples was carried out as described above. Samples were only processed if viability after the dissociation was >85%. The library prep was conducted by Northwestern's Center for Genetic Medicine using the 10x Chromium system. Sequencing was done using 100 bp, paired-end reads with a targeted 300–400 million reads per sample. Day 40 samples were sequenced on an Illumina HiSeq 4000 and day 88 samples were sequenced on an Illumina NovaSeq 6000 due to changes in Northwestern's scRNAseq pipeline. This is reflected in the average number of reads and genes captured per cell between timepoints (Fig. S8d), so DE analysis was performed separately. Raw data was demultiplexed and reads were aligned to GRCh38 at Northwestern's Center for Genetic Medicine using the CellRanger pipeline (v3.0.2 for C1, C3, CRISPR-R203W, and A1 day 40 samples; v4.0.0 for the A2 day 40 sample; and v7.0.1 for all day 88 samples due to the number years spanning this project).

After sequencing, matrix files were processed in parallel using Seurat (v4.0.0) and Monocle3 (v1.0.0) packages in R (v4.0.3-v4.1.1)[83–85]. In Seurat, data was filtered by number of features, counts, and percent mitochondrial reads (nFeature_RNA > 500 & nFeature_RNA < 6500 & percent.mt <15 for day 40 and nFeature_RNA >1000 & nFeature_RNA < 8000 & nCount_RNA > 3000 & nCount_RNA < 40000 & percent.mt <7.5 for day 88 samples). One day 40 A2 sample was eventually omitted altogether due to low number of cells captured and overall poor quality control metrics, but all other samples just had low-quality cells filtered. For each sample individually, doublets were first identified by performing standard processing steps and then running two separate doublet finding packages: DoubletFinder (v2.0.3) and DoubletDecon (v1.1.6)[86,87]. Barcodes were extracted for consensus cells that both packages identified as doublets (<5% of each sample).

Following doublet identification (a schematic is shown in Fig. S6a), analysis was re-started and filtered with the above parameters including removal of doublet barcodes. Count matrices of each sample were normalized to 10,000 counts using the NormalizeData function with default parameters. Top 3000 variable features were

identified using the FindVariableFeatures function. To integrate samples from different batches, anchor sets were identified between each organoid pair using canonical correlation analysis and then optimized through iterative pairwise integration using FindIntegrationAnchors (dims = 1:20) and IntegrateData functions (default parameters). On this integrated object, the functions ScaleData (regressing out the effect of percent.mt), RunPCA (default parameters), FindNeighbors (reduction = pca), FindClusters (resolution = 0.8), and RunUMAP (reduction = pca, n.neighbors = 30, min.dist = 0.2, dims = 1:20) were run to generate a processed Seurat object containing all samples for comparable visualization and analysis. Minor differences in technical metrics - like average number of genes per cell - were noted between day 40 and day 88 organoids due to advances at Northwestern's sequencing core between sample submissions (Fig. S8d). We performed DE analysis separately between organoids from different time points to avoid potential confounding factors caused by these technical differences, but for analyses where data was processed together such as Fig. 2d, the count matrix was subsetted to only include genes expressed in all organoids and re-normalized to 10,000 counts. Cluster identification was determined both by consensus of canonical marker gene expression and by bioinformatic comparison of each average cluster transcriptome to a reference dataset using the Clustifyr package (v1.2.0)[88]. The reference dataset was generated from human fetal tissue,[89] so the resulting correlations to specific neuronal subtypes were simplified to represent the broad cell type classification.

Following cell type identification, DE analysis was run between genotypes for each cell type and time point. A standard single-cell DE workflow was first performed using the fit_models and coefficient_table functions in the Monocle3 package where each cell is treated as an $n$[85]. The default Wald test and Benjamini-Hochberg corrections were used. To increase confidence of results, we validated DEGs using a pseudobulk approach with DESeq2 (v1.30.1)[90]. First, the counts for cells of the same subtype within each sample were aggregated into a summary count matrix. Features with <5 counts for any aggregated sample were discarded. DESeq2 was run using the default Wald test with Benjamini-Hochberg correction followed with an lfcShrink transformation (type = apeglm)[91]. These results were then joined by cell type, day, and direction of change. Hits with <0.5 absolute fold change or >0.05 $p_{adj}$ for either DEG identification method were discarded. Genes with a base mean of <60 were also discarded. Mitochondrial (MT-) and ribosomal (RPS/RPL) proteins were removed (only 5). 277 high-confidence DEGs across all cell types and both time points remained in the final list after using these stringent filters.

Following DEG identification, gene ontology enrichment analysis was performed using the topGO package (v2.42.0)[92]. Given the stringency of our DEG identification parameters, enrichment analysis was performed on hits grouped by cell class-specific categories including cycling radial glia, radial glia, intermediate progenitors, newborn neurons, immature neurons, mature neurons, or other. The entire set of features present in the combined count matrix served as the background gene list, and results were considered only if more than two genes contributed to the outcome. Significance assessment utilized topGO's Fisher's Exact test with the elim algorithm, which inherently considers hierarchical relationships within the gene ontology structure. The developers therefore consider the resulting $p$-values corrected due to the elimination of redundant testing within the algorithm[92].

To more wholistically visualize GO results, the ontology hierarchy was extracted and reduced using the rrvgo package (v1.9.1)[93] with the score of the term being $-\log_{10}(p_{elim})$. Results were then visualized using the PieDonut function from the webr package (v0.1.5)[94]. Additional downstream analysis was done using the Qiagen Ingenuity Pathway Analysis tool (v1.21.03). The filtered DEG list was uploaded using the mean $\log_2FC$ and $p_{adj}$ between Monocle3 and DESeq2 analyses. 55/65

DEGs mapped to the IPA software in the up + downregulated genes dataset and 36/43 mapped in the upregulated dataset. Neurotransmission ($p_{adj}$ = 0.0035) and synaptic transmission ($p_{adj}$ = 0.0039) were the top two enriched functions for the up + downregulated DEGs while dendritic growth/branching was 21st ($p_{adj}$ = 0.0193), but with many similar terms such as "shape change of neurites" ($p_{adj}$ = 0.0182) before this point. In the upregulated DEG dataset, G beta gamma signaling and GABA receptor signaling were equally the strongest enriched pathways at $p_{adj}$ = 0.0155. IPA statistics were calculated using a right-tailed Fisher's Exact test with a Benjamini-Hochberg correction for multiple testing.

### Synaptic quantifications in organoids
Day 79 organoid sections were stained for MAP2 and either SYN1/PSD95 or vGAT/GPHN primary antibodies as described in the immunohistochemistry methods. Multiple cell lines from each genotype were used; see Table S1 for further details. Confocal images were taken blinded on a Nikon W1 Dual CAM Spinning Disk microscope at 60x resolution with a 15 μm Z-stack using a 1 μm step size, then collapsed to one plane in ImageJ based on maximal intensity. Pre- and post-synaptic puncta channels were converted to binary and superimposed, then the number of colocalizations above 10 pixels in area was automatically counted using the Analyze Particles macro. MAP2 was used to confirm neuronal identity of the image field but was not factored into the colocalization scheme. Statistical analysis was then done in GraphPad Prism (v8.4.2) using a two-sided Mann–Whitney test. All quantifications were included in the final analysis, except for one SYN1/PSD95 value from the A1 line that was 2.2 standard deviations from the next highest value and 4.2 standard deviations from the mean.

### Calcium imaging in organoids
Calcium imaging was done across days 40–42 and 79–82 of organoid differentiation. To section, organoids were mounted in 3.5% low-melt agarose (Promega, V2111) dissolved in oxygenated aCSF (125 mM NaCl, 2.5 mM KCl, 26.2 mM NaHCO$_3$, 1 mM NaH$_2$PO$_4$, 11 mM glucose, 5 mM HEPES, 2 mM CaCl$_2$, 1.25 mM MgCl$_2$). Organoids were then sectioned as thin as possible - between 150 μm and 250 μm - in ice-cold oxygenated aCSF using a vibratome (Leica, VT1200S) set to 0.2 mm/s speed and 1.00 mm amplitude. To fix slices in place for imaging, sections were plated on 20 μg/mL matrigel-coated 8-chamber dishes (Ibidi, 80806) in maintenance media and left to recover overnight at 37 °C, 5% CO$_2$. 1 h prior to imaging, sections were loaded with Fluo-4 Direct (Thermo Scientific, F10473) prepared according to kit instructions in oxygenated aCSF. Before imaging, media was changed to aCSF only. Sections were imaged in a live-cell chamber at 37 °C 5% CO$_2$ with a Nikon W1 spinning disk confocal microscope for 5 min at 2 Hz. An equal volume of 100 μM bicuculline (BCU; Tocris) in aCSF (for 50 μM total) was then added to the section and incubated for 15 min at room temperature. Following incubation, sections were imaged for 3 min at 2 Hz. For sections exposed to TTX (Tocris Bioscience, CAS 18660-81-6), 1 μM total was applied to the section and imaged after 5 min incubation for 1 min at 2 Hz.

Calcium signal was extracted in ImageJ (Fiji v2.3.0) by calculating average intensity for each frame over regions of interest extracted based on z score. Traces were processed in Matlab (vR2022a) using code developed by Marc Dos Santos[95–97]. In brief, data were normalized to a rolling median background of 25 s. Peaks were detected using the findpeaks function (MinPeakDistance = 3 s, MinPeakHeight = Δ0.17 F/F$_0$, MinPeakProminence Δ0.10 F/F$_0$, MaxPeakWidth = 9 s). Cells with no detected peaks were removed from the analysis, except for paired TTX recordings. Output metrics were analyzed in R. A Shapiro test for normality was performed followed by either an ANOVA test with Tukey HSD correction if $p > 0.05$ or a Kruskal-Wallis test of individual variables with a Benjamini Hochberg correction if $p < 0.05$.

## NEUROG2-mediated differentiation of stem cells to glutamatergic neurons

iPSCs were differentiated using an adapted protocol based on NEUROG2 overexpression[40]. Stem cells were dissociated using Accutase (EDM Millipore, SCR005), then resuspended in mTeSR+ with 10 μM ROCK inhibitor. 600 k cells in 1 mL media were transferred to an eppendorf tube and incubated for 15 min at room temperature with 2.5 μL rtTA virus (Iowa Viral Vector Core, FIV3.2CAG-rtTA VSVG), 1.25 μL Tet-inducible NEUROG2::T2A::puroR (produced at the Northwestern University Gene Editing, Transduction, & Nanotechnology Core), and 8 μL of 8 mM polybrene. This solution was then transferred to a matrigel-coated 10 cm plate with 7 more mL mTeSR+ with 10 μM ROCKi. Infected cells were maintained in mTeSR+ and expanded to confluency. When confluent, half the media was changed to Neurobasal-A with 0.5x B27, 1x N2, 1x NEAA, and 1x Glutamax (called NBM media). 3 μg/mL doxycycline was added to the NBM:mTeSR media to induce NEUROG2 expression. On days 2–5, media was changed daily with the following components: Neurobasal-A with 0.5x B27, 1x N2, 1x NEAA, 1x Glutamax, 2 μg/mL dox, 10 μM SB431542, 100 nM LDN-193189, 2 μM XAV939, 2 μg/mL puromycin, and 10 μM ROCKi (day 2 only). On day 6, cells were dissociated with Accutase (EDM Millipore, SCR005) and frozen at 1 million per vial in freezing media (45% day 2–5 media without puro; 45% FBS; 10% DMSO). When thawed, neurons were maintained in Neurobasal-A with 1x B27, 1x N2, 1x NEAA, 1x Glutamax, 2 μg/mL doxycycline, 2% FBS, and 10 ng/mL BDNF with half media changes on a M/W/F schedule.

## Sholl analysis of NEUROG2-derived neurons

To perform Sholl analysis, NEUROG2-derived neurons generated using the protocol described above were plated at day 7 from frozen onto a glia-covered (Gibco, N7745100), Matrigel-coated 18 mm coverslip at 300 k/well. Neurons were maintained in Neurobasal-A with 1x B27, 1x N2, 1x NEAA, 1x Glutamax, 2 μg/mL doxycycline, 2% FBS, and 10 ng/mL BDNF. Half the media was changed on a M/W/F schedule. At day 43, neurons were transfected with a Clontech pEGFP-N2 plasmid (Gift from the lab of Peter Penzes). Prior to incubation, half the old media was removed and saved. In accordance with the Lipofectamine 3000 (Invitrogen, L3000-015) protocol, 125 μL per well of Opti-MEM was mixed with 2.5 μg/well of pEGFP-N2 plasmid and 4 μL/well of P3000 reagent. In a separate tube, 125 μL/well of Opti-MEM was mixed with 2 μL/well of Lipofectamine 3000. After thoroughly mixing both solutions and letting them sit at room temperature for 5 min, DNA and lipofectamine mixtures were added in a 1:1 ratio. The mixture was allowed to incubate at 37 °C for 20 min to facilitate formation of DNA-lipid complexes. Following incubation, 250 μL of the mixture was added to each well and put back in the incubator for 5 hours. Neurons were then gently washed 2x with warm Neurobasal-A and ½ old media (saved in the initial steps) + ½ fresh media was re-applied to the cells.

After culturing for 3 days to facilitate GFP expression, neurons were fixed for 15 min at room temperature with 4% PFA. Neurons were washed 3x with PBS for 10 min each, then blocked for 1 h in 10% NDS + 0.25% TX100 + PBS. Neurons were stained with a GFP antibody (Abcam, ab13970, Lot GPR236651-4, Dilution 1:200), as well as GFAP (Synaptic Systems, 173044, Dilution 1:200) and MAP2 (Invitrogen, MAB3418, Clone AP20, Dilution 1:200) in order to distinguish neurons and glia. Coverslips were imaged with a BZX710 Keyence microscope. Debris or other cells that would interfere with Sholl analysis were manually removed with ImageJ (Fiji v2.3.0). Images of isolated neurons were then converted to binary files and the Sholl plugin on Fiji was used (v4.0.1). Concentric circles 10 μm apart were drawn, up to 800 microns as needed. The number of intersections for each concentric circle was exported to R (v4.0.3-v4.1.1) where further analysis was performed. Length of longest dendrite was defined as furthest

concentric circle with more than one intersection to avoid a skewed value due to a long axon. *p* values were determined with two-sided *t* tests when Shapiro and Goldfeld-Quandt tests were not significant.

## Multielectrode array plating and recording of NEUROG2-derived neurons

48-well MEA plates (Axion biosystems, M768-tMEA-48W) were coated with 2x Matrigel by placing a 10 μL droplet of 20 μg/mL Matrigel (Corning, 354234) dissolved in DMEM/F12 directly on top of the electrode grid and incubating for 16 h at 37 °C. The following day, Day 7 neurons were thawed and re-counted. Enough for 50 k/well neurons per well were transferred to a conical vial. P2 rat primary cortical astrocytes (Thermo Scientific, N7745100) at 25 k/well were added to the same vial. After centrifugation, cells were resuspended at 50 μL per well in MEA media (Neurobasal-A with 1x B27, 1x N2, 1x NEAA, 1x Glutamax, 10 ng/mL BDNF, 2 μg/mL doxycycline, and 2% FBS). A 50 μL drop containing 50 k neurons and 25 k glia was placed in the center of each electrode grid in the 48 well plate and left in the incubator for 1 h to facilitate attachment of neurons directly over the electrodes. After 1 h, 250 μL more media was gently added to each well. Half the media was changed on a M/W/F schedule. At day 10, 1 μM AraC was added for 2 days to inhibit proliferative cells.

Starting at day 10, spontaneous and stimulated activity was recorded prior to media changing. Signal recording and processing parameters were established by Simkin and coworkers[50] and applied here with permission. Spontaneous activity was recorded using Axion Biosystems Maestro 768 channel amplifier and Axion Integrated Studios (AxIS) v2.5 software. The amplifier recorded from all channels simultaneously using a gain of 1200x and a sampling rate of 12.5 kHz/channel. After passing the signal through a Butterworth band-pass filter (300–5000 Hz), on-line spike detection (threshold = 6 x the root-mean-square of noise on each channel) was performed with the AxIS adaptive spike detector. All recordings were conducted at 37 °C in 5% $CO_2$/95% $O_2$. Spontaneous network activity was recorded for 5 min each day starting on day 10 of differentiation. Starting day 10, neurons were also electrically stimulated with 20 pulses at 0.5 and 0.25 Hz after spontaneous recordings were made to facilitate maturation and migration of neurons to the electrode field. After each recording session, cells were observed under a microscope to check for excessive clumping or other indicators of suboptimal recording conditions. Data was omitted if cells did not look healthy and distributed enough across the electrode field to produce high-quality recordings.

Spike files were processed with the Axis Neural Metric Tool (v3.1.7). Electrode burst settings were algorithm = Poisson surprise, min surprise = 5. Network burst settings were algorithm = ISI threshold, min # spikes = 50, max ISI (ms) = 100, min % of electrodes = 18, synchrony window = 20. Average network burst settings were network burst window start = 0 ms, network burst window end = 100 ms, bin size = 1 ms. Metrics generated from these output parameters were further investigated in R (v4.0.3-v4.1.1). *p* values of metrics over time between genotypes were calculated by conducting an ANOVA between a linear regression model with and without genotype included as a covariate. This trend-based approach was taken as opposed to examining each of 12 recording time points independently to analyze the results more wholistically. Data did not always meet all the assumptions of linear regression, so when appropriate, individual significances at specific time points were reported with *p* values obtained using a one-sided Kruskal-Wallis test and are reported in the legend.

To make sure there were no differences between genotypes in number of surviving neurons to confound results, 2 μL of AAV2-hSynapsin1::mCherry virus (gift from the lab of Tracy Gertler) was added to the plate at day 30. Expression was imaged at day 57 using a BZX710 Keyence epifluorescent microscope following a 15 min room-temperature fixation with 4% PFA. Number of mCherry⁺ cells within the electrode field were then counted blinded. Technical replicates

across experiments were averaged. A two-tailed unpaired $t$ test using GraphPad Prism (v8.4.2) did not show differences in number of surviving neurons between PACS1$^{(+/+)}$ and PACS1$^{(+/R203W)}$ lines ($t_{13} = 0.243$, $p = 0.812$).

## Statistics

Statistical analyses for each experiment were performed in R (v4.0.3-v4.1.1) or GraphPad Prism (v8.4.2-v9.4.1). For FOXG1$^+$/TBR1$^+$/SATB2$^+$ counts in Fig. 1d, the ratio of each marker to the total number of positive cells (averaged per organoid) was calculated in R using a two-way ANOVA with Tukey HSD correction if $p > 0.05$ in the Shapiro test for normality, which was true of FOXG1 and TBR1. This was not the case for SATB2 ($p_{Shapiro} = 0.001$) and so a one-sided Kruskal-Wallis test was done individually for genotype and day, then adjusted using a Benjamini Hochberg correction. The corresponding $F$ statistic for ANOVA results or $H$ statistic for Kruskal-Wallis results and degrees of freedom are reported in the figure legend. In Fig. 1e, the ratio of SOX2$^+$/CTIP2$^+$ cells was calculated in R and averaged per organoid. $p_{Shapiro} = 0.003$ so a one-sided Kruskal-Wallis test was done individually for genotype and day, then adjusted using a Benjamini Hochberg correction. The corresponding $H$ statistic and degrees of freedom are reported in the figure legend. In Fig. 1g, $p$ value for PACS1 protein levels was determined using a two-sided Mann-Whitney test. The corresponding $U$ statistic and $p$ value are reported in the figure legend.

To determine final $p$ values for differentially expressed genes identified with scRNAseq, a standard single-cell DE workflow was first performed using the fit_models and coefficient_table functions in the Monocle3 package where each cell is treated as an $n$[85]. The default Wald test and Benjamini-Hochberg corrections were used. To increase confidence of results, we validated DEGs using a pseudobulk approach with DESeq2 (v1.30.1)[90] using the default Wald test with Benjamini-Hochberg correction followed with an lfcShrink transformation (type = apeglm[91]). Hits with <0.5 absolute fold change or > 0.05 $p_{adj}$ for either DEG identification method were discarded, and the averaged $q$ values between the two methods were considered the final $p_{adj}$. These values are reported in Data S1. Following DEG identification, gene ontology enrichment analysis was performed using the topGO package (v2.42.0)[92]. Significance assessment utilized topGO's Fisher's Exact test with the elim algorithm, which inherently considers hierarchical relationships within the gene ontology structure. The developers consider the resulting $p$ values corrected due to the elimination of redundant testing within the algorithm[92]. Enriched functions and processes were also assessed using the Qiagen Ingenuity Pathway Analysis tool (v1.21.03), which utilized a right-tailed Fisher's Exact test with a Benjamini-Hochberg correction for multiple testing.

Synapse quantifications in Fig. 4b, c were done in Prism using a two-sided Mann-Whitney test. The corresponding $U$ statistic and $p$ value are reported in the figure legend. For calcium imaging analysis, a Shapiro test for normality was performed followed by either a two-way ANOVA test with Tukey HSD correction if $p_{Shapiro} > 0.05$ or one-sided Kruskal-Wallis of individual metrics with Benjamini Hochberg correction if $p_{Shapiro} < 0.05$. Corresponding $F$ statistics and degrees of freedom for ANOVA results or the $H$ statistic and degrees of freedom for Kruskal-Wallis results are reported in the figure legend. For Sholl analysis, Shapiro and Goldfeld-Quandt tests were first performed to check for a non-normal distribution and heteroscedasticity. $p > 0.05$ in both tests so a two-sided student's $t$ test was run. The corresponding $t$ statistic, degrees of freedom, and $p$ value are reported in the figure legend. In Fig. 5h, a two-sided unpaired parametric $t$ test was run in GraphPad Prism (v9.4.1) to assess the relationship between genotypes of the number of surviving synapsin$^+$ cells on MEA electrode fields. The $t$ statistic, degrees of freedom, and $p$ value are reported in the figure legend. For MEA metrics in Fig. 5i-o and Fig. S15b-e, $p$ values were calculated by conducting an ANOVA between a linear regression model

with and without genotype included as a covariate. This trend-based approach was taken as opposed to examining each of 12 recording time points independently to analyze the results more wholistically. The $F$ statistic, degrees of freedom, and $p$ value are listed in the legends of Fig. 5i-o and Fig. S15b-e. Data did not always meet all the assumptions of linear regression so when appropriate, individual significances at specific time points with $p$ values obtained using a one-sided Kruskal-Wallis test and are reported in the legend.

## Reporting summary

Further information on research design is available in the Nature Portfolio Reporting Summary linked to this article.

## Data availability

The FASTQ and CellRanger matrix files for the single-cell RNA sequencing (scRNAseq) data generated in this study have been deposited in the NCBI Gene Expression Omnibus (GEO) database under accession code GSE250386. scRNAseq reads were aligned to the publicly available human genome reference build GRCh38. The counts for organoid immunohistochemistry analysis (Fig. 1b-e), quantifications for PACS1 abundance (Fig. 1f, g), differentially expressed genes identified with scRNAseq (Fig. 3), Sholl analysis results (Fig. 5e-g), and key metrics for calcium imaging (Fig. 4) and multielectrode array recordings (Fig. 5) generated in this study are provided in the Source data file.

## Code availability

All code is available upon request. Code used in this study is a compilation from the following sources. Reads were aligned by Northwestern University's Center for Genetic Medicine using the 10X Genomics Cell Ranger pipeline found at https://support.10xgenomics.com/single-cell-gene-expression/software/pipelines/latest/using/tutorial_ov. The output matrix files were processed with Seurat and Monocle3 by following vignettes developed by the Satija lab (https://github.com/satijalab/seurat) and Trapnell lab (https://github.com/cole-trapnell-lab/monocle3) respectively. Differentially expressed genes were determined by consensus results from Monocle3 approaches and a DESeq2 workflow specifically developed for pseudobulk scRNAseq analysis (https://hbctraining.github.io/scRNA-seq/lessons/pseudobulk_DESeq2_scrnaseq.html). Code for calcium imaging analysis was developed by Marc Dos Santos[95–97] and adapted with permission by Lauren Rylaarsdam. Examples of this pipeline are available at https://github.com/marcdossantosPHD/PenzesLabimaging. R code for analyzing multielectrode array metrics calculated with the Axis Neural Metric Tool is available upon request.

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

## Acknowledgements

This work was funded by the PACS1 Syndrome Research Foundation (A.G.G.) and National Institutes of Health grant R01NS123163 (A.G.G.). Confocal microscopy was done at the Northwestern University Center for Advanced Microscopy funded by NCI CCSG P30 CA060553. Single-cell RNA sequencing was done at the Northwestern Center for Genetic Medicine funded by NIH grant 1S10OD025120. We thank Matthew Schipma, Xinkun Wang, Kara Pivarski, Laura Shihadah, and Jennifer Ching Man Wai at the Northwestern Center for Genetic Medicine for their assistance with single-cell RNA library preparation, sequencing, and running the CellRanger pipeline. The authors thank Gemma Carvill for her continued feedback on the project and the lab of Peter Penzes for their generous help with synapse immunohistochemistry experiments, calcium imaging, Sholl analysis, and MEA experiments. Specifically, authors would like to thank Euan Parnell for demonstration of calcium imaging protocols; Marc Dos Santos for generously sharing calcium imaging code, providing instruction for its use, and sharing antibodies and tips for synapse experiments; Nicolas Piguel for providing transfection vectors and sharing a protocol for Sholl analysis; and Lorenza Culotta for advice regarding MEA experimental protocols. The authors would additionally like to thank the lab of Evangelos Kiskinis, especially Dina Simkin, for generously sharing their Axion Biosystems Maestro recording system and teaching L.R. how to use it. L.R. would finally like to thank Ximena Gómez-Maqueo for her input on the DESeq2 analysis design and MEA statistics and Martín Fairbanks Santana for assistance in stem cell culturing for the karyotype and trilineage assays.

## Author contributions

This work was conceptualized by A.G.G. and L.R. Experiments, data analysis, and figures were done by L.R. unless otherwise specified. J.R. performed the western blot and quantification analysis in Fig. 1f, g, immunohistochemistry in Fig. S14b, c, imaging for Fig. S10, S14b, c, and provided regular guidance. EGP performed many of the immunohistochemistry stains in Fig. 1b, c, Fig. S4b, c, S10, and did all the quantifications in Fig. 1d, e (shown by line in Fig. S4d, e). L.R., J.R., and E.P. all contributed to the iPSC characterizations shown in Fig. S3. Writing was done by L.R. and edited by J.R. and A.G.G.

## Competing interests

The authors declare no competing interests.
