## [Peer Review File · Nature Communications]

iPSC-derived models of PACS1 syndrome reveal transcriptional and functional deficits in neuron activityReviewers' comments:

Reviewer #1 (Remarks to the Author):

Rylaarsdam et al. study a PACS1 syndrome, a neurodevelopmental disorder that causes intellectual disability and craniofacial abnormalities. They describe the effect of a PACS1 mutation (R203W) on the development and gene expression of human forebrain organoids. By using patient derived lines and additional edited cell lines, they are able to show that organoids carrying the mutation have an increased number of GABAergic cells, increased expression levels of associated genes, and a dysregulation in the number of inhibitory synapses. Furthermore, they show an overlap between the upregulated genes in the PACS1 mutant organoids and overexpressed genesets in ASD models, suggesting some phenotypic convergence.

Literature on PACS1 syndrome is very limited, and no in vitro human models have been used to date to understand PACS1 function during development or the consequences of the R203W mutation. The present study is original in its approach and findings and will be highly significant for understanding the mechanisms underlying PACS1 syndrome during brain development. Generally, the paper is well written and findings are well supported. Data is well presented and analyzed. However, there are some important issues surrounding the single cell RNA-seq analysis that need to be addressed to properly support the current interpretation and conclusions. Besides these meaningful issues, the methodology is sound and meets the expected standards. In most cases enough detail is provided in the methods section for the work to be reproduced. The cases where it is lacking have been highlighted.

This paper would be an important contribution to the field of neurodevelopmental disorders. However, the paper in its current form has concerns to be addressed before acceptance.

Major comments

1) While the authors specify that they are using forebrain organoids, the regional identity of those organoids has not been properly shown. This becomes particularly important as some of the DEG correspond with regional differences in mouse and human brain, such as ZIC1, ZIC2, LHX9, FOXG1; and considering some other surprising observations by the authors that are not frequent in forebrain organoid literature, such as PAX3 expression in intermediate progenitors and the abundant VLMC. The regional identity of these forebrain organoids could be shown, for example, with scRNA-seq and immunostainings showing extensive FOXG1 expression - as done in other papers in the field like Bhaduri et al. 2020, or EMX1 expression to show cortical fate as done in other works. Proper regional and cell type identities are of the utmost importance to correctly interpret changes in gene expression upon PACS1 mutation. It is also extremely relevant when choosing other datasets to compare for DEG calculation, so that the authors are sure to include datasets (or subsets of them) coming from organoids with similar identities (which should be shown with expression of the same genes).

2) In a similar line, the authors propose an upregulation of GABAergic genes, as well as other genes involved in regional patterning including ZIC1, ZIC2, LHX9 and FOXG1. However, these genes are not consistently overexpressed (compare for example ZIC1 and GAD1 feature plots) suggesting that there may be more than one phenotypic change. ZIC genes overexpression is suggestive of a change in regional identity, that should at least be considered. In light of these observations, it becomes important to address: i-is there a change in expression profiles or a change in proportions of neuronal subtypes? ii-what is the molecular identity of the GABAergic neurons? iii-is there a change in patterning or regional information in the mutant organoids? With respect to the first question, the excitatory neuron cluster seems indeed to include all kind of neurons. Markers used are pan neuronal (MAPT2, SYT1 and STMN2) and not suitable to distinguish between excitatory and inhibitory neurons. BCL11B is a neuronal subtype marker gene that is expressed by both excitatory and inhibitory neurons. Considering the restricted and high expression of inhibitory genes DLX and GAD1 (Fig. 3c) it is surprising that no cluster was annotated as inhibitory neurons to begin with in Fig. 1. The correct annotation is critical to fully understand if there is a change in cell proportions of different neuronal classes between control and patient derived organoids, and to be able to properly choose datasets for DEG comparison later in the paper. With respect to that last point, for calculation of DEG the authors chose to include published datasets (a smart idea to compensate for lower number of replicates and add power to the analysis). However, while it is stated that they were "generated with the same protocol and collected at the same time point next to data collected in this study" that is not fully

the case. Having neurons properly classified (at least between excitatory and inhibitory) then becomes critical to be able to set suitable DEG comparisons. For instance, control organoids from this paper contain at least one clear cluster of neurons that express inhibitory genes (ARX, DLX1, DLX2, DLX-AS1, GAD1, Fig. 3c), while very few cells are annotated as such in Bhaduri et al (see Fig. 1 in their paper). Importantly also, the authors are comparing to Bhaduri's most directed organoids, in which one would expect even less inhibitory cells as differentiation has been directed to a cortical fate. Differences in cell types proportions are also expected as a consequence of using different cell lines. Taking all these elements into consideration, it is recommended that comparisons between the three control groups are also carried out to confirm that the genes arising are not DEG among organoids protocols to begin with.

3) The authors use cell lines derived from 2 patients and 3 controls, plus one additional PACS1 R203W edited line. The use of multiple lines is important as it gives support to the relevance and penetrance of the phenotype. However, it is not emphasized throughout the paper in how many cell lines were the phenotypes observed, how many different organoids were analyzed, do they correspond to different batches of differentiation, etc. This is important information that would support and give credibility to the findings. Examples of this are scRNA-seq experiments (Figs. 1d-f, 3b, etc.) and immunostaining quantifications (Figs. 1c, 2e, 3e, etc.). Quantifications would benefit from distinguishing between sections from different cell lines or organoids for example.

4) The authors propose an excitatory/inhibitory imbalance. Understanding what impact these changes have on electrophysiological properties of developing circuits would be an important experiment to include. It would be desirable if the authors could show some functional consequences that could be a mechanistic basis for circuits abnormalities and the phenotype observed in humans. Some suggestions could be electrophysiological experiments to assess frequency of spontaneous inhibitory events or miniature inhibitory postsynaptic currents, or measure activity through microelectrode arrays or calcium imaging. It would be great if the authors could include any experiment in this direction.

5) The authors have very elegantly generated a collection of cell lines that include isogenic correction of the mutation, mutation in control individual, and loss of function lines. While it would be ideal in the single cell RNA-seq analysis was extended to these other lines, it is understandable that due to cost it may be difficult. However, the paper would be greatly improved if this different conditions and different lines generated were used for validation of the phenotype through immunostainings, in situ hybridization or others.

Minor comments

1) There is some inconsistency in the description of how many lines were generated and used for what. In particular between line 146 (2 patients and 2 control lines) and Fig. 1a (3 isogenic pairs). It is also unclear whether the isogenic pairs generated and mentioned in Suppl. Fig. 1e-f are indeed used throughout the paper, making it unclear whether it is proper to say isogenic in this case.

2) Legend of Suppl. Fig. 1d says there are similar protein levels of PACS1 between control and mutant neuronal progenitors. However, this is not evident from the Western blot presented and authors should consider including a quantification. Is it possible to speculate or propose follow up experiments to assay the mechanism?

3) In lines 144 and 151 the authors refer to lamination and layering of the organoids. These terms are misleading as one would expect they refer to the 6 layers organization of the cortex, while they are indeed referring to the separation between the ventricular zone and post-mitotic neurons. Another term would be preferred to refer to tissue organization of this kind.

4) Legends in Fig. 1c and Suppl. Fig. 1d should be clear as to what is each dot representing (averaged sections from different organoids? Individual sections?). If each dot corresponds to a different section, how many of those come from different organoids or different lines. Additionally, how is the ratio calculated (total number of cells in a given area?). Even if this information is in the methods section (which is not) it should be included in the legend briefly.

5) Leptomeningeal cells or VLMC appear as an abundant population. This is not a population frequently found in forebrain organoids and their presence should be further discussed, corroborated or questioned. Also see Major comment #2. Additionally, the legend in Fig. 1f refers to these cells as ependymal cells, which is a different cell type as VLMC.

6) In Fig. 2a differentially expressed genes calculated over the totality of the cells are referred to

us "Bulk", maybe "Pseudo-bulk" is a better term? Same in Suppl. Fig. 4c. Additionally, these two figures contain virtually the same information, making them redundant.

7) It is not clear why the authors have decided to quantify ZIC1+ cells relative to CTIP+ cells in Fig. 2e. Both markers do not seem to be co-expressed in the same cells, and the number of CTIP2+ cells may be affected by genotype. Relative number of ZIC1+ cells to total number of DAPI+ nuclei in an area or to the number of neurons would be a better way of quantification.

8) In line 201 the authors say that "... the most mature neuron group showed enrichment for terms related to GABAergic differentiation, even though we utilized a directed forebrain organoid protocol (Fig. 3a)". However, forebrain organoids contain interneurons, and is cortical organoids that lack them. It seems then that the regional identity of the organoids is not clear and should be clarified.

9) Terms of interest should be highlighted in Suppl. Fig. 5.

10) The pseudotime analysis presented in Fig. 4b shows a pseudotime trajectory finishing in what was identified by the authors as TBR1+ LHX9+ newborn neurons. This is unexpected if these indeed correspond to immature, newborn neurons. Following previous comments, the authors should consider further and careful classification of the neurons cluster. Is it possible they correspond to a different class of neurons instead of immature ones?

11) Correlation analysis in Suppl. Fig. 6b would be improved if p-values are shown. Additionally, coloring dots by sample or line origin would be desirable.

12) In the WGCNA analysis in Fig. 6d module brown is shown to be composed only of ribosomal proteins. However ribosomal proteins were removed from DEG analysis at an earlier point. It is suggested that the authors choose a consistent approach.

13) Tree map in Fig. 6f should be clarified in the legend. What does area correspond to?

14) PSD95+ puncta size is strikingly different between conditions in Fig. 5c. Is this a consistent thing that should be considered? Otherwise, a better more representative image could be used?

15) The ASD gene modules used in Fig. 6e lack explanation. How were they generated? Are they overexpressed in ASD? How was analysis done?

16) Incomplete methods: some specifications are missing about image quantification, parameters for single cell analysis, gene ontology, overlap between DEG and OMIM risk genes, overlap between WGCNA and ASD modules. Software, software versions and function parameters would be ideal to guarantee reproducibility of the analysis.

Reviewer #2 (Remarks to the Author):

The manuscript by Rylaarsdam et al uses brain organoids to model the p.R203W variant in the gene PACS1 that is associated with rare forms of mental retardation. There is a high need to understand the disease mechanisms and identify treatment for these defects. The work is therefore potentially highly relevant for the research and patient communities. CRISPR/Cas9 and brain organoid models are powerful technologies, and the authors take a good use of this approach through detailed single cell transcriptomics.

The main weakness of this work is that the findings appear preliminary, as the claims are not supported by sufficient experimental evidence.

Major points:

- The authors used CRISPR/Cas9 to introduce the p.R203W variant in one allele of a control iPSC line and to correct the same heterozygous variant in two patient-derived iPSCs. In the control background, they also generated a KO for the PACS1 gene. This is all potentially very nice, but the editing proofs and validation reported are not sufficient. There is no data showing that the karyotype of edited lines is normal. The sequencing of the region of interest is only conducted in some lines. The protein expression data is missing for the KO line. There are no experiments to monitor the lack of any off-target effects. The authors write that they purchased 3 iPSC lines (1 control and 2 patients) from Coriell. However, the catalog numbers they provided refer to skin fibroblasts, not to iPSCs. So it is not clear how these iPSCs were generated. Proofs of pluripotency and karyotype analysis of unedited iPSCs are also missing. Moreover, there are no indication in the text of the two additional control iPSCs, and staining images of control 3 are also missing.

- The manuscript is mainly based on one single cell RNA seq analysis. Therefore, a good experimental set up is critical, given the central role of RNAseq in this report. However, the samples used for RNAseq analysis are not clear. The samples are not mentioned in the methods. According to Fig S3b, the samples were only 7 (without replicates as far as I can tell) and included 2 controls, 4 patients (what are patient 2.2 and 2.3 is not clear to me), and 1 control line edited to harbor the p.R203W variant. Therefore if the samples used were really these, I am not sure if the data are strong enough to support the whole paper and all the claims. Different replicates should be included and the authors should show that these different independent replicates have similar profiles. These important quality controls should be performed and showed. Moreover, Fig 1A is then misleading as it suggests that 3 isogenic lines were used for all experiments (including RNAseq), while instead only 1 isogenic pair was included for sequencing.
- Brain organoids are analyzed at day 40. But the reason for this choice is not clear. The authors write "time point chosen based on the onset of high PACS1 expression shown in Allen Brain Atlas". And they use this time point for transcriptomics analysis. For what I can see from Fig S3a, the expression of PACS1 starts only after 12 weeks. Based on those data, one could argue that later time points (like day 70) may be more appropriate. In fact, in Fig 1g the expression of PACS1 seems to be associated mainly with more mature MAPT-positive neuronal population. In agreement with this, different expression of GABAergic neurons can be seen at day 79 (Fig 5).
- The claim that "PACS1 p.R203W variant does not impact maturation and laminar organization of forebrain organoids" does not appear to be supported by sufficient evidence. The authors should also look at later time point. It could well be that at day 79 there are differences in maturation and organization, in a similar way as the differences in GABAergic neurons expression. Moreover, the quantification in Fig 1c seems to have been done only with 5 mutant organoids. This seems quite a limited number of samples to support such broad claim. Additional time points and additional markers of mature neurons and progenitors should be used to validate this claim. Moreover, the overall growth size of the organoids should also be monitored over time.
- The authors aimed to determine whether the p.R203W variant may cause a loss or gain of function. This is an important point worth being investigated. They found that KO PACS1 organoids developed normally at day 40. However, this time point may be too early (see comments above). So the claim seems to me not supported by strong experimental evidence. In fact, there is no detailed transcriptional analysis of these KO organoids and the level of PACS1 protein expression is also not assessed in the KO line. It seems to me that the CTIP2/SOX2 ratio in KO organoids is different from that of control organoids (Fig S2d) (but I am not sure how many independent analyses were done). Since this is such an important aspect (even in the title), more experiments should be done to validate the claim. Extended time points and more markers of mature neurons vs progenitors (not only one for neurons and one for progenitors) should be analyzed in KO organoids, before claiming that KO organoids are similar to healthy organoids. The authors speculate that this could even be an angle to develop treatment strategies. But nothing was experimentally tested. There are several things that the authors could do to prove this claim. For example, they could knock-down PACS1 in mutant organoids and see whether this can really work as a corrective action. These experiments would be required to fully prove that the p.R203W variant cause only a gain of function. Some of the downstream function of PACS1 (e.g. related to the reported role in golgi-related function) should also be analyzed to prove that the KO does not produce loss of function effects. If these experiments are not included, then the claim about gain of function (even in the title) remain unsupported.
- The claim that "PACS1(+/R203W) organoids have a propensity towards GABAergic differentiation" would require more validations. One single marker for GABAergic neurons (Fig 3d-e) is not sufficient. Additional markers and maybe also different 2D differentiation experiments could be included to support this claim.
- The fact that the pathogenic variant leads to "higher density of GABAergic synapses" should be confirmed using other assays. The differences in Fig S6b-c also do not seem very strong. Functional assessments would be required, for example with calcium imaging, MEAs, or electrophysiology. The use of one pair of GABAergic synaptic markers (Fig 5) is not sufficient to prove functional changes.
- The data related to ASD-specific gene network (Fig 6 and Fig S7) and specifically the supposedly enrichment of PACS1 with respect to synaptic function (Fig 6f) remain quite speculative in my view, given the lack of validations. Overall I find the data of Fig 6 interesting but very speculative and preliminary if not supported by more experimental evidence.

Additional points:

- Information in all the figures is missing with respect to which lines were used and how many independent experiments were performed. It is not sufficient to say "n=9 PACS1(+/-R203W) organoids". It is important to know which line (patient or control background) and whether the organoids come from the same line. In fact, validations should be carried out on multiple lines and with different independent experiments.
- The pseudotime analysis in Fig 4 is in principle well done. I just find it less interesting and a bit redundant, since it arrives at the same conclusion (ZIC transcripts are altered) as we have seen already in Fig 2. Perhaps the authors can find a way to better merge these complementary analyses. In my opinion, some images (like Fig 4d) may be more appropriate for supplementary data.
- Fig 1d, 1f, 1g. It is not clear to me if these are data from control or mutant organoids. The profiles for both genotypes should be provided (with detailed description of which line and the number of replicates).
- Fig 2a, Fig S4c. It is not clear how the up or down regulated genes were identified. In some cases, there are clearly no differences (see for example MALAT1 in IPCs and newborn neurons). But still these transcripts are included in the images. This is confusing.
- Fig 2c and Fig S3d. It is not clear if these are the patterns of mutant organoids and the one of control organoids.
- Fig 2b. It should be better explained how these data were generated and what is their meaning.
- It is written that figure legends for Fig 2a can be found in Fig S4c but I could not find these.
- The quantification analyses of the staining are not explained in the methods. Details regarding the number of fields per slice analyzed should also be included. For synaptic marker quantification there is an explanation, but it is not entirely clear whether the co-localization with MAP2 was included, as this is mentioned in the text but not shown in the images.
- The authors mention in Table S1F that they generated homozygous iPSCs, but there is no proof or details for this line. I would suggest to eliminate this claim, if it is not supported by evidence.
- Additional transcriptomics datasets were included as shown in Fig S4a. However, it would be important to see that the expression pattern of PACS1 and of other key transcripts (like ZIC1, and GAD1) are similar between the original controls and these additional control organoids.

Reviewer #3 (Remarks to the Author):

In the current study, Rylaarsdam et al. set out to study the cellular and molecular changes associated with a recurrent mutation (R203W) in the PACS1 gene that is causal to the PACS1 syndrome. The authors used both patient and control iPSC lines, as well as isogenic iPSC lines generated by introducing or correcting the genetic variant. Organoids were generated using these cell lines for mechanistic insights. They performed single-cell RNA-seq at one selected time point and found changes in the inhibitory lineage cells associated with the pathogenic mutation. However, some major concerns remain.

1. The study used both male and female iPSC lines for organoid formation. However, it seemed that data analyses and discussions were solely based on the different genotypes. There was no information provided in terms of the similarity/reproducibility among organoids from different genetic backgrounds with or without the mutation, let alone the batch effects between different experiments. Also, it was not clear if any sex-dependent differences were noticed in both control and mutant conditions.
2. In Figure 1, the TSNE plot shows that PACS1 is expressed in different clusters with variable expression levels, it is good to show its level across clusters as for other marker genes in 1f. Does the variant have any impact on the RNA or Protein stability? Is PACS1 expression level consistent across the different genetic backgrounds in iPSC cells?
3. While the method used to make organoids preferentially generate excitatory neurons, it by no means excludes the existence of inhibitory neurons in the same organoids. This is consistent with the expression of the inhibitory neuron marker gene in the wild-type condition (figures 3 and 4). It is not clear why the authors considered cells that express GAD1, SLC32A1, DLX 1, etc to be

excitatory neurons. Similarly, it is puzzling that it was described that ectopic GABAergic expression was found in excitatory cells (Figure 4a). PSD95 and GRIN2B are not inhibitory neuron marker genes. Both excitatory and inhibitory neurons express receptors for glutamine and GABA. It is possible that more inhibitory neurons are generated in the mutant organoids. Did the authors detect more Nkx2.1, LHX6, SP8, or other genes that are critical for interneuron development? It would also be critical to determine if this phenotype is temporary and if the differences between the mutant and control organoids could sustain into more mature stages.

4. What are the genes in different modules in Figure 4d?

5. When analyzing the synaptic proteins, the quantification method is troublesome. The authors used puncta/soma to measure the density of synapses, which did not consider the number of neurons of all the cells included or the complexity of neurons, e.g., the number and length of dendrites.

Some minor points:

1. The authors mentioned that cell lines that were engineered using CRISPR had been confirmed using WES, which was undoubtedly rigorous and thorough. Were any off-target changes detected?

2. PAX6 staining in supplementary figure 2b: showed mostly cytoplasmic signal without clear cellular boundary. Is this an artifact from the staining?

3. Gm03651 from Coriell was labeled as both a male and a female line.

4. In Figure 2a, there is no legend for the color or dot size.

5. Supplemental figure 6: there are vesicle transporter genes in the list, these are not receptors.

Re: NCOMMS-22-18984-T

We would like to thank the editor and reviewers for their time and useful comments. We received many helpful suggestions from reviewers that we have put forth great effort to address, therefore substantially improving our manuscript. We hope it is now suitable for publication in *Nature Communications*. Briefly, to address Reviewer 2's concern that day 40 was too early a timepoint for single-cell RNA sequencing, we added four day 88 organoid samples (NEW Fig. 2-3). We performed both calcium imaging (NEW Fig. 4) and multielectrode array (MEA) experiments (NEW Fig. 5) at the suggestion of Reviewers 1 and 2 to provide a functional component to our analysis. Further quality control measures were taken including organoid immunohistochemistry analysis of dorsal forebrain markers (NEW Fig. 1b-c; Fig. S2b-c) and subsequent quantifications (NEW Fig. 1d-e) suggested by Reviewer 1, exome sequencing of CRISPR generated isogenic pairs (NEW Fig. S1b) suggested by Reviewer 3, and reclassification of cell types (NEW Fig. 2a) based on all reviewers' concerns. We have also taken care to be explicit about exactly how many samples from each cell line were involved in each experiment – a critique of all reviewers - by including NEW Table S1 and showing results by cell line (NEW Fig. S2d-e; Fig. S5; Fig. S7; Fig. S8b-d; and Fig. S9a). All three isogenic pairs were incorporated into further analysis at the suggestion of Reviewer 1 – particularly the MEA analysis in which each pair demonstrated an increase in network burst duration in PACS1^(+/R203W) neurons (NEW Fig. 5l; Fig. S9a). We thank the reviewers for their comments as they helped us discover important caveats to our previous manuscript, such as the GABAergic population present at day 40 is not maintained, a question asked by Reviewer 3. We have therefore shifted the overall focus of the manuscript to the mature glutamatergic population and the functional deficits found with calcium imaging and multielectrode arrays. Our point-by-point responses to reviewer comments in italics can be found below in red regular font.

Reviewer #1 (Remarks to the Author):

Rylaarsdam et al. study a PACS1 syndrome, a neurodevelopmental disorder that causes intellectual disability and craniofacial abnormalities. They describe the effect of a PACS1 mutation (R203W) on the development and gene expression of human forebrain organoids. By using patient derived lines and additional edited cell lines, they are able to show that organoids carrying the mutation have an increased number of GABAergic cells, increased expression levels of associated genes, and a dysregulation in the number of inhibitory synapses. Furthermore, they show an overlap between the upregulated genes in the PACS1 mutant organoids and overexpressed genesets in ASD models, suggesting some phenotypic convergence.

Literature on PACS1 syndrome is very limited, and no in vitro human models have been used to date to understand PACS1 function during development or the consequences of the R203W mutation. The present study is original in its approach and findings and will be highly significant for understanding the mechanisms underlying PACS1 syndrome during brain development.

Generally, the paper is well written and findings are well supported. Data is well presented and analyzed. However, there are some important issues surrounding the single cell RNA-seq analysis that need to be addressed to properly support the current interpretation and conclusions. Besides these meaningful issues, the methodology is sound and meets the expected standards. In most cases enough detail is provided in the methods section for the work to be reproduced. The cases where it is lacking have been highlighted.

This paper would be an important contribution to the field of neurodevelopmental disorders. However, the paper in its current form has concerns to be addressed before acceptance.

Major comments

1) While the authors specify that they are using forebrain organoids, the regional identity of those organoids has not been properly shown. This becomes particularly important as some of the DEG correspond with regional differences in mouse and human brain, such as ZIC1, ZIC2, LHX9, FOXG1; and considering some other surprising observations by the authors that are not frequent in forebrain organoid literature, such as PAX3 expression in intermediate progenitors and the abundant VLMC. The regional identity of these forebrain

organoids could be shown, for example, with scRNA-seq and immunostainings showing extensive *FOXG1* expression - as done in other papers in the field like Bhaduri et al. 2020, or *EMX1* expression to show cortical fate as done in other works. Proper regional and cell type identities are of the utmost importance to correctly interpret changes in gene expression upon *PACS1* mutation. It is also extremely relevant when choosing other datasets to compare for DEG calculation, so that the authors are sure to include datasets (or subsets of them) coming from organoids with similar identities (which should be shown with expression of the same genes).

We agree with Reviewer 1 that this very important aspect was not fully characterized in the initial report. NEW Fig. 1b-c and NEW Fig. S2b-c now show extensive expression of *FOXG1*, *EMX1*, and *SATB2* (at day 80) in addition to the previously shown *PAX6*, *SOX2*, *TBR1*, and *CTIP2*. This supports a dorsal forebrain identity similar to other differentiations produced with this protocol. Our intermediate progenitors are now characterized by colocalization of *EOMES*, *HES6*, and *NEUROG1* (NEW Fig. S4c) instead of *PAX3*. Many of these regional DEGs are no longer relevant with our new, more rigorous DEG scheme (see methods, page 14).

We acknowledge Reviewer 1's concern over the presence of markers such as *ZIC1* and *COL1A2* (VLMC) markers in our data. We would like to point out that these markers also are present in publicly available data from organoids generated with the same protocol. Below are plots from the web platform GECO, which show relative expression levels of transcripts in forebrain organoids generated by Sergiu Pasca's group. GECO shows robust expression of some "problematic" transcripts like *DLX1*, *ZIC1*, and *COL1A2* at comparable levels to *BCL11B/CTIP2* (please see plot below).

The methods paper of this protocol Sloan et al., *Nat Protoc.*, 2018 (PMID: 30202107), Table 1 also reports sporadic *GAD67* and *GABA* expression in the dorsal forebrain organoids. While the VLMC group also expresses markers of radial glia, we concluded it was more accurate to call them VLMC due to the strong colocalization of *LUM* and *DCN* (see NEW Fig. S4c). In our incorporation of public data at day 40, a corresponding group was resolved in both the Kriegstein and Testa lab organoids which hierarchically clusters to our VLMCs (please see heatmap below).

A

The choroid plexus group is briefly mentioned in the Birey *et al.*, *Nature*, 2017 paper: “Astroglia from both hCS and hSS clustered together and close to a small group of cells that resemble the choroid plexus (*TTR*⁺, *SLC13A4*⁺).” *FOLR1* is also listed as a top marker of this group in Supplementary Table 3. We have shown feature plots of these markers below, which clearly overlaps with the choroid plexus group in our data. Finally, we have no longer included the public data for DEG analysis in this revised manuscript, which is discussed more in-depth in response to future comments.

2) In a similar line, the authors propose an upregulation of GABAergic genes, as well as other genes involved in regional patterning including *ZIC1*, *ZIC2*, *LHX9* and *FOXP1*. However, these genes are not consistently overexpressed (compare for example *ZIC1* and *GAD1* feature plots) suggesting that there may be more than one phenotypic change.

We see Reviewer 1's point and acknowledge that this upregulation was more prevalent in some cell lines than others. We tried to get around this by constructing a consensus scheme with comparisons between multiple groups of controls and batches, but our results suffered from each cell being treated as an "n". For this resubmission we have initiated a much more rigorous DEG identification scheme with pseudobulk validation that has a very low tolerance for DEGs mostly driven by one or two samples (see Methods, page 14). As a result, our DEGs at day 40 have been reduced from 637 to 41. We now focus on phenotypic changes like an increase in network burst duration that was prevalent in all three isogenic pairs (NEW Fig. S9a).

ZIC genes overexpression is suggestive of a change in regional identity, that should at least be considered. In light of these observations, it becomes important to address: i-is there a change in expression profiles or a change in proportions of neuronal subtypes?

We agree with Reviewer 1 that it is of utmost importance to distinguish between a change in expression profile vs. proportions of neuronal subtypes. In this revised manuscript we have added four day 88 samples for single-cell RNA sequencing at the suggestion of reviewers. *ZIC1* showed high variability at day 88 and, due to lack of consistency, was therefore not further considered. We do not find any change in the proportion of subtypes as shown in NEW Fig. S5c and the transcriptomes of each sample generally cluster together by cell type as shown in NEW Fig. 2d, suggesting they are comparable across samples. For these reasons we do not think a change in proportions of neuronal subtypes is occurring.

ii-what is the molecular identity of the GABAergic neurons?

We thank Reviewer 1 for bringing up this very important question. These GABAergic neurons were not positive for MGE markers, but did have slight colocalization of *SP8*, potentially suggesting LGE/CGE identity. At the suggestion of reviewers, we have incorporated four day 88 samples for single-cell RNA sequencing in our revised manuscript. This helped illuminate important caveats to our previous study, including the observation that the GABAergic population present at day 40 appears to be transient as it is diminished at day 88 (NEW Fig. S5c). We have therefore moved on to a different focus, but this could be a valuable topic of future investigation.

iii-is there a change in patterning or regional information in the mutant organoids? With respect to the first question, the excitatory neuron cluster seems indeed to include all kind of neurons. Markers used are pan neuronal (*MAPT2*, *SYT1* and *STMN2*) and not suitable to distinguish between excitatory and inhibitory neurons.

BCL11B is a neuronal subtype marker gene that is expressed by both excitatory and inhibitory neurons.

Reviewer 1 is correct that *BCL11B* can also be expressed in inhibitory neurons - thanks for bringing this to our attention. After feedback from reviewers, we have changed our cell type annotation to include a group of immature GABAergic neurons as well as another group of general immature neurons, which expresses both GABAergic and glutamatergic markers (NEW Fig. 2a, Fig. S4c). We have also further distinguished our glutamatergic neuron clusters as seen in NEW Fig. 2a. However, it won't be accurate to further specify them according to *in vivo* neuron populations – for example, “layer V-VI corticofugal projection neurons”, as the absence of the identical spatial and temporal cues in the developing brain leads to differentiation of similar but more general cell types.

Considering the restricted and high expression of inhibitory genes DLX and GAD1 (Fig. 3c) it is surprising that no cluster was annotated as inhibitory neurons to begin with in Fig. 1. The correct annotation is critical to fully understand if there is a change in cell proportions of different neuronal classes between control and patient derived organoids, and to be able to properly choose datasets for DEG comparison later in the paper.

We understand Reviewer 1's concern that the organoids have not been properly annotated. The average transcriptome of these cells was determined to be most similar to excitatory neurons overall with our bioinformatic comparison to a reference human fetal transcriptome (Nowakowski *et al.*, *Science*, 2017. PMID: 29217575). However, upon further investigation, we agree with Reviewer 1 and have now called these cells immature GABAergic neurons. An additional cluster which weakly expresses both GABAergic and glutamatergic markers is now called "immature neurons" as it seems to be a mix (NEW Fig. 2a, NEW Fig. S4c).

With respect to that last point, for calculation of DEG the authors chose to include published datasets (a smart idea to compensate for lower number of replicates and add power to the analysis). However, while it is stated that they were "generated with the same protocol and collected at the same time point next to data collected in this study" that is not fully the case.

We thank Reviewer 1 for pointing this out. Our study uses the human cortical spheroid protocol developed by Sergiu Pasca's group published in Birey *et al.*, *Nature*, 2017 (PMID: 28445465) and Sloan *et al.*, *Nat Protoc*, 2018 (PMID: 30202107). We revisited the supplementary methods of papers used for public data and confirmed that the Lopez-Tobon *et al.*, *Stem Cell Reports*, 2019 (PMID: 31607568) study used this protocol, but Reviewer 1 is correct that we made an error with the Bhaduri *et al.*, *Nature*, 2020 (PMID:31996853) study. We sincerely thank Reviewer 1 for pointing this out and apologize for not being more careful. The error arose from a misunderstanding in terminology and looking at the supplementary methods; we now see that Bhaduri *et al.* considers the Pasca protocol directed - not most directed - but uses an adapted version and should not be compared. The public data has been removed from the study and we thank Reviewer 1 for catching what we should have caught ourselves.

To compensate for lower number of replicates, we have developed a more stringent DEG identification scheme in which traditional approaches are validated with a pseudobulk method. We have now updated the methods (page 14): "Following cell type identification, differential expression analysis was run between genotypes for each cell type and time point. A standard single-cell differential expression workflow was first performed using the `fit_models` function in the `monocle3` package according to the standard workflow where each cell is treated as an "n". To increase confidence of results, we validated hits using a pseudobulk approach with DESeq2 (v1.30.1). First the count matrix was aggregated for each day/cell time and features with <5 counts for any aggregated sample were discarded. DESeq2 was the run followed with an `lfcShrink` transformation (type = `apeglm`) of results. These results were then joined by cell type, day, and direction of change. Genes with < 0.5 absolute fold change or > 0.05 p_{adj} for either method were discarded. Genes with a base mean of < 60 were also discarded. Mitochondrial (MT-) and ribosomal (RPS/RPL) proteins were removed (only 5). 277 DEGs across all cell types and both time points remained in the final list."

Having neurons properly classified (at least between excitatory and inhibitory) then becomes critical to be able to set suitable DEG comparisons. For instance, control organoids from this paper contain at least one clear cluster of neurons that express inhibitory genes (ARX, DLX1, DLX2, DLX-ASI, GAD1, Fig. 3c), while very few cells are annotated as such in Bhaduri et al (see Fig. 1 in their paper).

We agree with R1's concern and now have included a class of cells labeled as GABAergic neurons in day 40 organoids (NEW Fig. 2a). We have increased the resolution of our cluster distinction in an effort to find true DEGs and not results driven by subtle differential proportions of cell types. We have also extended our transcriptomic analysis at a later timepoint. At day 88, fewer of these GABAergic cells are seen (NEW Fig. S5c), suggesting that this is a transient population, and therefore they are no longer the focus of this study.

Importantly also, the authors are comparing to Bhaduri's most directed organoids, in which one would expect even less inhibitory cells as differentiation has been directed to a cortical fate. Differences in cell type proportions are also expecting as a consequence of using different cell lines. Taking all these elements into consideration, it is recommended that comparisons between the three control groups are also carried out to confirm that the genes arising are not DEG among organoids protocols to begin with.

This is a helpful recommendation. We had set up the DEG schema so that a DEG would not be considered a hit unless it was significant when compared to all three groups of controls. However, we should have also looked at the DEGs between control groups, which might have helped us catch the error with the Bhaduri *et al.* dataset. We will certainly do this if we incorporate public data in the future in addition to further quality control, such as the figure below which shows the transcriptome of day 40 organoids divided by sample and cell type cluster together across different labs. This supports they are comparable if DEG analysis is constructed appropriately. For this study, we have chosen to eliminate public data and instead institute a new DEG scheme including a pseudobulk validation step. This will avoid some of the inevitable discrepancies between samples produced from different labs. Please see methods (page 14) or the excerpt above.

3) *The authors use cell lines derived from 2 patients and 3 controls, plus one additional PACS1 R203W edited line. The use of multiple lines is important as it gives support to the relevance and penetrance of the phenotype. However, it is not emphasized throughout the paper in how many cell lines were the phenotypes observed, how many different organoids were analyzed, do they correspond to different batches of differentiation, etc. This is important information that would support and give credibility to the findings. Examples of this are scRNA-seq experiments (Figs. 1d-f, 3b, etc.) and immunostaining quantifications (Figs. 1c, 2e, 3e, etc.). Quantifications would benefit from distinguishing between sections from different cell lines or organoids for example.*

We thank Reviewer 1 for the suggestion. We strived to be more transparent in this manuscript and have included a table (NEW Table S1) that thoroughly describes the number of samples from each line used in each experiment. We also show results of major conclusions by cell line in the supplements (NEW Fig. S2d-e; Fig. S5; Fig. S7; Fig. S8b-d; and Fig. S9a).

4) *The authors propose an excitatory/inhibitory imbalance. Understanding what impact these changes have on electrophysiological properties of developing circuits would be an important experiment to include. It would be desirable if the authors could show some functional consequences that could be a mechanistic basis for circuit abnormalities and the phenotype observed in humans. Some suggestions could be electrophysiological experiments to assess frequency of spontaneous inhibitory events or miniature inhibitory postsynaptic currents, or measure activity through microelectrode arrays or calcium imaging. It would be great if the authors could include any experiment in this direction.*

We wholeheartedly agree with Reviewer 1 that functional analysis would add a lot to the study and went to great lengths to include both calcium imaging (NEW Fig. 4) and multielectrode array data (NEW Fig. 5) in the revised paper, as we did not have the capabilities for patch clamping. Calcium imaging in organoids (NEW Fig. 4) showed an increase in spike frequency in PACS1^(+/R203W) lines that was independent of GABA-A receptors (GABA may have an excitatory effect at this time point due to higher expression of NKCC1 than KCC2; NEW Fig. S8e). Because the GABAergic population was not maintained at day 88, we instead focused on the glutamatergic populations at a higher resolution with MEA data and found a consistent and drastic increase in network burst duration in PACS1^(+/R203W) neurons (NEW Fig. 5l; Fig. S9a).

5) *The authors have very elegantly generated a collection of cell lines that include isogenic correction of the mutation, mutation in control individual, and loss of function lines. While it would be ideal in the single cell RNA-seq analysis was extended to these other lines, it is understandable that due to cost it may be difficult. However, the paper would be greatly improved if this different conditions and different lines generated were used for validation of the phenotype through immunostainings, in situ hybridization or others.*

We agree with Reviewer 1 and acknowledge that for the single-cell analysis, the isogenic lines were not utilized to their fullest potential, partly due to the chronology of sample submission and line generation. For follow-up analysis (NEW Fig. 4-5; Fig. S8b-d; Fig. S9a), we made a strong effort to more fully utilize the isogenic pairs. We have also included a clearer breakdown of major conclusions by cell line in the supplements (NEW Fig. S2d-e; Fig. S5; Fig. S7; Fig. S8b-d; and Fig. S9a). The MEA experiments, for example, included all three sets with multiple differentiations, results of which are shown by isogenic pair in NEW Fig. S9a and are consistent for all conditions. NEW Table S1 now clearly states what was used for each experiment.

Minor comments

1) *There is some inconsistency in the description of how many lines were generated and used for what. In particular between line 146 (2 patients and 2 control lines) and Fig. 1a (3 isogenic pairs). It is also unclear whether the isogenic pairs generated and mentioned in Suppl. Fig. 1e-f are indeed used throughout the paper, making it unclear whether it is proper to say isogenic in this case.*

We apologize for lack of clarity. NEW Table S1 is now provided to clearly state what was used for each experiment. We have also included a clearer breakdown of major conclusions by cell line in the supplements (NEW Fig. S2d-e; Fig. S5; Fig. S7; Fig. S8b-d; and Fig. S9a). Our strongest conclusion is the increase in

network burst duration in PACS1^(+/R203W) neurons, results of which are shown by isogenic pair in NEW Fig. S9a and are consistent for all conditions.

2) *Legend of Suppl. Fig. 1d says there are similar protein levels of PACS1 between control and mutant neuronal progenitors. However, this is not evident from the Western blot presented and authors should consider including a quantification.*

We now include a western blot of PACS1 day 40 organoids in NEW Fig. 1f with quantification in g.

Is it possible to speculate or propose follow up experiments to assay the mechanism?

We now propose follow-up experiments in the discussion to better understand the mechanism of the PACS1 p.R203W variant, such as this line on the discussion (page 10): “Future studies will involve crystallography to try and resolve the structure of the FBR in the mutant and control PACS1, immunoprecipitation assays in neurons to establish if binding partners are lost or acquired upon introduction of the variant, binding affinity assays to determine if p.R203W binds variably to targets, and immunohistochemistry of PACS1 and its targets to assess how the variant impacts subcellular localization in order to better inform therapeutic development.” We also suggest differentially expressed ion channels such as TRPC5 as promising pharmacological targets for further evaluation.

3) *In lines 144 and 151 the authors refer to lamination and layering of the organoids. These terms are misleading as one would expect they refer to the 6 layers organization of the cortex, while they are indeed referring to the separation between the ventricular zone and post-mitotic neurons. Another term would be preferred to refer to tissue organization of this kind.*

We thank Reviewer 1 for pointing out this error. It has been corrected and we no longer refer to the separation between the ventricular-like zone and post-mitotic zone as lamination.

4) *Legends in Fig. 1c and Suppl. Fig. 1d should be clear as to what is each dot representing (averaged sections from different organoids? Individual sections?). If each dot corresponds to a different section, how many of those come from different organoids or different lines. Additionally, how is the ratio calculated (total number of cells in a given area?). Even if this information is in the methods section (which is not) it should be included in the legend briefly.*

We apologize for this lack of clarity. What each dot is representing and how many organoids they came from is now clearly stated in each legend. NEW Table S1 has been included to show the number of samples from each line for every experiment. We have also included that the ratio refers to the number of cells visible in an image field in both the legend and the methods (page 12).

5) *Leptomeningeal cells or VLMC appear as an abundant population. This is not a population frequently found in forebrain organoids and their presence should be further discussed, corroborated or questioned. Also see Major comment #2. Additionally, the legend in Fig. 1f refers to these cells as ependymal cells, which is a different cell type as VLMC.*

We thank the reviewer for pointing out the mismatch in the figure legend and have corrected it. These cells express late radial glia markers (NEW Fig. S4c), but more strongly express classical VLMC markers like *COL1A2*, *LUM* and *DCN*. We therefore concluded it was more accurate to call them VLMC instead of late radial glia. As shown in the clustered heatmap above, similar populations are resolved from the public datasets at day 40. The sample with the highest population of VLMCs was discarded in this submission as it also had other features indicating poor quality. While the day 40 samples did have a rather high percentage of VLMCs, our day 88 samples have a much lower percentage (NEW Fig. S5c), which is now the timepoint of most interest.

6) In Fig. 2a differentially expressed genes calculated over the totality of the cells are referred to us “Bulk”, maybe “Pseudo-bulk” is a better term? Same in Suppl. Fig. 4c. Additionally, these two figures contain virtually the same information, making them redundant.

We thank Reviewer 1 for pointing out this potentially confusing terminology. We had included the figure in the supplement with each legend in hopes to make the main figure legend simpler but have not included this type of plot in the resubmission. Our new analysis does not include DEGs over the whole organoid – what we were referring to as “bulk” – but instead only use the term pseudobulk which refers to each organoid being treated as an n when doing DEG analysis by cell type. We hope the consistency helps to eliminate confusion.

7) It is not clear why the authors have decided to quantify ZIC1+ cells relative to CTIP+ cells in Fig. 2e. Both markers do not seem to be co-expressed in the same cells, and the number of CTIP2+ cells may be affected by genotype. Relative number of ZIC1+ cells to total number of DAPI+ nuclei in an area or to the number of neurons would be a better way of quantification.

We acknowledge this logic but found it inaccurate to do quantifications relative to DAPI, as minor changes in the minimum size threshold made such drastic differences in the DAPI count that they totally obscured the protein of interest. The presence of CTIP2 was used as an internal positive control, as another nuclear antibody with similar efficacy was needed to verify that area of tissue was healthy and receptive to staining. Regardless, these are not included in our revised manuscript because we sought to focus on pathway enrichment in the DEGs followed by functional consequences rather than singling out specific genes. We reasoned specific gene quantifications brought too much focus to certain DEGs when the goal of the study was to be broader.

8) In line 201 the authors say that “... the most mature neuron group showed enrichment for terms related to GABAergic differentiation, even though we utilized a directed forebrain organoid protocol (Fig. 3a)”. However, forebrain organoids contain interneurons, and is cortical organoids that lack them. It seems then that the regional identity of the organoids is not clear and should be clarified.

We thank Reviewer 1 for pointing this out. We are using the human cortical organoid protocol developed by Sergiu Pasca’s group and apologize for the inconsistency in terminology. We will change “forebrain organoid” to “cortical organoid” throughout the manuscript. Our organoids show markers such as FOXG1, TBR1, SATB2, SOX2, CTIP2, PAX6, and EMX1 (NEW Fig. 1b-e, Fig. S2b-c), suggesting our differentiations are producing similar results to what has been previously shown with this protocol.

9) Terms of interest should be highlighted in Suppl. Fig. 5.

We agree that previous Fig. S5 was overwhelming. We have not included this type of graph in our resubmission as we are now using hierarchical visualizations to reduce GO term redundancy (NEW Fig. 3c).

10) The pseudotime analysis presented in Fig. 4b shows a pseudotime trajectory finishing in what was identified by the authors as TBR1+ LHX9+ newborn neurons. This is unexpected if these indeed correspond to immature, newborn neurons. Following previous comments, the authors should consider further and careful classification of the neurons cluster. Is it possible they correspond to a different class of neurons instead of immature ones?

Reviewer 1 brings up a good point that perhaps the pseudotime analysis should have been run to allow loop structures so that an additional converging trajectory towards mature neurons along the TBR1+ branch would be followed. The pseudotime method is agnostic to biological meaning and is only dependent on a cell distance from the root node. We have carefully reassessed our neuron classifications and no longer find pseudotime analysis to be useful for the revised manuscript.

11) Correlation analysis in Suppl. Fig. 6b would be improved if p-values are shown. Additionally, coloring dots by sample or line origin would be desirable.

We have not included this analysis in the revised manuscript but have made efforts to show results more clearly by sample/cell line, such as NEW Fig. S2d-e; Fig. S5; Fig. S7; Fig. S8b-d; and Fig. S9a.

12) *In the WGCNA analysis in Fig. 6d module brown is shown to be composed only of ribosomal proteins. However ribosomal proteins were removed from DEG analysis at an earlier point. It is suggested that the authors choose a consistent approach.*

Reviewer 1 is correct that a consistent approach should be used. We suspected the brown module to be an artifact because of all the RPS/RPL genes and recognize the potential confusion with taking them out of DEG analysis but leaving them in WGCNA results. WGCNA is no longer included in the revised manuscript as we instead focus on functional characterization of PACS1^(+/R203W) neuron activity, but we will be take care to be more consistent in the future.

13) *Tree map in Fig. 6f should be clarified in the legend. What does area correspond to?*

We thank Reviewer 1 for pointing this out. While we don't include this exact chart anymore, we have similar representations in NEW Fig. 3c and now clearly state in the legend the width correlates to $-\log(p_{\text{adj}})$ of the GO term.

14) *PSD95+ puncta size is strikingly different between conditions in Fig. 5c. Is this a consistent thing that should be considered? Otherwise, a better more representative image could be used?*

We thank Reviewer 1 for pointing this out. This image was not representative of the typical PSD95+ puncta size and we have therefore replaced this panel (NEW Fig. 4b).

15) *The ASD gene modules used in Fig. 6e lack explanation. How were they generated? Are they overexpressed in ASD? How was analysis done?*

We apologize for the lack of clarity. These modules were from bulk RNA seq experiments in public data identified with WGCNA. We no longer include WGCNA analysis as we instead focus on functional characterization of PACS1^(+/R203W) neuron activity, and there is some discrepancy in the field whether it is appropriate to use each cell as an n when combining WGCNA and single-cell data.

16) *Incomplete methods: some specifications are missing about image quantification, parameters for single cell analysis, gene ontology, overlap between DEG and OMIM risk genes, overlap between WGCNA and ASD modules. Software, software versions and function parameters would be ideal to guarantee reproducibility of the analysis.*

We thank Reviewer 1 for pointing out these insufficiencies and have now made our methods and parameters as clear as possible. Software and package versions are now included throughout the methods (pages 10-19). Image quantification parameters are clearly stated on page 12: "For quantifications in Fig. 1d-e, one field of view per section was imaged with up to five sections per organoid. Images were analyzed using ImageJ (Fiji v2.3.0) and CellProfiler (v4.2.4). To make visualizations in Fig. 1d-e comparable, the total number of positive cells was determined per image field and the fraction each marker contributing to the number of positive cells was calculated, then plotted as averages per genotype and day in Fig. 1d and per organoid in Fig. S2d. For FOXG1/TBR1/SATB2 statistics, distribution of each marker fraction (averaged per organoid) was first tested for normality using a Shapiro test. FOXG1 and TBR1 returned $p > 0.05$ in the normality test so a two-way ANOVA with Tukey HSD correction was run. This was not the case for SATB2 ($p = 0.001$) and so a Kruskal Wallis test was done for genotype and day, then adjusted using a Benjamini Hochberg correction. In Fig. 1e, the ratio of SOX2+/CTIP2+ cells was calculated in R and averaged per organoid. A Shapiro test determined data did not follow a normal distribution ($p = 0.003$) and so a Kruskal Wallis test was done for genotype and day, then adjusted using a Benjamini Hochberg correction." Each major step of single-cell analysis is now outlined, with function parameters, on the methods (pages 13-14). We no longer include the OMIM and WGCNA analysis.

Reviewer #2 (Remarks to the Author):

The manuscript by Rylaarsdam et al uses brain organoids to model the p.R203W variant in the gene PACS1 that is associated with rare forms of mental retardation. There is a high need to understand the disease mechanisms and identify treatment for these defects. The work is therefore potentially highly relevant for the research and patient communities. CRISPR/Cas9 and brain organoid models are powerful technologies, and the authors take a good use of this approach through detailed single cell transcriptomics.

The main weakness of this work is that the findings appear preliminary, as the claims are not supported by sufficient experimental evidence.

Major points:

- The authors used CRISPR/Cas9 to introduce the p.R203W variant in one allele of a control iPSC line and to correct the same heterozygous variant in two patient-derived iPSCs. In the control background, they also generated a KO for the PACS1 gene. This is all potentially very nice, but the editing proofs and validation reported are not sufficient. There is no data showing that the karyotype of edited lines is normal. The sequencing of the region of interest is only conducted in some lines.

We thank Reviewer 2 for pointing out these insufficiencies. In addition to Sanger sequencing, we performed whole exome or genome sequencing on all isogenic pairs to confirm and validate the editing. NEW Fig. S1b now includes an IGV view of reads from all three isogenic line pairs. The KO, while successfully generated, is not included as we chose to focus on characterizing the difference between the PACS1^(+/+) and PACS1^(+/R203W) developing neurons in this study and more fully explore the mechanism of the variant with PACS1^(-/-), PACS1^(+/-), and PACS1^(R203W/R203W) lines in the future.

The protein expression data is missing for the KO line.

We thank Reviewer 2 for pointing out the missing PACS1 protein data for the KO line. We have since performed western blot analysis for PACS1 levels in day 40 organoids and included the KO line. However, since we no longer discuss the KO in the story, it is not in NEW Fig. 1f-g. The entire blot is shown here below.

There are no experiments to monitor the lack of any off-target effects.

We thank Reviewer 2 for pointing out we did not include this important control. We have submitted all three isogenic iPSC pairs for whole exome or genome sequencing. We checked the top off-target sites and did not find any unintended base edits. We have now updated the Methods (page 11) to include this information: "Of the top 100 off-target effects predicted by the IDT CRISPR design tool, only two other sites were exonic (MVP chr16:-29844665; PACS2 chr14:-105355130) and did not have any mismatched base pairs in any of the three isogenic sets of lines. The top 5 off-target sites predicted by CHOPCHOP (v3) were all non-coding, but analysis of the CRISPR R203W and CRISPR A1 lines - which received whole genome sequencing - did not reveal any unintended edits". An image of the IGV views of the three coding and top five non-coding sites is attached.

PACS1 p.R203W chr11:-66211212

MVP chr16:-29844665

PACS2 chr14:-105355130

chr1:155964574

chr6:127445345

chr3:43102718

chr1:231182256

chr17:15316928

The authors write that they purchased 3 iPSC lines (1 control and 2 patients) from Coriell. However, the catalog numbers they provided refer to skin fibroblasts, not to iPSCs. So it is not clear how these iPSCs were generated.

Reviewer 2 is correct, the catalog number that was provided refers to skin fibroblast. We sincerely apologize for this mistake. The control line S033751*B/GM27160 and a patient line S033745*B/GM27159 were purchased as iPSCs from Coriell before they were transferred to WiCell, where are now only available (PACS1001i-GM27160 and PACS1002i-GM27159). An additional patient line, PACS1003i-GM27161, was purchased as iPSCs from WiCell. This has been corrected in the methods (page 10).

Proofs of pluripotency and karyotype analysis of unedited iPSCs are also missing. Moreover, there are no indication in the text of the two additional control iPSCs, and staining images of control 3 are also missing.

WiCell carried out karyotyping for patient lines (Affected 1-S033745*B/GM27159 and Affected 2-GM27161), Control 1 (GM30651), and Control 3 (S033751*B/GM27160), as well as a pluripotency test for Control 3 and Affected 2. Normal karyotypes and generation of all three germ layers was demonstrated. We have now indicated this in the methods (page 10). Available karyotypes and pluripotency tests are shown below. For simplicity, Fig. S1c has been abbreviated to show a representative image from each genotype. Control 3 is shown below (DAPI is greyscale; Tra-1-60 is red; OCT4A is green). We now indicate more clearly in the text how many lines were used on page 4: “To investigate how the p.R203W variant in PACS1 impacts developing neural cell types, we generated dorsal cortical organoids from three control and two patient lines. In addition, we introduced the heterozygous variant which leads to p.R203W into one control line and corrected it in two patient lines using CRISPR/Cas9 gene editing technology.”

Gene	Fold change	Gene	Fold change	Gene	Fold change	Gene	Fold change
OCT4	0	PAX6	115	T	267	AFP	104773
SOX2	0	NES	0	RUNX1	2	SOX17	62
NANOG	0	TP63	5	DES	0	FOXA2	42
GDF3	2	KRT14	0	PECAM1	1	SOX7	1
REX01	0	NOG	20	TAL1	3		

Gene	Fold change	Gene	Fold change	Gene	Fold change	Gene	Fold change
OCT4	1	PAX6	37	T	32	AFP	4257
SOX2	0	NES	1	RUNX1	2	SOX17	72
NANOG	0	TP63	0	DES	0	FOXA2	682
GDF3	0	KRT14	0	PECAM1	0	SOX7	2
REX01	0	NOG	4	TAL1	0		

- The manuscript is mainly based on one single cell RNA seq analysis. Therefore, a good experimental set up is critical, given the central role of RNAseq in this report. However, the samples used for RNAseq analysis are not clear. The samples are not mentioned in the methods. According to Fig S3b, the samples were only 7 (without replicates as far as I can tell) and included 2 controls, 4 patients (what are patient 2.2 and 2.3 is not clear to me), and 1 control line edited to harbor the p.R203W variant. Therefore if the samples used were really these, I am not sure if the data are strong enough to support the whole paper and all the claims. Different replicates should be included and the authors should show that these different independent replicates have similar profiles. These important quality controls should be performed and showed.

We apologize for the lack of clarity and have now included NEW Table S1 to clearly show what was used. The patient 2.1, 2.2, and 2.3 initially referred to replicates of the same Affected 2 sample. In this resubmission we discarded sample 2.1 as it was of poor quality. 2.2 and 2.3 were combined as they were technical replicates. We therefore had five samples from day 40 and four samples and day 88 organoids. Due to cost of single cell RNA Seq, we could not add replicates to this experiment, but we made sure to do so for all other analyses. We agree with Reviewer 2 that validations should be carried out on multiple lines and with different independent experiments thus have now also included a clearer breakdown of major conclusions by cell line in the supplements (NEW Fig. S2d-e; S5; Fig. S7; Fig. S8b-d; Fig. S9a). For example, in our MEA analysis, we repeated it three times with three isogenic pairs and three neuron differentiations (NEW Fig. S9a). An increase in network burst duration was consistently demonstrated.

Moreover, Fig 1A is then misleading as it suggests that 3 isogenic lines were used for all experiments (including RNAseq), while instead only 1 isogenic pair was included for sequencing.

We agree with Reviewer 2 that Fig. 1a was misleading. We removed the n = 3 annotation from the NEW Fig. 1a.

- Brain organoids are analyzed at day 40. But the reason for this choice is not clear. The authors write “time point chosen based on the onset of high PACS1 expression shown in Allen Brain Atlas”. And they use this time point for transcriptomics analysis. For what I can see from Fig S3a, the expression of PACS1 starts only after 12 weeks. Based on those data, one could argue that later time points (like day 70) may be more appropriate. In fact, in Fig 1g the expression of PACS1 seems to be associated mainly with more mature MAPT-positive neuronal population. In agreement with this, different expression of GABAergic neurons can be seen at day 79 (Fig 5).

We agree with Reviewer 2 that PACS1 is in fact more abundantly expressed in more mature populations (NEW Fig. S5e) and recognized the value in a later timepoint. Therefore, we have added day 88 organoids to our transcriptomic analysis (NEW Fig. 2-3; Fig. S3-S7). It was clear from this new data that the GABAergic population was transient and have instead shifted our focus to the mature glutamatergic population. We thank Reviewer 2 for this suggestion.

- The claim that “PACS1 p.R203W variant does not impact maturation and laminar organization of forebrain organoids” does not appear to be supported by sufficient evidence. The authors should also look at later time point. It could well be that at day 79 there are differences in maturation and organization, in a similar way as the differences in GABAergic neurons expression.

Reviewer 2 brings up a valid point and we have included more extensive staining and quantification of dorsal forebrain markers in both day 40 and 80 organoids (NEW Fig. 1b-e and Fig. S2b-e). We do not find evidence of differential cell type distributions between genotypes at day 40 or 80 with either immunohistochemistry (NEW Fig. 1d-e; Fig S2d-e) or transcriptomic analysis (NEW Fig. S5a-c).

Moreover, the quantification in Fig 1c seems to have been done only with 5 mutant organoids. This seems quite a limited number of samples to support such broad claim. Additional time points and additional markers of mature neurons and progenitors should be used to validate this claim.

We agree with Reviewer 2 and have included additional markers of the dorsal forebrain and subsequent quantifications (NEW Fig. 1b-e, NEW Fig. S2b-e). Moreover, we increased the number of organoids and both days 40 and 80 are quantified (NEW Table S1). NEW Fig. 1b-c and NEW Fig. S2b-c now show extensive expression of FOXP1, EMX1, and SATB2 (at day 80) in addition to the previously shown PAX6, SOX2, TBR1, and CTIP2; and NEW Table S1 has been included to show the number of samples from each line for every experiment.

Moreover, the overall growth size of the organoids should also be monitored over time.

We measured the organoid size over time, but due to their strong tendency to fuse, determined this data was not an accurate metric of the actual growth rate of the organoids. BrdU labeling experiments in 2D neural precursor cell (NPC) cultures did not show any difference between PACS1^(+/+) and PACS1^(+/R203W) cells (shown below).

- The authors aimed to determine whether the p.R203W variant may cause a loss or gain of function. This is an important point worth being investigated. They found that KO PACS1 organoids developed normally at day 40. However, this time point may be too early (see comments above). So the claim seems to me not supported by strong experimental evidence. In fact, there is no detailed transcriptional analysis of these KO organoids and the level of PACS1 protein expression is also not assessed in the KO line. It seems to me that the CTIP2/SOX2 ratio in KO organoids is different from that of control organoids (Fig S2d) (but I am not sure how many independent analyses were done). Since this is such an important aspect (even in the title), more experiments should be done to validate the claim. Extended time points and more markers of mature neurons vs progenitors (not only one for neurons and one for progenitors) should be analyzed in KO organoids, before claiming that KO organoids are similar to healthy organoids.

We agree with Reviewer 2 that more data should have been gathered to definitively conclude the R203W is a gain-of-function variant. Though our data still suggests it is GOF, we chose to focus on characterizing deficits observed in the PACS1^(+/R203W) neurons, as this is the first study to look at the impact of this variant in the context of the developing nervous system using a knock-in model. Future studies will aim to mechanistically characterize the impact of the p.R203W variant with additional genotypes such as PACS1^(+/-) and PACS1^(R203W/R203W).

The authors speculate that this could even be an angle to develop treatment strategies. But nothing was experimentally tested. There are several things that the authors could do to prove this claim. For example, they could knock-down PACS1 in mutant organoids and see whether this can really work as a corrective action. These experiments would be required to fully prove that the p.R203W variant cause only a gain of function.

Some of the downstream function of PACS1 (e.g. related to the reported role in golgi-related function) should also be analyzed to prove that the KO does not produce loss of function effects. If these experiments are not included, then the claim about gain of function (even in the title) remain unsupported.

We thank Reviewer 2 for the helpful suggestions that would strongly support the GOF of the variant. We have been in the process with the PACS1 Research Foundation of obtaining ASOs, which would ideally target p.R203W-specific PACS1. Experiments were planned and all PACS1^(-/-) neuron lines were differentiated in order to test ASOs on multielectrode arrays. Unfortunately, we have been waiting for the ASOs for years and do not yet have access to them. We will therefore address this in a future paper with multiple additional lines including PACS1^(+/-) and PACS1^(R203W/R203W) to characterize the effects of the variant more thoroughly. For this study, we have chosen to focus on the differences between PACS1^(+/+) and PACS1^(+/R203W) neurons.

- The claim that “PACS1(+/R203W) organoids have a propensity towards GABAergic differentiation” would require more validations. One single marker for GABAergic neurons (Fig 3d-e) is not sufficient. Additional markers and maybe also different 2D differentiation experiments could be included to support this claim.

We agree with Reviewer 2 that more evidence should be provided to support this claim. While we started 2D differentiation experiments, new information provided by the day 88 organoid data reveals the GABAergic population is not maintained (NEW Fig. 2c; Fig S5c). We therefore shifted the focus of the paper as the network burst phenotype more likely underlies what is experienced by patients.

- The fact that the pathogenic variant leads to “higher density of GABAergic synapses” should be confirmed using other assays. The differences in Fig S6b-c also do not seem very strong. Functional assessments would be required, for example with calcium imaging, MEAs, or electrophysiology. The use of one pair of GABAergic synaptic markers (Fig 5) is not sufficient to prove functional changes.

We agree with Reviewer 2 and have now included both calcium imaging (NEW Fig. 4) and MEAs (NEW Fig. 5) which both suggest synaptic function changes in PACS1^(+/R203W) cells as suggested by the DEG analysis (NEW Fig. 3). However, the findings in this paper are likely not GABA-mediated, and so a less directed organoid protocol, MEA co-culture, or mouse model could help address this in a future study.

- The data related to ASD-specific gene network (Fig 6 and Fig S7) and specifically the supposedly enrichment of PACS1 with respect to synaptic function (Fig 6f) remain quite speculative in my view, given the lack of validations. Overall I find the data of Fig 6 interesting but very speculative and preliminary if not supported by more experimental evidence.

We agree with Reviewer 2 and have now omitted the WGCNA analysis from Figure 6 as the final focus of the paper is now on the functional data.

Additional points:

- Information in all the figures is missing with respect to which lines were used and how many independent experiments were performed. It is not sufficient to say “n=9 PACS1(+/R203W) organoids”. It is important to know which line (patient or control background) and whether the organoids come from the same line. In fact, validations should be carried out on multiple lines and with different independent experiments.

We thank Reviewer 2 for bringing this to our attention. We have included NEW Table S1 which clearly outlines how many samples of each line were used in the different experiments. Number of independent differentiations is listed in the figure legend. We agree with Reviewer 2 that validations should be carried out on multiple lines and with different independent experiments thus have now also included a clearer breakdown of major conclusions by cell line in the supplements (NEW Fig. S2d-e; S5; Fig. S7; Fig. S8b-d; Fig. S9a). Our strongest conclusion is the increase in network burst duration in PACS1^(+/R203W) neurons, results of which are shown by isogenic pair, differentiation, and experiment in NEW Fig. S9a.

- The pseudotime analysis in Fig 4 is in principle well done. I just find it less interesting and a bit redundant, since it arrives at the same conclusion (ZIC transcripts are altered) as we have seen already in Fig 2. Perhaps the authors can find a way to better merge these complementary analyses. In my opinion, some images (like Fig 4d) may be more appropriate for supplementary data.

R2 is correct, we understand this concern and recognize the redundant aspects. Since the GABAergic population was not abundant in organoids at day 88 (NEW Fig. 2c; Fig. S5c), we no longer included this pseudotime analysis in the resubmission. We did perform pseudotime analysis for day 88 organoids but did not find differences worth reporting between PACS1^(+/+) and PACS1^(+/R203W) cells.

- Fig 1d, 1f, 1g. It is not clear to me if these are data from control or mutant organoids. The profiles for both genotypes should be provided (with detailed description of which line and the number of replicates).

We apologize for the lack of clarity. Similar plots are shown in the resubmission (NEW Fig. 2a, b, and f). We think it is valuable to show the UMAP dimensionality reduction as a whole in addition to divided by sample. It is now stated in the figure legend what samples are being displayed in these plots. NEW Fig. S5 covers each sample individually in greater detail and NEW Table S1 again states which lines and how many were used.

- Fig 2a, Fig S4c. It is not clear how the up or down regulated genes were identified. In some cases, there are clearly no differences (see for example MALAT1 in IPCs and newborn neurons). But still these transcripts are included in the images. This is confusing.

We completely agree with Reviewer 2 that MALAT1 was an artifact, likely due to its very high base mean. Our DEG analysis has been completely restructured (see Methods; page 14) which has eliminated many of these concerning genes including MALAT1. Previous analysis was done with Seurat's FindMarkers function, with multiple pairwise DEG analyses in an effort to avoid these artifacts. This new analysis is done with the Monocle3 fit_models function and subsequent validation by a pseudobulk method with DESeq2 where each organoid is treated as an "n". We also have implemented other measures such as count minimums that each sample must meet for a gene to be considered instead of an average, which overall has given us a much smaller but very conservative DEG list.

- Fig 2c and Fig S3d. It is not clear if these are the patterns of mutant organoids and the one of control organoids.

We apologize for the lack of clarity. We now state in figure legends (NEW Fig. 2a, Fig. 2f, Fig. S4c) when all sequenced cells are included.

- Fig 2b. It should be better explained how these data were generated and what is their meaning.

We apologize for the lack of clarity. We agree with Reviewer 2 that we should have better explained Fig. 2b. We still see overlap of neuron DEGs with risk genes of other neurodevelopmental disorders, but we no longer included this analysis in the resubmission as we feel it does not add to the other results.

- It is written that figure legends for Fig 2a can be found in Fig S4c but I could not find these.

We apologize for confusion. In the previous Fig. 2a, we had abbreviated the legend for clarity by showing that the size of the dot in general indicated percent of cells expressing a feature, while color indicated average expression. The numbers for each size and color scale were unique to each graph and shown in previous Fig. S4c. We have omitted the dot plots in the resubmission and instead made a version of a volcano plot as seen in NEW Fig. 3b and Fig. S6, which more clearly displays top DEGs.

- The quantification analyses of the staining are not explained in the methods. Details regarding the number of fields per slice analyzed should also be included.

We now have included a more detailed explanation of how the images for immunohistochemistry analysis were gathered and quantified in the methods (page 12), including number of fields per slice analyzed (1). We apologize for initially omitting this information.

For synaptic marker quantification there is an explanation, but it is not entirely clear whether the colocalization with MAP2 was included, as this is mentioned in the text but not shown in the images.

We apologize for the lack of clarity. We followed what has been previously done in organoid literature (Samarasinghe *et al.*, *Nat. Neurosci.*, 2021, PMID: 34426698; Fig. 5b). MAP2 was included in our immunohistochemistry staining in order to be more aware of the surrounding neuron structure in the organoid section, but it was not used for colocalization. Dendritic spines – the site of many synaptic connections – are not necessarily MAP2+ and we did not want to eliminate these values. We now more clearly state that MAP2 was not used in the colocalization in the methods (page 15).

- The authors mention in Table S1F that they generated homozygous iPSCs, but there is no proofs or details for this line. I would suggest to eliminate this claim, if it is not supported by evidence.

We thank Reviewer 2 for bringing this to our attention. This line did not end up being included in the paper and so we have taken it out per suggestion of Reviewer 2.

- Additional transcriptomics datasets were included as shown in Fig S4a. However, it would be important to see that the expression pattern of PACS1 and of other key transcripts (like ZIC1, and GAD1) are similar between the original controls and these additional control organoids.

We thank Reviewer 2 for pointing this out as this was certainly a necessary control. The public data is no longer in the paper due to shift of focus to the day 88 timepoint. However, we would like to show that in our day 40 data, control ZIC1 and GAD2 expression is more similar to public data levels than to PACS1^(+/R203W) samples.

Reviewer #3 (Remarks to the Author):

In the current study, Rylaarsdam et al. set out to study the cellular and molecular changes associated with a recurrent mutation (R203W) in the PACS1 gene that is causal to the PACS1 syndrome. The authors used both patient and control iPSC lines, as well as isogenic iPSC lines generated by introducing or correcting the genetic variant. Organoids were generated using these cell lines for mechanistic insights. They performed single-cell RNA-seq at one selected time point and found changes in the inhibitory lineage cells associated with the pathogenic mutation. However, some major concerns remain.

1. The study used both male and female iPSC lines for organoid formation. However, it seemed that data analyses and discussions were solely based on the different genotypes. There was no information provided in terms of the similarity/reproducibility among organoids from different genetic backgrounds with or without the mutation, let alone the batch effects between different experiments. Also, it was not clear if any sex-dependent differences were noticed in both control and mutant conditions.

Reviewer 3 points out a potentially confounding factor that was not discussed in the initial submission. We apologize for the lack of clarity and have now included NEW Table S1 which clearly outlines how many samples of each line were used in the different experiments. Number of independent differentiations is listed in the figure legends and have now also included a clearer breakdown of major conclusions by cell line in the supplements (NEW Fig. S2d-e; Fig. S5; Fig. S7; Fig. S8b-d; Fig. S9a). Moreover, our new day 88 single-cell RNA sequencing data includes samples one female and one male line for each time point, which will hopefully alleviate sex-specific effects. Our strongest finding – that PACS1^(+/R203W) neurons have an increased network burst duration – is consistent across sex, differentiations, experiments, and isogenic pairs. We now show these comparisons more clearly in NEW Fig. S9a. This is in concordance with PACS1 syndrome presentation, which is not different between males and females. However, Reviewer 3 makes a good point that the sex of the iPSC-derived line donors is a potentially critical confounding factor and will be factored into future differential expression analysis schemes when possible and appropriate.

2. In Figure 1, the TSNE plot shows that PACS1 is expressed in different clusters with variable expression levels, it is good to show its level across clusters as for other marker genes in 1f.

We thank Reviewer 3 for pointing out that this important information was missing. Expression of PACS1 for each cell type and sample can now be found in NEW Fig. S5e.

Does the variant have any impact on the RNA or Protein stability? Is PACS1 expression level consistent across the different genetic backgrounds in iPSC cells?

We thank Reviewer 3 for pointing out this valuable information was not properly addressed in the initial submission. We do not see a difference in levels of PACS1 RNA between either genotype of genetic backgrounds in single cell data (NEW Fig. S5e) or PACS1 protein in western blots (NEW Fig. 1f-g).

3. While the method used to make organoids preferentially generate excitatory neurons, it by no means excludes the existence of inhibitory neurons in the same organoids. This is consistent with the expression of the inhibitory neuron marker gene in the wild-type condition (figures 3 and 4). It is not clear why the authors considered cells that express GAD1, SLC32A1, DLX 1, etc to be excitatory neurons.

We understand Reviewer 3's concern that cells expressing GAD1, SLC32A1, DLX 1, were called excitatory neurons. The average transcriptome of these cells was determined to most highly correlate to excitatory neurons overall with our bioinformatic comparison to a reference human fetal transcriptome (Nowakowski *et al.*, *Science*, 2017. PMID: 29217575). However, upon further investigation, we agree with Reviewer 3 and have now changed our cell type annotation to include these cells on a group called immature GABAergic neurons (NEW Fig. 2a). An additional cluster which weakly expresses both GABAergic and

glutamatergic markers is now called “immature neurons” as it seems to be a mix (NEW Fig. 2a; NEW Fig. S4c). We have also further distinguished our glutamatergic neuron clusters as seen in NEW Fig. 2a.

Similarly, it is puzzling that it was described that ectopic GABAergic expression was found in excitatory cells (Figure 4a). PSD95 and GRIN2B are not inhibitory neuron marker genes. Both excitatory and inhibitory neurons express receptors for glutamine and GABA. It is possible that more inhibitory neurons are generated in the mutant organoids. Did the authors detect more Nkx2.1, LHX6, SP8, or other genes that are critical for interneuron development? It would also be critical to determine if this phenotype is temporary and if the differences between the mutant and control organoids could sustain into more mature stages.

We agree with Reviewer 3 that it was critical to determine if the GABAergic phenotype is temporary. Therefore, we have added day 88 organoids to our analysis. It was clear from this new data that Reviewer 3 is correct as at day 88, much less of these GABAergic cells are seen (NEW Fig. 2a, Fig. S5c), suggesting that this is a transient population. Thus, we have now instead shifted our focus to the mature glutamatergic population. In addition, we have now defined a population as immature GABAergic neurons, which does not express *NKX2.1* or *LHX6* but does lowly express *SP8*. This could suggest an LGE/CGE identity but is no longer the focus of this study.

4. What are the genes in different modules in Figure 4d?

We apologize for not including which genes were present in the different modules and were going to attach it as a supplement in the resubmission. However, we have since decided to no longer include the module analysis.

5. When analyzing the synaptic proteins, the quantification method is troublesome. The authors used puncta/soma to measure the density of synapses, which did not consider the number of neurons of all the cells included or the complexity of neurons, e.g., the number and length of dendrites.

We understand Reviewer 3’s concern. We followed what has been previously done in organoid literature (Samarasinghe *et al.*, *Nat. Neurosci.*, 2021, PMID: 34426698; Fig. 5b) and confirmed neuronal identity of the image field with MAP2. Unfortunately, we are not able to assess the number and length of dendrites with this method as a cell would have to be perfectly in-plane with the section and it is impossible to detangle the web of MAP2 in an organoid to trace cells. A viral sparse labelling approach would be more appropriate.

Some minor points:

1. The authors mentioned that cell lines that were engineered using CRISPR had been confirmed using WES, which was undoubtedly rigorous and thorough. Were any off-target changes detected?

We thank Reviewer 3 for pointing out this missing information have included a more descriptive protocol for off-target detection in our methods (page 11). The bam files from whole genome or exome sequencing were imported to IGV (v2.8.10) and manually checked for off-target effects. Of the top 100 off-target effects predicted by the IDT CRISPR design tool, only two other sites were exonic (MVP chr16:-29844665; PACS2 chr14:-105355130) and did not have any mismatched base pairs in any of the three isogenic sets of lines. The top 5 off-target sites predicted by CHOPCHOP (v3) were all non-coding, but analysis of the CRISPR R203W and CRISPR A1 lines - which received whole genome sequencing - did not reveal any unintended edits. An image of the three coding and top five non-coding sites is attached here.

PACS1 p.R203W chr11:-66211212

MVP chr16:-29844665

PACS2 chr14:-105355130

chr1:155964574

chr6:127445345

chr3:43102718

chr1:231182256

chr17:15316928

2. *PAX6 staining in supplementary figure 2b: showed mostly cytoplasmic signal without clear cellular boundary. Is this an artifact from the staining?*

We agree that this image did not show clear PAX6 signal. R3 is correct that this was most likely an artifact from the staining. The PAX6 immunos have been replaced in NEW Fig. S2c, which shows PAX6+ rosettes at day 40 and sporadic expression where rosettes used to be in day 80 organoids.

3. *Gm03651 from Coriell was labeled as both a male and a female line.*

We sincerely thank Reviewer 3 for pointing out this mistake; GM03651 is a female line. We have now corrected it in NEW Table S1.

4. *In Figure 2a, there is no legend for the color or dot size.*

We apologize for the confusion. In an attempt to enhance the clarity of the figure, we had put a more general legend in Fig. 2a with a numerical legend for each plot in Fig. S4c. We now display top DEGs in a volcano-style plot in NEW Fig. 3b as we believe this more clearly illustrates the data.

5. *Supplemental figure 6: there are vesicle transporter genes in the list, these are not receptors.*

We thank Reviewer 3 for pointing out this inaccurate terminology. This component of the analysis has not been included in the revised manuscript and we will be more careful in the future.

REVIEWER COMMENTS

Reviewer #1 (Remarks to the Author):

This revised manuscript from Rylaarsdam and collaborators has improved greatly since its first version. The authors have addressed many all the concerns raised by the reviewers.

The authors have extended the breadth of their work by including an additional later timepoint during organoid development in which the phenotype is analyzed both at the molecular and functional levels. This new data has shown that the original phenotype described was either transient or not fully present at all and has shifted the manuscript to the analysis of differentially expressed synaptic proteins and differences in firing bursts.

They have improved the differential expression analysis, making it more solid now. I would love to see validation of some of these differentially expressed genes (specially the top candidates) in the tissue per immunostaining or in situ hybridization.

Importantly, they identified functional deficits in the mutant organoids and derived excitatory neurons, that would operate downstream of the differentially expressed ion channels and synaptic proteins identified. I understand that this is the first work describing functional and molecular defects in a PACS1 syndrome model, which makes this discovery exciting.

One of my additional concerns was about regional identity of the organoids. This was nicely addressed with immunostainings and the authors also provided additional information in their response to clarify some of my doubts. It would be useful to have these genes' expression also shown at the single cell level (added to figure s4 for example).

Finally, while the full reason behind the phenotype itself is not very clear yet and will need more work in the future, the authors suggest a first mechanistic insight. I believe this will be a meaningful and important work that informs future research on PACS1 syndrome. The work is carefully done and analyzed and sheds light on the mechanisms behind the condition.

Besides my few comments above, I have a few more suggestions that the authors can likely easily address and would improve clarity of the manuscript.

- 1- Bar plots comparing cell type proportions for stainings and single cell data are difficult to distinguish between control and affected conditions. Authors should consider a graphical way of easily distinguishing bars, such as shading, lighter colors, edge, etc. (1d,e; 2c; S2d,e; S5b,c)
- 2- The use of spatial in this quote is misleading and should be better defined. "PACS1 is enriched in neurons but broadly expressed across all cell groups (Fig. 2f, shown by line in Fig. S5e) while a spatially restricted progression of proliferative to mature neuron markers is observed (Fig. 2e-f; Fig. S4c)."

Reviewer #2 (Remarks to the Author):

The authors performed several new experiments and their revised manuscript is now greatly improved. However, in my opinion there are still issues that require attention.

- I understand that the authors now want to focus on the functional results. However, I find it quite surprising that in the abstract there is no mention of the organoid data and the lack of neurodevelopmental defects. I find these findings quite important as they suggest that this mutation does not seem to be sufficient to cause neurodevelopmental alterations. I would suggest the authors to include this in the abstract and discussion. Also I would be curious to know if this fits with the patient data. Are there any reports of microcephaly (or macrocephaly) in the patients? In fact, it is very nice that the authors measured NPC proliferation with BRDU labeling. I do not understand why these important data are only shown in the rebuttal letter and not added in the manuscript.

- The presentation of the lines is still confusing in my opinion. It is good to have the Table S1 but this is not sufficient. For example, what are the numbers indicating? The legend says that they indicate the number of organoids. But out of how many independent experiments? And MEA were done in NGN2 neurons and not organoids. So this is confusing. Also I do not know which data refer to which figure. I do not understand why the authors cannot simply mention in each figure legend something like this "images for Control 1 and A1 are shown" or "data shown are for PACS1 (+/+) (Control 2 and CRISPR R203W) and PACS (+/R203W) (A1 and A2)". This would make all the images so much easier to read and understand.
- In the description of the lines, the text is still not clear to me. I think it can be made much easier to understand if the author could simplify the text like this for example: "Three control iPSC lines were used. Control 1 was... Control 2 was... Control 3 was... Two patient iPSC lines were used. Affected 1 (A1) was... Affected 2 (A2) was... CRISPR/Cas9 was used to introduce the mutation in the iPSC line Control 1 to generate the line CRISPR R203W, and to correct the mutation in the iPSC lines A2 and A3 to generate the lines CRISPR A1 and CRISPR A2". In this way all lines are clearly indicated and now the authors can easily indicate the lines in the text using these names and mention them in all figure legends.
- In the characterization of the lines, it is great to hear that karyotype is available for the purchased iPSC lines (Control 3, A1 and A2). In the rebuttal letter, the authors provide also the karyotype of Control 1, and the tri-lineage differentiation of two lines (I think Control 3 and A2). But how about the karyotype of Control 2 and the differentiation potential of both Control 1 and Control 2? Also, these data should be seen by all readers not only by the reviewers. More importantly, we do not know the karyotype of the three edited lines (CRISPR R203W, CRISPR A1, and CRISPR A2). It is great to see that the authors performed WGS or WES on the lines (although it is not clear to me which analysis on which line). However, the state of the karyotype is not mentioned. Please provide these data to clarify that the edited lines do not harbor chromosomal abnormalities. Moreover, I do not understand why the lack of off-target changes are shown in the rebuttal letter but are not included in the manuscript.
- FigS1. The authors could make an effort to better explain the image in S1B. Which one is the patient mutation, which ones are the silent mutations introduced? Please either mention this in the text or indicate in the figure. And which iPSC lines are shown in S1C? This is not indicated in Table S1. This is why I think that mentioning the lines in the legend is simply easier.
- It is great that the authors added a WB image, which according to Table S1 was done with only one organoid per genotype. Is the quantification based on only this image, or was the data replicated in independent experiments?
- I wonder what the author think it is the possible mechanism of action of the disease. Based on this single WB, the authors conclude that protein amount of PACS1 is not altered by the mutation. But then what could the mutation cause? A change in the folding of the protein (could be predicted by alpha fold)? Or in its function? It would be nice if the authors could speculate on any of this, since it is not clear to me what this lack of protein expression changes mean for the disease pathogenesis.
- I think that it is interesting to see that the WB fits with the transcriptional data shown in Fig S5e. There, it is clear that PACS1 is not differently expressed in the mutant organoids. Interesting PACS1 expression is increased in the later time point. I think that is quite relevant for the disease pathogenesis and worth highlighting it in the main text. Do the authors know if NGN2 neurons also express PACS1 and maintain it at similar level regardless of the mutation? This would be important to know, as NGN2 neurons are now taken as a model to investigate PACS1 defects.
- Please indicate in the main text the abbreviations used (e.g. DEG, GOF). Also please indicate in the legends or main text what the various abbreviation mean (e.g. cRG (S), tRG, BCU and so on).
- The DE analysis highlighted 5 transcripts related to synaptic function CNTNAP4, GPRIN3, CACNA2D3, GABRG3, and TRPC5 (Fig 3e). This is great, but the connection with the downstream experiments is less clear in my opinion. Are these transcripts related to glutamatergic or GABAergic synapses?
- What is the difference between Fig4b-c and S8b-c? Why S8c is not significant but 4c is significant? What are the dots indicating in these figures? If I understood Table S1, the authors performed 3 independent experiments for PSD95 for Fig4b and S8b. But then how is it possible that only 1 organoid was visualized for Control 3? Are the different differentiations done with different lines? Or was it one organoid per each differentiation?
- It would be helpful if the authors could add some interpretation to the lack of differences following the application of a GABA antagonist (Fig 4h). So there is a change in GABA synapses but

not in GABA transmission? What were the authors expecting? That after the exposure to the antagonist the patient organoids would keep firing more? But maybe the antagonist is simply too strong to allow any residual activity? Maybe there are other mild antagonists that can be used? Or maybe a GABA agonist (e.g. baclofen) can be used, and this can be used to show increase activity in mutant cells upon stimulation? If the authors do not want to perform these additional experiments, at least they should acknowledge the current limitations (only one antagonist) and put forward some interpretations. And if the GABA synapses are more present, then it may be recommended that patients should not take GABA agonists? Perhaps that are reports in the literature about baclofen and intellectual disabilities? Alternatively, if the GABA transmission is not relevant, which other alternative explanations could there be? What does it mean that a "mechanism independent of GABA-A receptors was responsible for the change in spike frequency"? What could such mechanism be? It would be interesting to hear potential speculations, also given the fact that the authors saw changes in GABA (both via staining in and via RNAseq and IPA) and not in glutamatergic synapses, but then they go ahead and use glutamatergic NGN2 neurons for following experiments. Also there are no experiments with glutamatergic agonists/antagonists, so we do not know whether glutamatergic signaling can be manipulated differently in mutant cells.

- Several pieces of data suggest potential GABA dysfunction, but then since there is no difference in the calcium response in mutant organoids compared to controls after the application of the GABA antagonist, the authors disregard all these evidence and in the discussion put forward only the idea of glutamatergic defects. I am not sure I agree with this strategy. I think that the evidence regarding GABA should also be mentioned. A recent paper from Arlotta's lab suggested that defective development of GABA neurons could underlie the pathogenesis of ASD (PMID: 35110736). Perhaps this is worth mention in the discussion, especially since several of the findings of Rylaarsdam et al also suggest a dysregulation of GABA neurons. I also do not understand why the authors said that the differences in GABA is not maintained at later time points. The synaptic marker staining was done at a late time point (day 79 according to the figure legend of fig 4c).
- FigS9a. I do not understand why in some cases the curves are shorter or not existent. Please clarify.
- Based on the MEA profile of NGN2 neurons, the authors assume that the defects are only in glutamatergic neurons. However maybe similar changes might be seen using GABA neurons. This is in fact quite likely and cannot be dismissed by the authors. So we do not know if GABA neurons are affected or not. Moreover, there are systems that can be used to easily generate GABA neurons (PMID: 28504679) and study their relation to glutamatergic neurons (<https://doi.org/10.1101/2020.05.07.082453>), so perhaps this can be considered for future studies or to be added in the discussion.
- In the abstract, the authors conclude that their work suggests "aberrant regulation of ionic flux". I am not sure that this can be easily stated. They did not perform any analysis of channel activity (only general calcium imaging). I would modify those bold statements. Also, in the abstract they mention that their findings "underlies the neurological phenotypes experienced by patients". Is this true? In the discussion the authors mention that the EEG reports from patients are "rare and variable". So are their changes similar and consistent to what is reported in patients or not? I would suggest the authors to be more specific with respect to their interpretation and be more careful when describing the implications in the abstract.

Reviewer #3 (Remarks to the Author):

The authors were responsive to most of the comments. However, after incorporating the new dataset, the revised manuscript switched the focus/conclusion. Therefore, some further questions were raised.

- In Figure 2d, what is the color legend for "genotype" and "day"? Can the authors discuss/speculate why the cells of different genotypes do not segregate even within the same cell types.
- The revised manuscript adds a new dataset consisting of Day 88 organoids. First, however, it needs to be clarified how and if any data integration is performed. As we know, scRNAseq from different experiments can be highly variable due to differences in sample preparation and

sequencing technologies. Therefore, data integration is a critical step that involves combining scRNA-seq data from multiple experiments or using different cell lines to generate a unified dataset for downstream analyses.

- In figure 3b, what is the base of the log value? What is the cut-off?
- How frequently did vGAT signal appear? Is it rare? The authors mentioned in the rebuttal letter, "the GABAergic population was not maintained at day 88".
- The authors stated that "At both day 40 and 80, spontaneous TTX-susceptible calcium spikes were observed in PACS1(+ / +) and PACS1(+ / R203W) organoids". However, throughout the manuscript, no data was provided to demonstrate the TTX susceptibility.
- How similar are the excitatory neurons generated in 2D versus 3D? While dual SMAD inhibitors and WNT inhibitors were briefly included in the medium, it would be reasonable to confirm the cortical identity of the neurons generated in the 2D system, especially considering the induction of markers of sensory neurons and periphery nerve cells observed in Ngn2 neurons. Are the DEGs identified using the organoid system also identified in the 2D neurons? Notably, the authors speculated that "DEGs such as KCNMB2, TRPC5, or CACNA2D3 could certainly underlie this phenotype" (in the 2D culture). Were the gene expression differences confirmed using qPCR or other methods? Spike frequency no longer shows a difference between the two genotypes. Is this due to the differences in the excitatory neurons generated? Figure 5 i-o, what does each dot represent?
- Table S1 is now provided to show which cell lines were used in different experiments. I appreciate that the authors used all data collected from various lines for data analyses. However, it needs to be clarified how the data generated using different lines were analyzed, integrated, and interpreted. Were there any dropouts? It would be great to present data collected using different lines in the figures. It is both informative and essential to show the variability of experiment results.
- While MEA provides information on overall network activity, it doesn't offer insights into the underlying mechanisms. For example, the phenotypes could be due to intrinsic deficits of the mutant neurons as indicated by the changed channel protein expression. Patch-clamp would be more suitable and informative to probe the mechanism.

Re: NCOMMS-22-18984A-Z

We would like to thank the editor and reviewers for their time and insightful comments. We have put forth great effort to address concerns and present a substantially improved manuscript, which we hope is now suitable for publication in *Nature Communications*. Our point-by-point responses to italicized reviewer comments can be found below in blue regular font. All text changes and additions made to the manuscript are also highlighted in blue font throughout both the main manuscript and supplementary documents.

Reviewer #1 (Remarks to the Author):

This revised manuscript from Rylaarsdam and collaborators has improved greatly since its first version. The authors have addressed many of the concerns raised by the reviewers. The authors have extended the breadth of their work by including an additional later timepoint during organoid development in which the phenotype is analyzed both at the molecular and functional levels. This new data has shown that the original phenotype described was either transient or not fully present at all and has shifted the manuscript to the analysis of differentially expressed synaptic proteins and differences in firing bursts.

They have improved the differential expression analysis, making it more solid now. I would love to see validation of some of these differentially expressed genes (specially the top candidates) in the tissue per immunostaining or in situ hybridization.

We thank Reviewer 1 for the suggestion and have now included immunostaining for top upregulated DEGs in PACS1(+R203W) organoids identified with single-cell RNA sequencing in Fig. S10.

Importantly, they identified functional deficits in the mutant organoids and derived excitatory neurons, that would operate downstream of the differentially expressed ion channels and synaptic proteins identified. I understand that this is the first work describing functional and molecular defects in a PACS1 syndrome model, which makes this discovery exciting. One of my additional concerns was about regional identity of the organoids. This was nicely addressed with immunostainings and the authors also provided additional information in their response to clarify some of my doubts. It would be useful to have these genes' expression also shown at the single cell level (added to figure s4 for example).

We agree with Reviewer 1 that it would be useful to show the RNA expression of genes used for immunohistochemistry validation. Expression of genes encoding PAX6, SOX2, FOXP1, EMX1, CTIP2, and SATB2 are now included in feature plots on Fig. S7.

Finally, while the full reason behind the phenotype itself is not very clear yet and will need more work in the future, the authors suggest a first mechanistic insight. I believe this will be a meaningful and important work that informs future research on PACS1 syndrome. The work is carefully done and analyzed and sheds light on the mechanisms behind the condition.

Besides my few comments above, I have a few more suggestions that the authors can likely easily address and would improve clarity of the manuscript.

1- Bar plots comparing cell type proportions for stainings and single cell data are difficult to distinguish between control and affected conditions. Authors should consider a graphical way of easily distinguishing bars, such as shading, lighter colors, edge, etc. (1d,e; 2c; S2d,e; S5b,c)

We agree that genotypes were difficult to distinguish in the bar graphs on Figs 1d-e, 2c, S2d-e (now S4d-e), and S5b,c (now S8b-c). We have now included a subtle pattern over the bars of PACS1^(+/R203W) groups, which differentiates the genotypes while still clearly illustrating the data.

2- The use of spatial in this quote is misleading and should be better defined. “PACS1 is enriched in neurons but broadly expressed across all cell groups (Fig. 2f, shown by line in Fig. S5e) while a spatially restricted progression of proliferative to mature neuron markers is observed (Fig. 2e-f; Fig. S4c).”

We thank Reviewer 1 for pointing out how our use of the term spatial - which was meant in the context of the UMAP dimensionality reduction plot - was not clear. We now mention that the 16 cell types identified with single-cell RNA sequencing express “distinct transcriptomic signatures,” but have removed the sentence with the misleading spatial term.

Reviewer #2 (Remarks to the Author):

The authors performed several new experiments and their revised manuscript is now greatly improved. However, in my opinion there are still issues that require attention.

I understand that the authors now want to focus on the functional results. However, I find it quite surprising that in the abstract there is no mention of the organoid data and the lack of neurodevelopmental defects. I find these findings quite important as they suggest that this mutation does not seem to be sufficient to cause neurodevelopmental alterations. I would suggest the authors include this in the abstract and discussion. Also I would be curious to know if this fits with the patient data. Are there any reports of microcephaly (or macrocephaly) in the patients? In fact, it is very nice that the authors measured NPC proliferation with BRDU labeling. I do not understand why these important data are only shown in the rebuttal letter and not added in the manuscript.

We agree that our observation of no changes in proliferation or cell death of neural precursor cells in the presence of the variant is an important and disease-relevant finding. At the suggestion of Reviewer 2, we have now incorporated our findings in neural precursor cells throughout the abstract and manuscript. For example, we now include in the second paragraph of the results: “This hypothesis was further supported by our initial analyses using neural precursor models, which did not show a difference in rates of proliferation or apoptosis (Fig. S5). All this suggests that p.R203W PACS1 does not affect early proliferation and specification dynamics during cortical development, which is in line with no consistent clinical reports of microcephaly or macrocephaly in PACS1 syndrome patients.” In addition, we mention in the discussion: “We have shown that the variant seems to have a disproportionate impact on more mature neuron populations, providing hope that a therapeutic intervention could be effective after diagnosis.” The authors thank Reviewer 2 for bringing to our attention the importance of these findings and encouraging us to incorporate them into the manuscript.

The presentation of the lines is still confusing in my opinion. It is good to have the Table S1 but this is not sufficient. For example, what are the numbers indicating? The legend says that they indicate the number of organoids. But out of how many independent experiments? And MEA were done in NGN2 neurons and not

organoids. So this is confusing. Also I do not know which data refer to which figure. I do not understand why the authors cannot simply mention in each figure legend something like this “images for Control 1 and A1 are shown” or “data shown are for PACS1 (+/+) (Control 2 and CRISPR R203W) and PACS (+/R203W) (A1 and A2)”. This would make all the images so much easier to read and understand.

We apologize that Table S1 was insufficient. At the suggestion of Reviewer 2, we have now listed how many samples from each cell line were used in all figure legends. We have also included more information in Table S1 in an effort to be as clear as possible.

In the description of the lines, the text is still not clear to me. I think it can be made much easier to understand if the author could simplify the text like this for example: “Three control iPSC lines were used. Control 1 was... Control 2 was... Control 3 was... Two patient iPSC lines were used. Affected 1 (A1) was... Affected 2 (A2) was... CRISPR/Cas9 was used to introduce the mutation in the iPSC line Control 1 to generate the line CRISPR R203W, and to correct the mutation in the iPSC lines A2 and A3 to generate the lines CRISPR A1 and CRISPR A2”. In this way all lines are clearly indicated and now the authors can easily indicate the lines in the text using these names and mention them in all figure legends.

The authors again apologize for lack of clarity and have changed the introduction of the cell lines to match the structure proposed by Reviewer 2: “To investigate how the p.R203W variant in PACS1 impacts developing neural cell types, we generated dorsal cortical organoids from induced pluripotent stem cell (iPSC) lines derived from three control (PACS1^(+/+)) and two affected (PACS1^(+/R203W)) individuals: Control 1 (C1) was derived from a healthy female, C2 from a healthy male, and C3 from an unaffected mother of a PACS1 syndrome patient; Affected 1 (A1) was derived from a male with PACS1 syndrome, and A2 from the affected daughter of the C3 donor. In addition, we utilized CRISPR/Cas9 gene editing technology to introduce the heterozygous variant which leads to p.R203W into the C1 background (CRISPR R203W) and corrected it in the two patient lines (CRISPR A1 and CRISPR A2) for a total of three isogenic pairs (Fig. 1a; see Fig. S1-S3 for line characterization; see Table S1 for breakdown of lines used in all experiments).”

In the characterization of the lines, it is great to hear that karyotype is available for the purchased iPSC lines (Control 3, A1 and A2). In the rebuttal letter, the authors also provide the karyotype of Control 1, and the tri-lineage differentiation of two lines (I think Control 3 and A2). But how about the karyotype of Control 2 and the differentiation potential of both Control 1 and Control 2? Also, these data should be seen by all readers not only by the reviewers. More importantly, we do not know the karyotype of the three edited lines (CRISPR R203W, CRISPR A1, and CRISPR A2). It is great to see that the authors performed WGS or WES on the lines (although it is not clear to me which analysis on which line). However, the state of the karyotype is not mentioned. Please provide these data to clarify that the edited lines do not harbor chromosomal abnormalities. Moreover, I do not understand why the lack of off-target changes are shown in the rebuttal letter but are not included in the manuscript.

We apologize for the missing information. Karyotypes and differentiation potential for all lines are now provided in Fig. S2 and S3b-d, demonstrating no chromosomal abnormalities nor differentiation defects for parent or CRISPR lines. We also mention the results in the methods: “Karyotype of all lines was done by WiCell and determined normal (Fig. S2).” “Pluripotency tests of iPSC lines using the STEMdiff™ Trilineage Differentiation Kit (Stem Cell Technologies, 05230) according to manufacturer instructions demonstrated differentiation capacity of all three germ layers (Fig. S3).” Off-target analysis is now incorporated in Fig. S1.

FigS1. The authors could make an effort to better explain the image in S1B. Which one is the patient mutation, which ones are the silent mutations introduced? Please either mention this in the text or indicate in the figure. And which iPSC lines are shown in S1C? This is not indicated in Table S1. This is why I think that mentioning the lines in the legend is simply easier.

We thank Reviewer 2 for encouraging us to clarify the figure. The pluripotency immunostaining shown in previous Fig. S1C is now expanded in Fig. S3a where all cell lines are included with clear labeling of the cell line of origin. In addition, we have annotated Fig. S1b more clearly and described it more thoroughly in the figure legend: “Sequencing validates introduction of the heterozygous c.607C>T variant in the CRISPR R203W line - also observed in patient lines A1 and A2 (boxed in yellow) - and successful correction of the patient variant in CRISPR A1 and CRISPR A2 lines. In CRISPR-edited lines, the two downstream additional variants (boxed in blue) are the silent mutations. The guide target region is boxed in red with the PAM site towards the 5’ end.” These annotations are also labeled on Fig. S1b in addition to the descriptions in the legend.

It is great that the authors added a WB image, which according to Table S1 was done with only one organoid per genotype. Is the quantification based on only this image, or was the data replicated in independent experiments?

The WB in Fig. 1f-g was done with four organoids per genotype, but since our submission we have added more data points from independent experiments. Fig. 1f-g now contains data and quantifications from n = 7 PACS1^(+/+) organoids (two C1, three C2, and two CRISPR A1) and n = 8 PACS1^(+/R203W) organoids (four CRISPR R203W and four A1) from two independent differentiations.

I wonder what the authors think is the possible mechanism of action of the disease. Based on this single WB, the authors conclude that the protein amount of PACS1 is not altered by the mutation. But then what could the mutation cause? A change in the folding of the protein (could be predicted by alpha fold)? Or in its function? It would be nice if the authors could speculate on any of this, since it is not clear to me what this lack of protein expression changes mean for the disease pathogenesis.

Reviewer 2 brings up a good point that much is left to understand regarding exactly how the variant impacts PACS1 function. At the time we began this research, ours was the very first study to investigate the impact of the p.R203W variant in the developing brain, so much is left to be explored. We expand on this more in the discussion at the suggestion of Reviewer 2: “We did not find the variant impacted PACS1 abundance on the RNA or protein level, so it is likely that some kinetic or steric property could be altered by the introduction of the variant. This hypothesis is supported by the observation that the c.607C>T substitution results in a positive arginine being replaced by a large, nonpolar tryptophan within the Furin Binding Region of PACS1, which *in silico* modeling with AlphaFold predicts would strengthen its interaction with targets. Future studies will involve crystallography to try and resolve the structure of the FBR in the mutant and control PACS1, immunoprecipitation assays in neurons to establish if binding partners are lost or acquired upon introduction of the variant, and immunohistochemistry analysis of PACS1 and its targets to assess how the variant impacts subcellular localization in order to better inform therapeutic development.” We hope this now properly acknowledges how the variant could impact protein function independent of stability.

I think that it is interesting to see that the WB fits with the transcriptional data shown in Fig S5e. There, it is clear that PACS1 is not differently expressed in the mutant organoids. Interesting PACS1 expression is increased in the later time point. I think that is quite relevant for the disease pathogenesis and worth highlighting it in the main text. Do the authors know if NGN2 neurons also express PACS1 and maintain it at a

similar level regardless of the mutation? This would be important to know, as *NGN2* neurons are now taken as a model to investigate *PACS1* defects.

We agree with Reviewer 2 that it is very important to validate neurons generated with the *NEUROG2* overexpression method strongly express *PACS1*. Our data increasingly suggests that *PACS1* expression is highest in more mature neurons, which also display the strongest disease phenotypes compared to immature neural cells. Our preliminary experiments for future work demonstrate robust expression of *PACS1* in *NEUROG2* neurons at both the RNA (see **a** below) and protein (**b**) level. This supports *NEUROG2*-derived neurons are an appropriate model to investigate *PACS1*-mediated mechanisms of disease.

We certainly agree with Reviewer 2 that the observation mature neurons are more impacted by the p.R203W variant is a critical finding to understand from a therapeutic perspective, which was not sufficiently highlighted in the previous manuscript. This is now discussed more here: “There were no differences in ratios of these markers between genotypes (Fig. 1d-e, shown by line in Fig. S4d-e) suggesting that the p.R203W variant did not affect organoid gross cytoarchitectural organization. This hypothesis was further supported by our initial analyses using neural precursor models, which did not show a difference in rates of proliferation or apoptosis (Fig. S5). All this suggests that p.R203W *PACS1* does not affect early proliferation and specification dynamics during cortical development, which is in line with no consistent clinical reports of microcephaly or macrocephaly in *PACS1* syndrome patients.” We also mention in the discussion: “We have shown that the variant seems to have a disproportionate impact on more mature neuron populations, providing hope that a therapeutic intervention could be effective after diagnosis.”

Please indicate in the main text the abbreviations used (e.g. DEG, GOF). Also please indicate in the legends or main text what the various abbreviation mean (e.g. cRG (S), tRG, BCU and so on).

We apologize for using abbreviations without defining them properly. The authors have now made sure to expand all abbreviations prior to their use.

The DE analysis highlighted 5 transcripts related to synaptic function *CNTNAP4*, *GPRIN3*, *CACNA2D3*, *GABRG3*, and *TRPC5* (Fig 3e). This is great, but the connection with the downstream experiments is less clear in my opinion. Are these transcripts related to glutamatergic or GABAergic synapses?

We thank Reviewer 2 for pointing out that this connection was not very clear. The purpose for including these DEGs was two-fold: to provide confidence that our DEG analysis produced real results by showing top DEGs have a visually apparent difference in expression levels in feature plots; and to highlight DEGs particularly relevant to synaptic signaling. We have now included an expanded description of each gene’s

function in order to make this connection clearer: “These enrichments resulted from aberrantly expressed transcripts such as *CNTNAP4*, which encodes a presynaptic neuexin involved in GABAergic transmission; *GPRIN3*, which encodes a GPCR involved in neuron projection development; *CACNA2D3*, which encodes a voltage-dependent calcium channel subunit; *GABRG3*, which encodes a GABA receptor type A subunit; and *TRPC5*, which encodes a CNS-enriched Ca²⁺-permeable cation channel. The expression levels of these genes are visually different between genotypes in feature plots (Fig. 3f-g; shown by line in Fig. S12), providing support for the validity of GO and IPA predictions.”

What is the difference between Fig4b-c and S8b-c? Why S8c is not significant but 4c is significant? What are the dots indicating in these figures?

The authors apologize for the confusion caused by Fig4b-c and S8b-c. They are the same figure but with dots colored by cell line in the supplements. This was not included in the main figure because the authors felt the extra information was visually cluttering and detracted from the main message of the figure. S8c is also significant but this was not indicated in the supplementary figure; we have changed this in NEW Fig. S13a. We thank Reviewer 2 for pointing out this inconsistency.

If I understood Table S1, the authors performed 3 independent experiments for PSD95 for Fig4b and S8b. But then how is it possible that only 1 organoid was visualized for Control 3? Are the different differentiations done with different lines? Or was it one organoid per each differentiation?

Regarding Table S1: the same cell lines were not necessarily used for each independent experiment. For example, in glutamatergic synapse staining experiments, only one Control 3 organoid section from one differentiation was used, but total data was collected over the course of three staining experiments using organoids from three differentiations. We have added a note in the table legend to make this clearer: “Please note that independent differentiations indicates the total number of differentiations present across all samples in a given experiment, but does not necessarily indicate the experiment was repeated that many number of times with the exact same lines.”

It would be helpful if the authors could add some interpretation to the lack of differences following the application of a GABA antagonist (Fig 4h). So there is a change in GABA synapses but not in GABA transmission? What were the authors expecting? That after the exposure to the antagonist the patient organoids would keep firing more? But maybe the antagonist is simply too strong to allow any residual activity? Maybe there are other mild antagonists that can be used? Or maybe a GABA agonist (e.g. baclofen) can be used, and this can be used to show increased activity in mutant cells upon stimulation? If the authors do not want to perform these additional experiments, at least they should acknowledge the current limitations (only one antagonist) and put forward some interpretations. And if the GABA synapses are more present, then it may be recommended that patients should not take GABA agonists? Perhaps that are reports in the literature about baclofen and intellectual disabilities? Alternatively, if the GABA transmission is not relevant, which other alternative explanations could there be? What does it mean that a “mechanism independent of GABA-A receptors was responsible for the change in spike frequency”? What could such mechanism be? It would be interesting to hear potential speculations, also given the fact that the authors saw changes in GABA (both via staining in and via RNAseq and IPA) and not in glutamatergic synapsis, but then they go ahead and use glutamatergic NGN2 neurons for following experiments. Also there are no experiments with glutamatergic agonists/antagonists, so we do not know whether glutamatergic signaling can be manipulated differently in mutant cells.

In our previous manuscript, we expected the addition of bicuculline - a GABA-A antagonist - would suppress the calcium event rate in both genotypes, but to a greater extent in PACS1^(+/R203W) organoids. We hypothesized this intervention could return the calcium event rate of PACS1^(+/R203W) organoids to control levels because we had observed a greater density of GABAergic synapses through immunohistochemistry and transcriptomic analysis of chloride cotransporters suggested the GABA switch from excitatory to inhibitory had not yet occurred by day 80. Further discussion on our interpretation of the lack of differences between genotypes upon bicuculline administration is now added here: “It is possible no differential response to BCU was observed because GABA-B receptors play a larger role in PACS1 syndrome pathology, or the relatively low abundance of GABAergic neurons in our dorsal forebrain model dilutes our ability to detect GABAergic mechanisms of disease. Ultimately our model system was not best suited to interrogate GABAergic interneuron properties as we used the dorsal forebrain organoid protocol and a *NEUROG2* overexpression model to generate 2D neurons – both of which produce primarily excitatory neurons.”

We agree with Reviewer 2 that the use of only one antagonist was a significant limitation to our study and have now acknowledged this in the text: “A critical future direction is therefore utilizing more applicable models – such as ventral forebrain organoids, or a 2D *NEUROG2*-derived excitatory neuron and *ASLC/DLX*-derived inhibitory neuron coculture system – in conjunction with a diverse panel of agonists and antagonists to better understand the relative contributions role of excitatory and inhibitory neurons in PACS1 syndrome pathology.” Baclofen is indeed a good candidate as its use has been shown to improve phenotypes of Fragile X Syndrome and 16p11.2 mouse models of autism (PMID 36167501 and 28984295). We thank Reviewer 2 for the suggestion.

We had not intended to imply that GABAergic transmission was completely unaffected in PACS1 syndrome pathology, merely that our results warranted deeper investigation into electrophysiological deficits in glutamatergic neurons. In fact - as Reviewer 2 points out - results such as increased GABAergic density do implicate a role of GABAergic signaling in PACS1 syndrome. Our models were not optimally suited for studying this as very few GABAergic cells were found in day 80 organoids since we utilized a dorsal forebrain differentiation protocol. We intend to pursue this in future work and have begun generation of ventral forebrain organoids to better model the effects of p.R203W PACS1 on GABAergic neurons. We now discuss all this at length in the fourth paragraph of the discussion, provided here as well for convenience: “In addition, we reported an increase in GABAergic synaptic density and an upregulation of DEGs involved in GABA-A signaling such as *GABRG3*, but largely focused on deficits in glutamatergic neurons after application of one GABA-A receptor antagonist BCU did not differentially affect PACS1^(+/R203W) organoids. The role of GABAergic interneurons in PACS1 syndrome pathology is a key area of future exploration as an abundance of literature points to their role in ASD pathology.^{42,75–80} It is possible no differential response to BCU was observed because GABA-B receptors play a larger role in PACS1 syndrome pathology, or the relatively low abundance of GABAergic neurons in our dorsal forebrain model dilutes our ability to detect GABAergic mechanisms of disease. Ultimately our model system was not best suited to interrogate GABAergic interneuron properties as we used the dorsal forebrain organoid protocol and a *NEUROG2* overexpression model to generate 2D neurons – both of which produce primarily excitatory neurons. A critical future direction is therefore utilizing more applicable models – such as ventral forebrain organoids, or a 2D *NEUROG2*-derived excitatory neuron and *ASLC/DLX*-derived inhibitory neuron coculture system – in conjunction with a diverse panel of agonists and antagonists to better understand the relative contributions role of excitatory and inhibitory neurons in PACS1 syndrome pathology. Eventually, complementary studies with a model that has functional circuits between diverse brain regions are also needed to better understand how phenotypes observed here fit into an *in vivo* system. For example, increased density of GABAergic synapses is initially a counterintuitive mechanism of epilepsy, but it could be temporally or spatially restricted in a manner that initiates epileptiform activity *in*

vivo. Collaborative efforts within the PACS1 Syndrome Research Foundation²⁹ are therefore underway to generate a mouse model in order to better characterize the electrophysiological deficits across different brain regions, the resulting cognitive impact, and the safety of candidate therapies.”

Several pieces of data suggest potential GABA dysfunction, but then since there is no difference in the calcium response in mutant organoids compared to controls after the application of the GABA antagonist, the authors disregard all these evidence and in the discussion put forward only the idea of glutamatergic defects. I am not sure I agree with this strategy. I think that the evidence regarding GABA should also be mentioned. A recent paper from Arlotta’s lab suggested that defective development of GABA neurons could underlie the pathogenesis of ASD (PMID: 35110736). Perhaps this is worth mention in the discussion, especially since several of the findings of Rylaarsdam et al also suggest a dysregulation of GABA neurons. I also do not understand why the authors said that the differences in GABA is not maintained at later time points. The synaptic marker staining was done at a late time point (day 79 according to the figure legend of fig 4c).

We thank Reviewer 2 for bringing this to our attention. We had intended to focus our manuscript on the glutamatergic neurons as our models were best suited to understand glutamatergic neuron pathology, but we see how our writing could be perceived as disregarding the potential involvement of GABAergic dysfunction to PACS1 syndrome. On the contrary, we agree that GABA pathology in PACS1 syndrome is an intriguing and valuable area of future investigation given our own results and a plethora of literature supporting the contribution of GABAergic neurons to ASD pathology (in addition to PMID: 35110736, we also feel 34145239, 21068835, 11450824, 34848882, 33390904, and 19719810 are relevant to the discussion). We have therefore taken the advice of Reviewer 2 and devoted the entire fourth paragraph of the discussion (see previous answer) towards expanding on the importance of GABAergic signaling to PACS1 syndrome. We hope this revised discussion highlights the investigation of p.R203W PACS1’s impact on GABAergic signaling as a critical area of future investigation.

FigS9a. I do not understand why in some cases the curves are shorter or not existent. Please clarify.

We apologize for the confusion. This is now explained more clearly in the figure legend: “Missing values are either due to either: the neurons not yet being mature enough to elicit network bursting activity; the line not being included in the MEA experiment; or omission of the data upon observation of excessive clumping or other indicators of poor viability.”

Based on the MEA profile of NGN2 neurons, the authors assume that the defects are only in glutamatergic neurons. However, similar changes might be seen using GABA neurons. This is in fact quite likely and cannot be dismissed by the authors. So we do not know if GABA neurons are affected or not. Moreover, there are systems that can be used to easily generate GABA neurons (PMID: 28504679) and study their relation to glutamatergic neurons (<https://doi.org/10.1101/2020.05.07.082453>), so perhaps this can be considered for future studies or to be added in the discussion.

We completely agree with Reviewer 2 that the impact of the p.R203W variant in PACS1 on GABAergic neurons is a very intriguing avenue of exploration. This is a high priority and we have already begun to establish the Pasca ventral forebrain organoid protocol in order to further study GABAergic neurons. We will also strongly consider the protocol mentioned by Reviewer 2 for viral generation of GABAergic neurons for applications in which proliferative populations would be detrimental, such as MEAs. This study was primarily focused on glutamatergic neurons as a preliminary investigation into the potential mechanisms underlying patient symptoms such as epilepsy. We have now included an extensive acknowledgement to the potential

relevance of GABAergic neurons to PACS1 syndrome pathology and the need for future investigation in the fourth paragraph of the discussion (also previously copied above in response to Reviewer 2's comments).

In the abstract, the authors conclude that their work suggests “aberrant regulation of ionic flux”. I am not sure that this can be easily stated. They did not perform any analysis of channel activity (only general calcium imaging). I would modify those bold statements. Also, in the abstract they mention that their findings “underlies the neurological phenotypes experienced by patients”. Is this true? In the discussion the authors mention that the EEG reports from patients are “rare and variable”. So are their changes similar and consistent to what is reported in patients or not? I would suggest the authors to be more specific with respect to their interpretation and be more careful when describing the implications in the abstract.

We recognize that our statement suggesting aberrant regulation of ionic flux underlies patient symptoms was conjecture. While this is a valuable future direction, we agree with Reviewer 2 that analysis of channel activity such as patch clamping would be needed to support this claim and it is therefore too preliminary to include in the abstract. It is now removed. Our comparison to patient EEG recordings in the discussion is meant to highlight some potential correlations to our *in vitro* data, but we note that “these are very different systems, and it is possible that variable firing patterns in neurons initiate epileptic activity but are not reflected in EEG recordings. Further experiments with more diverse systems would be needed to fully make this connection.”

Reviewer #3 (Remarks to the Author):

The authors were responsive to most of the comments. However, after incorporating the new dataset, the revised manuscript switched the focus/conclusion. Therefore, some further questions were raised.

In Figure 2d, what is the color legend for “genotype” and “day”? Can the authors discuss/speculate why the cells of different genotypes do not segregate even within the same cell types.

We thank Reviewer 3 for bringing to our attention that the legend for Fig. 2d was incomplete. A legend for genotype and day have been added. For the heatmap in Figure 2d, we do not think it is unexpected that the cells of different genotypes do not segregate within the same cell types. Hierarchical clustering was performed between expression of the top 2,000 most variable genes between cell types. Differentially expressed genes could very well not be a strong component of this subset, and we would not hypothesize they would be given that no differences in cell type proportions were observed across multiple assays. We have expanded on this in the text: “Genotype is not a strong driver of clustering, indicating expression patterns of top variable features – e.g., canonical marker genes - are similar between groups. In line with this observation, transcriptomic analysis showed no differences in the proportions of cell types between PACS1^(+/R203W) and control organoids at either time point (Fig. 2b-c, shown by line in Fig. S8b-c).”

The revised manuscript adds a new dataset consisting of Day 88 organoids. First, however, it needs to be clarified how and if any data integration is performed. As we know, scRNA-seq from different experiments can be highly variable due to differences in sample preparation and sequencing technologies. Therefore, data integration is a critical step that involves combining scRNA-seq data from multiple experiments or using different cell lines to generate a unified dataset for downstream analyses.

We agree with Reviewer 3 that integration is a challenging and critical step of single-cell RNA sequencing data processing. We took great care to integrate batches properly by following recommendations from the Satija lab, the developers of the widely used Seurat R package. This is now discussed more thoroughly in the methods: “To integrate samples from different batches, anchor sets were identified between each organoid pair using canonical correlation analysis and then optimized through iterative pairwise integration using FindIntegrationAnchors (dims = 1:20) and IntegrateData functions (default parameters). On this integrated object, the functions ScaleData (regressing out the effect of percent.mt), RunPCA (default parameters), FindNeighbors (reduction = pca), FindClusters (resolution = 0.8), and RunUMAP (reduction = pca, n.neighbors = 30, min.dist = 0.2, dims = 1:20) were run to generate a processed Seurat object containing all samples for comparable visualization and analysis. Minor differences in technical metrics - like average number of genes per cell - were noted between day 40 and day 88 organoids due to advances in Northwestern’s sequencing core between sample submissions (Fig. S8d). We performed DEG analysis separately between organoids from different time points to avoid potential confounding factors caused by these technical differences, but for analyses where data was processed together such as Fig. 2d, the count matrix was subsetted to only include genes expressed in all organoids and re-normalized to 10,000 counts.”

In further support that samples were properly integrated, UMAP dimensionality reduction plots between samples show remarkably translatable coordinates of different neuron populations (see Fig. S8a below). This strongly suggests integration of anchor sets occurred as expected.

In figure 3b, what is the base of the log value? What is the cut-off?

We thank Reviewer 3 for pointing out this information was missing from Fig. 3b. The transformation was log base 10 and the adjusted p-value cutoff was 0.05. This information is now included in the figure legend.

How frequently did vGAT signal appear? Is it rare? The authors mentioned in the rebuttal letter, “the GABAergic population was not maintained at day 88”.

Though the GABAergic population was not abundant at day 88, expression of genes encoding GABAergic synaptic components is still observed. Below we show the expression of genes encoding the

proteins assessed in Fig. 4b-c in day 88 cells. *SYN1*, *DLG4* (which encodes PSD95), *SLC32A1* (which encodes vGAT), and *GPHN* are all prevalent transcripts.

The authors stated that “At both day 40 and 80, spontaneous TTX-susceptible calcium spikes were observed in PACS1^(+/+) and PACS1^(+R203W) organoids”. However, throughout the manuscript, no data was provided to demonstrate the TTX susceptibility.

We agree with Reviewer 3 that the susceptibility should have been quantified as opposed to qualitatively observed. We thank Reviewer 3 for bringing this to our attention as analysis of the TTX recordings revealed some noise that was escaping filtering parameters and resulting in false positive spikes. Upon this discovery, we completely re-analyzed the calcium imaging data with more stringent parameters, including a higher amplitude threshold and decreased maximum spike length. These adjustments successfully filtered out noise and TTX data is now included in Fig. S13b.

This extensive reanalysis of the data prompted the discovery that the three-way ANOVA previously used to calculate the effect of genotype, treatment, and day on calcium event rate was not the appropriate test as the metrics before and after addition of bicuculline were often repeated measures of the same sample. In addition, the calcium event rate distribution was no longer gaussian upon implementation of the more stringent filtering criteria, necessitating a non-parametric test. Calcium event frequency was therefore no longer different between *PACS1*^(+/+) and *PACS1*^(+R203W) organoids after performing individual Kruskal Wallis tests for the effect of day and genotype on the event rate of non-treated sections followed by a Benjamini Hochberg correction for multiple testing. We have therefore updated our interpretation of the calcium imaging results in the manuscript. Fig. 4e-i and Fig. S13b-f now illustrate this re-analyzed data.

*How similar are the excitatory neurons generated in 2D versus 3D? While dual SMAD inhibitors and WNT inhibitors were briefly included in the medium, it would be reasonable to confirm the cortical identity of the neurons generated in the 2D system, especially considering the induction of markers of sensory neurons and periphery nerve cells observed in *Ng2* neurons.*

Reviewer 3 brings up a very important point. To address these concerns, we performed immunohistochemistry for dorsal forebrain markers on *NEUROG2* neurons and found they express CUX2 and FOXG1 (Fig S14). While we acknowledge that growing evidence in the field suggests that *NEUROG2*-derived neurons are not as specified as once thought, the presence of glutamatergic neuron markers CUX2, FOXG1 and MAP2 suggests they are an applicable model to study the impacts of the p.R203W variant in excitatory neurons.

Are the DEGs identified using the organoid system also identified in the 2D neurons? Notably, the authors speculated that " DEGs such as KCNMB2, TRPC5, or CACNA2D3 could certainly underlie this phenotype" (in the 2D culture). Were the gene expression differences confirmed using qPCR or other methods?

We agree with Reviewer 3 that since much of our language in the previous manuscript assumed DEGs in mature glutamatergic neurons from organoids were translatable to the *NEUROG2* neurons, it would be valuable to experimentally validate this. We therefore performed bulk RNA sequencing on $n = 5$ PACS1^(+/+) and $n = 5$ PACS1^(+/R203W) *NEUROG2* neuron samples differentiated 50 days in culture. Unfortunately, experimental challenges resulted in the dropout of two control samples and results were deemed too preliminary to publish. Since the validity of a similar mechanism was not confirmed in our study, we drastically reduced language speculating which DEGs identified with single-cell analysis may lead to phenotypes observed in *NEUROG2*-neurons.

Spike frequency no longer shows a difference between the two genotypes. Is this due to the differences in the excitatory neurons generated? Figure 5 i-o, what does each dot represent?

After extensive reanalysis of the calcium imaging data, we no longer report an increased calcium event frequency in PACS1^(+/R203W) neurons. In addition, these are not completely comparable metrics due to the slow kinetics of the calcium sensor: the MEA system recorded at a rate of 12.5 kHz/channel while calcium imaging was recorded at only 2Hz as widths of calcium events were multiple seconds in length. However, Reviewer 3 brings up the valid concern that the two model systems are not identical and we have taken greater care to acknowledge the differences in our updated manuscript.

We apologize for the lack of clarity in Fig. 5i-o. The legend has been updated: "MEA metrics for neurons days 30-57 of differentiation. Dots represent averages from each cell line, while the smoothed curve shows genotype averages with a 95% confidence interval. $n = 7$ PACS1^(+/+) lines (three C1, three CRISPR A1, and one CRISPR A2) and $n = 7$ PACS1^(+/R203W) lines (three CRISPR R203W, three A1, and one A2) from three independent experiments."

Table S1 is now provided to show which cell lines were used in different experiments. I appreciate that the authors used all data collected from various lines for data analyses. However, it needs to be clarified how the data generated using different lines were analyzed, integrated, and interpreted. Were there any dropouts? It would be great to present data collected using different lines in the figures. It is both informative and essential to show the variability of experiment results.

We agree with Reviewer 3 that conveying variability between lines is important. To this end, we have included detailed results by cell line in Fig. S1-S3, S4d-e, S5f, S6b, S8, S10, and S12-S15. We have also added symbols to denote the parent organoid a metric was obtained from in Fig. S13c-f to illustrate inter-organoid variability in addition to cell line variability. Fig. S15a shows MEA results from different experiments, differentiations, and isogenic pairs. In the main figures, we primarily summarized results by genotype in order to illustrate the data as clearly as possible.

In addition, we have expanded in the methods how samples were integrated and removed from analysis. Proper data integration is particularly critical for the single-cell analysis section: “After sequencing, we analyzed the data using Seurat (v4.0.0) and monocle3 (v1.0.0) packages in R (v4.0.3-v4.1.1).^{78–80} In Seurat, data was filtered by number of features, counts, and percent mitochondrial reads ($nFeature_RNA > 500$ & $nFeature_RNA < 6500$ & $percent.mt < 15$ for day 40 and $nFeature_RNA > 1000$ & $nFeature_RNA < 8000$ & $nCount_RNA > 3000$ & $nCount_RNA < 40000$ & $percent.mt < 7.5$ for day 88 samples). One day 40 A2 sample was eventually omitted altogether due to extremely low number of cells captured and overall poor quality control metrics, but all other samples just had low-quality cells omitted... To integrate samples from different batches, anchorsets were identified between each organoid pair using canonical correlation analysis and then optimized through iterative pairwise integration using FindIntegrationAnchors (dims = 1:20) and IntegrateData functions (default parameters). On this integrated object, the functions ScaleData (regress out the effect of percent.mt), RunPCA (default parameters), FindNeighbors (reduction = pca), FindClusters (resolution = 0.8), and RunUMAP (reduction = pca, n.neighbors = 30, min.dist = 0.2, dims = 1:20) were run to generate a processed Seurat object containing all samples for comparable visualization and analysis. Minor differences in technical metrics - like average number of genes per cell - were noted between day 40 and day 88 organoids due to advances in Northwestern’s sequencing core between sample submissions (Fig. S8d). We performed DEG analysis separately between organoids from different time points to avoid potential confounding factors caused by these technical differences, but for analyses where data was processed together such as Fig. 2d, the count matrix was subsetted to only include genes expressed in all organoids and re-normalized to 10,000 counts.”

While MEA provides information on overall network activity, it doesn’t offer insights into the underlying mechanisms. For example, the phenotypes could be due to intrinsic deficits of the mutant neurons as indicated by the changed channel protein expression. Patch-clamp would be more suitable and informative to probe the mechanism.

We certainly agree with Reviewer 3 that MEAs offer poor resolution compared to a technique such as patch clamping. This would be a particularly useful technique to interrogate specific channels of interest and a valuable future direction to complement this study. Patch clamping is now mentioned in multiple places in the text as a valuable area of future investigation: “Which channels are contributing the most to the increased inter-spike interval in mature glutamatergic neurons can be hypothesized by our DEG analysis but should be further investigated with pharmacological intervention studies and patch clamp recordings;” and in the discussion: “Patch clamp recording in combination with pharmacological modulation of candidate ion channels are critical future directions in order to better understand how p.R203W PACS1 impacts action potential dynamics, potentially revealing targets for therapeutic intervention.”

REVIEWERS' COMMENTS

Reviewer #1 (Remarks to the Author):

The authors have responded thoroughly to the concerns raised by the reviewers and put effort into generating new data and figures as well as re-analyzing their work. I think this work is of significance to the field and will impact future research on PACS1 syndrome. The methodology is sound and appropriate, and enough detail is provided to reproduce the work. I support the findings will be published in this journal.

One minor comment remains: FOXP1 and EMX1 appear as glutamatergic neuron markers. They are not, they label telencephalic and cortical identities respectively.

Reviewer #2 (Remarks to the Author):

The authors have addressed all my concerns. I only have minor remarks.

- Regarding the abstract, I think the information that the authors employed brain organoids is still missing. I would encourage the authors to highlight this aspect in the abstract to clarify which models they have used to reach their conclusions. For example they could write that "PACS1 p.R203W variant does not impact cytoarchitectural organization of cortical organoids" or something like this. I think that for the readers it would be interesting to know that the experiments have been conducted also in 3D models.
- The information regarding the iPSC lines is reported only in the results section ("Control 1 (C1) was derived from a healthy female, C2 from.." and so on). I would encourage the authors to please add these details and information also in the methods section.
- It is great to see that NGN2 neurons express PACS1. However, these data are only present in the rebuttal letter. I would suggest to the authors to please include these findings also in the manuscript, as they provide important support to their choice of neuronal model to investigate PACS1 pathology.

Reviewer #3 (Remarks to the Author):

The revised manuscript represents a significant improvement over previous versions. The authors have effectively addressed many of the reviewers' comments by providing additional details and conducting further analyses. Although not all questions raised by the reviewers could be fully addressed due to experimental constraints, the authors have appropriately modified the language to acknowledge the preliminary nature of these experiments. They have also highlighted the limitations and expanded the discussion to provide a greater perspective. From the reviewer's standpoint, this study has made considerable contributions in characterizing the phenotypes associated with the recurrent mutation r203W in PACS1 across multiple systems. The findings offer valuable insights that are likely to be of broad interest to the field.

Re: NCOMMS-22-18984A-Z

We would like to thank the editor and reviewers for their efforts and are thrilled to hear our manuscript is accepted in principle. We sincerely appreciate the reviewers' attention to detail as their remaining suggestions have helped us produce a more thorough and accurate manuscript. Our point-by-point responses to italicized reviewer comments can be found below in blue regular font. All text changes and additions made to the manuscript are also highlighted in blue font throughout both the main manuscript and supplementary documents.

Reviewer #1 (Remarks to the Author):

The authors have responded thoroughly to the concerns raised by the reviewers and put effort into generating new data and figures as well as re-analyzing their work. I think this work is of significance to the field and will impact future research on PACS1 syndrome. The methodology is sound and appropriate, and enough detail is provided to reproduce the work. I support the findings will be published in this journal.

One minor comment remains: FOXG1 and EMX1 appear as glutamatergic neuron markers. They are not, they label telencephalic and cortical identities respectively.

We sincerely thank Reviewer 1 for bringing this inaccuracy in to our attention. Headings in Figure S7c have been changed to reflect *FOXG1* and *EMX1* as telencephalic and cortical identities respectively. In addition, the text has been updated to properly describe *FOXG1* and *EMX1* where they are mentioned. For example: "After 40 days of differentiation, organoids of both genotypes strongly expressed dorsal forebrain markers such as the telencephalic protein FOXG1, cortical protein EMX1, and dorsal precursor protein PAX6, but lacked mature upper cortical layer marker SATB2." The supplementary text has also been changed, for example the legend of Figure S7c now reads: "glutamatergic neurons express telencephalic marker *FOXG1* and cortical transcript *EMX1*."

Reviewer #2 (Remarks to the Author):

The authors have addressed all my concerns. I only have minor remarks.

Regarding the abstract, I think the information that the authors employed brain organoids is still missing. I would encourage the authors to highlight this aspect in the abstract to clarify which models they have used to reach their conclusions. For example they could write that "PACS1 p.R203W variant does not impact cytoarchitectural organization of cortical organoids" or something like this. I think that for the readers it would be interesting to know that the experiments have been conducted also in 3D models.

The authors thank Reviewer 2 for pointing out this information was missing from the abstract. We agree it is important to highlight the bulk of this work was conducted in a 3D model of the developing human brain and have modified the following portion of the abstract: "Here we differentiated stem cells towards neuronal models including cortical organoids to investigate the impact of the PACS1 syndrome-causing variant on neurodevelopment."

The information regarding the iPSC lines is reported only in the results session (“Control 1 (C1) was derived from a healthy female, C2 from..” and so on). I would encourage the authors to please add these details and information also in the methods section.

We thank Reviewer 2 for bringing to our attention that the nomenclature of our lines used throughout the paper was not present in the methods, nor was the sex of the lines listed. We have revised this section of the methods as follows: “This study utilized five PACS1^(+/+) and three PACS1^(+/R203W) induced pluripotent stem cell (iPSC) lines in total. Two control lines derived from a healthy female (C1; GM03651) and male (C2; GM03652) were purchased from fibroblasts from Coriell and reprogrammed to pluripotency by the Northwestern University Stem Cell Core using Sendai viral vectors containing OCT4, SOX2, KLF4 and CMYC⁸¹. A third control line was derived from an unaffected mother (C3; S033751*B/GM27160) of a daughter with PACS1 syndrome (A2; S033745*B/GM27159). These lines were purchased as iPSCs from Coriell but are now available at Wicell (PACS1001i-GM27160 and PACS1002i-GM27159). An additional male patient line (A1; PACS1003i-GM27161) was purchased as iPSCs from WiCell. Three isogenic pairs were generated in total using CRISPR/Cas9 gene editing technology by introducing the heterozygous variant which leads to p.R203W into the C1 background (CRISPR R203W) and correcting the variant in both patient lines (CRISPR A1 and CRISPR A2; **Fig. S1**).”

It is great to see that NGN2 neurons express PACS1. However, these data are only present in the rebuttal letter. I would suggest to the authors to please include these findings also in the manuscript, as they provide important support to their choice of neuronal model to investigate PACS1 pathology.

We agree with Reviewer 2 that the expression of PACS1 in NGN2/NEUROG2-derived neurons is critical to show in order to support their use as an appropriate model to study PACS1 syndrome. Fig. S14a now shows PACS1 protein expression in day 14 NEUROG2-derived neurons. We have updated this reference in the text: “Neurons generated with this approach (Fig. 5b) express PACS1 (Fig. S14a), develop complex morphology (Fig. 5c-d), and express neuronal forebrain markers including CUX1 and FOXG1 (Fig. S14b-c).”

Reviewer #3 (Remarks to the Author):

The revised manuscript represents a significant improvement over previous versions. The authors have effectively addressed many of the reviewers' comments by providing additional details and conducting further analyses. Although not all questions raised by the reviewers could be fully addressed due to experimental constraints, the authors have appropriately modified the language to acknowledge the preliminary nature of these experiments. They have also highlighted the limitations and expanded the discussion to provide a greater perspective. From the reviewer's standpoint, this study has made considerable contributions in characterizing the phenotypes associated with the recurrent mutation r203W in PACS1 across multiple systems. The findings offer valuable insights that are likely to be of broad interest to the field.

We are pleased to read Reviewer 3 believes the revised manuscript makes significant contributions to the field while simultaneously acknowledging where further research is needed. We sincerely thank Reviewer 3 for their contributions throughout this process.